# Disobeying Directions: Switching Random Walk Filters for Unsupervised Node Embedding Learning on Directed Graphs

**Ciwan Ceylan**                                                                  *ciwan@kth.se*
*Division of Robotics, Perception and Learning*
*KTH Royal Institute of Technology*

**Kambiz Ghoorchian**                                            *kambiz.ghoorchian@seb.com*
*SEB Group*

**Danica Kragic**                                                                  *dani@kth.se*
*Division of Robotics, Perception and Learning*
*KTH Royal Institute of Technology*

**Reviewed on OpenReview:** *https://openreview.net/forum?id=yngjRgVA5A*

## Abstract

Unsupervised learning of node embeddings for directed graphs (digraphs) requires careful handling to ensure unbiased modelling. This paper addresses two key challenges: (1) the obstruction of information propagation in random walk and message-passing methods due to local sinks, and (2) the representation of multiple multi-step directed neighbourhoods, arising from the distinction between in- and out-neighbours. These challenges are interconnected—local sinks can be mitigated by treating the graph as undirected, but this comes at the cost of discarding all directional information. We make two main contributions to unsupervised embedding learning for digraphs. First, we introduce ReachNEs (Reachability Node Embeddings), a general framework for analysing embedding models and diagnosing local sink behaviour on digraphs. ReachNEs defines the reachability filter, a matrix polynomial over normalized adjacency matrices that captures multi-step, direction-sensitive proximity. It unifies the analysis of message-passing and random walk models, making its insights applicable across a wide range of embedding methods. Second, we propose DirSwitch, a novel embedding model that resolves both local sink bias and neighbourhood multiplicity via switching random walks. These walks use directed edges for local steps, preserving directional structure, then switch to undirected edges for long-range transitions, enabling escape from local sinks and improving information dispersal. Empirical results on node classification benchmarks demonstrate that DirSwitch consistently outperforms state-of-the-art unsupervised digraph proximity embedding methods, and also serves as a flexible digraph extension for self-supervised graph neural networks.

## 1 Introduction

Directed graphs (digraphs) and node embeddings are both essential modelling tools for graph data. Digraphs are ubiquitous for describing networks with asymmetric relationships, such as citation networks (McCallum et al., 2000), online dating platforms (Takac & Zabovsky, 2012), financial transactions (Fire & Guestrin, 2020), brain connectomes (Winding et al., 2023), and the Internet (Faloutsos et al., 1999). Similarly, node embeddings are key to solving machine learning and data mining tasks on graphs, including node classification (Hou et al., 2023), node clustering (Wang et al., 2017b), graph alignment (Heimann et al., 2018), and link prediction (Virinchi & Saladi, 2023). Despite their combined prevalence, the field of unsupervised embedding learning for digraphs remains under-explored. Most research has focused on undirected graphs and (semi-)supervised learning, even though task-specific labelled data is often scarce (Hu et al., 2020b).

Embedding modelling for digraphs presents several challenges. A primary requirement is the ability to capture the asymmetric nature of digraphs. Conventional spectral-based approaches fall short, as asymmetric matrices do not admit eigendecomposition (Zhang et al., 2021b). As a result, it is common practice to treat digraphs as undirected, thereby discarding important relational information (Rossi et al., 2023).

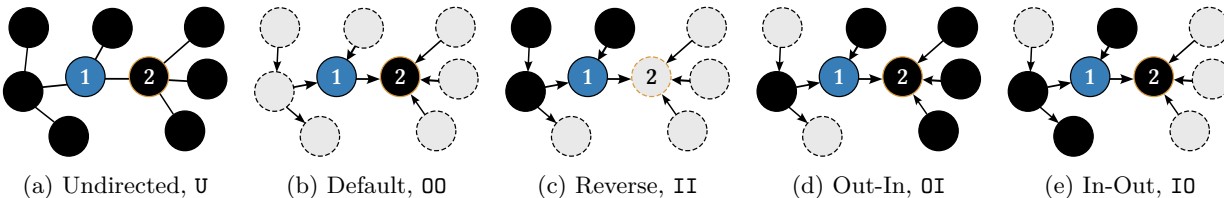

(a) Undirected, `U`  (b) Default, `OO`  (c) Reverse, `II`  (d) Out-In, `OI`  (e) In-Out, `IO`

Figure 1: Visualization of local sinks and neighbourhood multiplicity. Each digraph shows a valid definition of the 2-step neighbourhood of node 1, highlighted in black. Note that each directed neighbourhood is unique, carrying independent information. (a) uses undirected edges, resulting in the maximum number of reachable nodes at the expense of the digraph structure. (b) shows the neighbourhood using the default, outgoing edge directions. The local sink, node 2, obstructs the information flow, so the neighbourhood consists of a single node. In (c), the reverse, incoming edges are followed, and (d) and (e) use mixed edge directions. (d) uses outgoing edges for the first step and incoming edges for the second, and vice versa for (e).

Recent research on digraph node embeddings has focused on adapting random walk and message-passing approaches to capture digraph asymmetry (Zhou et al., 2017; Khosla et al., 2020; Tong et al., 2020; Zhang et al., 2021b; Virinchi & Saladi, 2023; Rossi et al., 2023). These approaches rely on information propagation between nodes to generate embeddings that represent the connectivity of multi-step node neighbourhoods. To account for digraph asymmetry, these models separate propagation along outgoing (default) edges from reverse (incoming) edges, creating two sets of paired node embeddings.

While these models address digraph asymmetry, other challenges have been overlooked. This work tackles two such challenges: *local sinks* and *neighbourhood multiplicity*, visualized in Figure 1. The unidirectionality of digraph edges means information flows in only one direction. As a result, digraphs contain *local sinks*, i.e., nodes without outgoing edges. As shown in Figure 1b, these sinks obstruct information propagation and hinder exploration of multi-step neighbourhoods. This leads to a disproportionate influence of sink nodes, resulting in biased embedding models. For instance, in Figure 1b, node 1 only sees the sink node (node 2).

As highlighted in Figure 1, the multi-step neighbourhood in digraphs is not uniquely defined. This multiplicity arises from the fact that each node has two distinct sets of neighbours: out-neighbours and in-neighbours. When multi-step neighbourhoods are considered, this multiplicity is exponentially amplified, as each node can have up to $2^r$ distinct $r$-step neighbourhoods. The challenge for unsupervised node embedding models is to represent this multitude of distinct neighbourhoods as accurately as possible, since it is generally unknown a priori which neighbourhoods are key for solving downstream tasks.

In this work, we present two key contributions to address the challenges of unsupervised embedding learning for digraphs: (1) ReachNEs (Reachability Node Embeddings), a unifying framework for analysing node embedding models, and (2) DirSwitch, a flexible approach for mitigating the issues of local sinks and neighbourhood multiplicity, applicable to both random walk and message-passing embedding models.[1]

ReachNEs is a mathematical framework designed to study and analyse the challenges of digraph node embedding modelling. Flexibility and analytical tractability are prioritized in its design. To achieve this, ReachNEs is built around the *reachability graph filter*, a matrix computed as a polynomial of normalized adjacency matrices, capturing asymmetric node proximity through multi-step random walk transition probabilities between node pairs in the graph.

The reachability matrix unifies random walk and message-passing embedding models. To generate random walk-based proximity embeddings (Zhou et al., 2017; Zhu et al., 2021a), matrix factorization is used to decompose the reachability matrix into low-dimensional factors. Conversely, the reachability matrix can serve as a neighbourhood smoothing filter to produce linear message-passing embeddings (Wu et al., 2019). This versatility makes the insights drawn from ReachNEs analysis applicable to a wide range of embedding techniques.

The ReachNEs framework is also flexible with respect to modelling different neighbourhood scales, supporting multi-scale embeddings, which are crucial in the unsupervised setting (Rozemberczki et al., 2021). This control is provided by the coefficients of the reachability matrix's defining polynomial, which are interpreted as random walk length probabilities.

---

[1]Our source code: `https://github.com/ciwanceylan/dirswitch-experiments-tmlr2025.git`

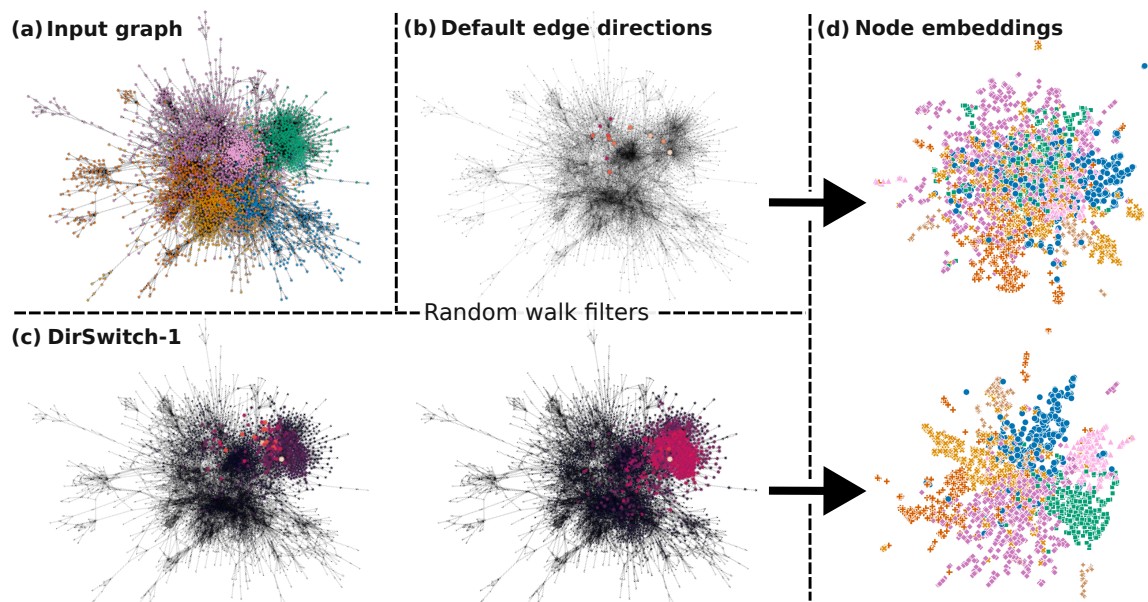

Figure 2: A qualitative illustration of DirSwitch. (a): The Cora-ML citation graph, with nodes coloured by class label (for visualization purposes only; not used in the model). (b) and (c): Random walk transition probabilities from a specific initialization node. The graph in (b) shows the default edge directions, while the graphs in (c) employ our DirSwitch-1 approach. DirSwitch-1 performs one directed step using either default (*left*) or reverse (*right*) edge directions, followed by undirected transitions. (d): UMAP (Sainburg et al., 2021) visualizations of the resulting node embeddings. The DirSwitch-1 embedding space better reflects the spatial ground truth class separation, demonstrating its improved neighbourhood smoothing on digraphs.

We use ReachNEs to quantify the effects of local sinks on information flow and to mathematically characterize the multiplicity of distinct directed neighbourhoods in digraphs. This analysis reveals a tension between these two issues. The local sink problem can be easily resolved by treating digraphs as undirected, allowing information to flow freely within each weakly connected component. However, this comes at the cost of losing the distinction between directed neighbourhoods, reducing the expressivity of the embedding model. Therefore, we seek a model that can address local sinks while preserving the capture of directed neighbourhoods.

Our second contribution, DirSwitch, achieves this balance. The core idea of DirSwitch is to decouple local, short-range random walk behaviour from global, long-range behaviour. Specifically, DirSwitch preserves local edge direction expressivity by performing directed random walks in the initial steps. It then switches to treating the graph as undirected, allowing information to propagate freely and escape local sinks. The effectiveness of this approach is illustrated in Figure 2.

We empirically evaluate DirSwitch both within and beyond the ReachNEs framework. First, we verify the that DirSwitch both improves information propagation on digraphs compared to fully directed approaches, without comprimisign the representation of local neighbourhood multiplicity.

We then demonstrate that this results in higher-quality embeddings and increased effectiveness by benchmarking DirSwitch on 14 standard node classification benchmark datasets. Compared to ReachNEs models that do not distinguish between local and global random walk behaviours, DirSwitch either achieves the highest accuracy or performs within one standard deviation of the best result across all datasets. This holds true for both the random walk-based proximity embedding setting and the message-passing setting. Importantly, DirSwitch performs well on both homophilic and heterophilic datasets (Zhu et al., 2020). This is desirable as it implies a broad applicability of the DirSwitch in real-world unsupervised settings where the validity of a homophily assumption is in question (Wang et al., 2023).

To further assess DirSwitch's practical effectiveness, we evaluate it against recent state-of-the-art digraph methods. On six proximity embedding benchmarks without node attributes, DirSwitch achieves the highest accuracy on all datasets except one, where it performs within one standard deviation of the best.

For message-passing embeddings, we apply DirSwitch as a digraph extension of the GraphSAGE graph neural network (GNN) model (Hamilton et al., 2017) and integrate it with self-supervised loss functions, which have previously been limited to undirected graphs (Zhang et al., 2021a; Hou et al., 2023). This demonstrates how insights from ReachNEs can be leveraged to enhance embedding models more broadly. As a baseline, we use the recent digraph GNN extension by Rossi et al. (2023), designed for supervised learning. Again, DirSwitch achieves higher accuracy on all but one dataset, underscoring the need for additional considerations when modelling digraphs in the unsupervised setting.

## 2 ReachNEs: Reachability Node Embeddings

This section introduces the ReachNEs framework for learning and analysing node embeddings in digraphs. It consists of two components: the reachability graph filter, which represents multi-step node relations, and reduction methods, which extract embeddings. Section 2.1 derives reachability filters from random walks, while Section 2.2 describes the walk length distributions that determine the filter's smoothing properties. Finally, Section 2.3 presents reduction methods that unify message-passing and proximity-based embeddings.

### 2.1 The random walk reachability filter

Let $\mathcal{G} = (\mathbb{N}, \mathbb{M})$ be a directed, potentially weighted graph without self-loops, where $\mathbb{N}$ is the set of nodes and $\mathbb{M}$ is the set of edges. The graph has $n = |\mathbb{N}|$ nodes and $m = |\mathbb{M}|$ edges. The adjacency matrix $\boldsymbol{A} \in \mathbb{R}^{n \times n}$ is defined as follows, along with the out-degree and in-degree of a node $j$:

$$A_{i,j} = \begin{cases} W_{j \mapsto i} & \text{if } (j,i) \in \mathbb{M}, \\ 0 & \text{otherwise,} \end{cases} \qquad \deg_{\mathrm{O}}(j) = \sum_{k=1}^{n} A_{k,j}, \qquad \deg_{\mathrm{I}}(j) = \sum_{k=1}^{n} A_{j,k}. \qquad (1)$$

Here, $W_{j \mapsto i} \in \mathbb{R}_{\geq 0}$ is a nonnegative weight associated with the edge $(j,i)$. Note that the columns of $\boldsymbol{A}$ correspond to out-neighbourhoods, while the rows correspond to in-neighbourhoods.

We treat undirected graphs as a special case of the above definitions, where $A_{i,j} = A_{j,i}$ for all edges in $\mathbb{M}$. Consequently, a digraph can be transformed into its corresponding undirected graph by adding the reverse edge $(j,i)$ to $\mathbb{M}$ for every edge $(i,j) \in \mathbb{M}$, forming the edge set $\mathbb{M}_{\mathrm{undir}}$. In terms of the adjacency matrix, this symmetrization is achieved by summing $\boldsymbol{A}$ and its transpose $\boldsymbol{A}^{\mathsf{T}}$, $\boldsymbol{A}_{\mathrm{undir}} = \boldsymbol{A} + \boldsymbol{A}^{\mathsf{T}}$.

Next, we define three *random walk normalized adjacency matrices*, denoted as $\boldsymbol{A}_{\mathrm{O}}$, $\boldsymbol{A}_{\mathrm{I}}$, and $\boldsymbol{A}_{\mathrm{U}}$, which we collectively refer to as $\boldsymbol{A}_{*} \in [0,1]^{n \times n}$. Each of these matrices is a column-stochastic matrix (Horn & Johnson, 2012, Ch. 8.7) and serves as a random walk state transition matrix. Specifically, given a probability state vector $\boldsymbol{p}^{(k)} \in [0,1]^{n}$, where $p_i^{(k)}$ represents the probability of a random walker being at node $i$ after $k$ steps, the state vector after $k+1$ steps is computed as $\boldsymbol{p}^{(k+1)} = \boldsymbol{A}_{*} \boldsymbol{p}^{(k)}$.

Each normalized adjacency matrix defines a different random walk behaviour with respect to the edge directions. For $\boldsymbol{A}_{\mathrm{O}}$, a random walker follows the default edge directions, transitioning from a node to its neighbours along outgoing edges. Conversely, transitions using $\boldsymbol{A}_{\mathrm{I}}$ follow the *reverse* edge directions, meaning the walker traverses incoming edges. For $\boldsymbol{A}_{\mathrm{U}}$, the edges are treated as undirected, allowing the random walker to move across both outgoing and incoming edges. These behaviours are formally defined as follows:

$$A_{\mathrm{O}\,i,j} = \begin{cases} 1 & \text{if } i = j \text{ and } \deg_{\mathrm{O}}(j) = 0, \\ \frac{A_{i,j}}{\deg_{\mathrm{O}}(j)} & \text{otherwise,} \end{cases} \qquad A_{\mathrm{I}\,i,j} = \begin{cases} 1 & \text{if } i = j \text{ and } \deg_{\mathrm{I}}(j) = 0, \\ \frac{A_{j,i}}{\deg_{\mathrm{I}}(j)} & \text{otherwise,} \end{cases}$$

$$A_{\mathrm{U}\,i,j} = \begin{cases} 1 & \text{if } i = j \text{ and } \deg(j) = 0, \\ \frac{A_{\mathrm{undir}\,i,j}}{\deg(j)} & \text{otherwise,} \end{cases} \qquad \deg(i) = \sum_{k=1}^{n} A_{\mathrm{undir}\,k,i}.$$

To ensure column stochasticity, diagonal elements are set to 1 for nodes without legal edges to follow. Such nodes are referred to as *sink nodes*, as a random walker gets stuck on these nodes. For digraphs without sink nodes, the above definitions can be expressed compactly in matrix form:

$$\boldsymbol{A}_{\mathrm{O}} = \boldsymbol{A}\boldsymbol{D}_{\mathrm{O}}^{-1}, \qquad\qquad \boldsymbol{A}_{\mathrm{I}} = \boldsymbol{A}^{\mathsf{T}}\boldsymbol{D}_{\mathrm{I}}^{-1}, \qquad\qquad \boldsymbol{A}_{\mathrm{U}} = \boldsymbol{A}_{\mathrm{undir}}\boldsymbol{D}^{-1},$$

where $\boldsymbol{D}_{\mathrm{O}}$, $\boldsymbol{D}_{\mathrm{I}}$, and $\boldsymbol{D}$ are diagonal matrices of the respective node degrees.

Each $\boldsymbol{A}_*$ describe immediate neighbourhoods, which can be extended to capture multi-step connectivity through matrix powers. Specifically, the $j$th column of the power matrix $\boldsymbol{A}_*^k$ encodes the transition probabilities of a random walk of length $k$ initialized at node $j$. This follows directly from the state transition formula $\boldsymbol{p}^{(k+1)} = \boldsymbol{A}_* \boldsymbol{p}^{(k)}$. Accordingly, we denote element $(i, j)$ of $\boldsymbol{A}_*^k$ as $P(j \rightsquigarrow i | k)$, representing the probability of transitioning from node $j$ to node $i$ in exactly $k$ steps.

Each matrix power $\boldsymbol{A}_*^k$ provides a snapshot of transition probabilities at a specific walk length $k$. However, any fixed $k$ inherently restricts the description of node neighbourhoods and introduces potential biases. For example, setting $k = 2$ might fail to capture immediate neighbours if no 2-step paths exist between them, resulting in zero transition probabilities for such nodes.

To mitigate the bias and sensitivity introduced by single walk lengths, we instead impose a probability distribution over the walk length: $P_w(k)$. Using this distribution, the node transition probabilities are computed by marginalizing over the walk length. This leads to the definition of the *reachability matrix* $\boldsymbol{R} \in \mathbb{R}^{n \times n}$, where each element $R_{i,j}$ represents the probability of transitioning from node $j$ to node $i$, with the walk length sampled from $P_w(k)$:

$$\boldsymbol{R}(\boldsymbol{A}_*; P_w) = \sum_{k=0}^{\infty} P_w(k) \boldsymbol{A}_*^k, \qquad R_{i,j}(\boldsymbol{A}_*; P_w) = P(j \rightsquigarrow i | P_w) = \sum_{k=0}^{\infty} P_w(k) P(j \rightsquigarrow i | k). \qquad (2)$$

Figure 3a shows an example of the reachability matrix $\boldsymbol{R}(\boldsymbol{A}_U; P_w)$ using the U.S. political blogs graph (Adamic & Glance, 2005). Each column represents the transition probabilities from a random walk starting at the corresponding node. The two prominent blocks reflect the political alignments in the dataset. Figure 3b visualizes one such column vector, overlaid on a ridiculogram of the graph. The separation between the two political communities appears clearly as two clusters.

## 2.2 Walk length distributions

As defined in Equation 2, the walk length distribution $P_w(k)$ controls the weighting of each adjacency power $\boldsymbol{A}_*^k$, thereby shaping the filter's smoothing behaviour. While ReachNEs is compatible with any such distribution, in this work we focus on four commonly used ones: geometric, binomial, Poisson, and uniform.

Table 1 lists the corresponding probability mass functions (pmfs), each parameterized by the expected walk length $\tau = \mathbb{E}[P_w]$. Figure 3c visualizes the pmfs for $\tau = 2$. We briefly describe each distribution below; further analysis appears in Appendix C.

The geometric distribution is central to the PageRank algorithm (Page et al., 1999), where $\alpha = \frac{\tau}{\tau+1}$ denotes the probability of continuing a walk, and $1 - \alpha$ the restart probability. It is monotonically decreasing for all $\tau$, and remains mode-centred at $k = 0$, even as it flattens with increasing $\tau$. Owing to PageRank's influence, the geometric distribution is widely used in embedding methods (Zhou et al., 2017; Yan et al., 2024).

The binomial distribution has a finite maximum walk length $K$, where $\alpha = \frac{\tau}{K}$ denotes the probability of stepping, and $1 - \alpha$ the probability of remaining in place. As discussed in Appendix C.5, this distribution naturally arises when self-loops are added to graphs, a common practice in graph convolutional networks (Kipf & Welling, 2017; Wu et al., 2019).

Table 1: Overview of the studied random walk length distributions $P_w(k; \tau)$, where $k$ represents the walk length variable. Each distribution is parameterized by the average walk length $\tau$. For readability, we use $\alpha$ as a substitution variable in the geometric and binomial distributions. The binomial distribution includes an additional parameter, $K$, representing the maximum walk length. Distribution plots are shown in Figure 3c.

| Name | Geometric | Binomial | Poisson | Uniform |
|------|-----------|----------|---------|---------|
| Abbreviation | Geom | Binom | Pois | $\mathcal{U}$ |
| $P_w(k; \tau)$ | $(1-\alpha)\alpha^k, \quad \alpha = \frac{\tau}{\tau+1}$ | $\binom{K}{k}(1-\alpha)^{K-k}\alpha^k, \quad \alpha = \frac{\tau}{K}$ | $e^{-\tau}\frac{\tau^k}{k!}$ | $\begin{cases} \frac{1}{2\tau+1} & \text{if } k \in [0, 2\tau], \\ 0 & \text{otherwise} \end{cases}$ |

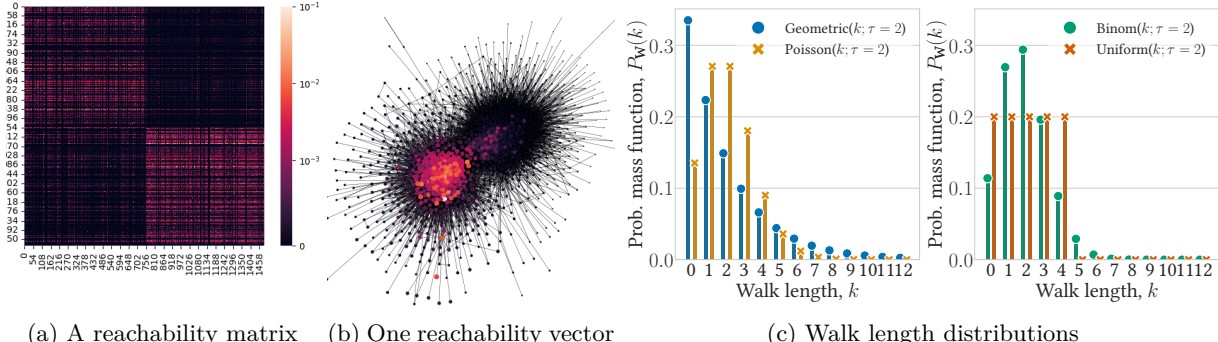

(a) A reachability matrix     (b) One reachability vector     (c) Walk length distributions

Figure 3: (a) The reachability matrix $\boldsymbol{R}(\boldsymbol{A}_{\mathtt{U}}; \mathrm{Pois}(\tau = 2))$ for the Polblogs graph. Each column encodes the transition probabilities of a random walk starting at the corresponding node. (b) A single column of $\boldsymbol{R}$ visualized on a ridiculogram of the graph, illustrating the locality of the reachability distribution. (c) The four walk length distributions used in this work, evaluated at $\tau = 2$: Geometric, Poisson, Binomial, and Uniform. These distributions weight the contribution of different walk lengths in the computation of $\boldsymbol{R}$.

In the limit $K \to \infty$, the binomial distribution converges to the Poisson distribution. The Poisson distribution has been used in structural embedding methods (Donnat et al., 2018; Zhu et al., 2021a; Ceylan et al., 2022) and connects reachability to heat diffusion on graphs.

Unlike the geometric distribution, both the binomial and Poisson distributions tend toward non-informative filters as $\tau \to \infty$. Their modes shift with $\tau$, emphasizing larger $k$ values, which leads to over-smoothing: columns of $\boldsymbol{R}$ become nearly uniform, reducing embedding discriminability. This effect, analysed in detail in Appendix C, plays a key role in interpreting some of our experimental results.

Finally, the *uniform distribution* is widely used in node embedding models, as it naturally arises from common negative sampling schemes (Perozzi et al., 2014; Qiu et al., 2018; Chanpuriya & Musco, 2020).

## 2.3 Reachability reduction into node embeddings

To complete the ReachNEs framework, the pairwise multi-step relationships encoded in the reachability matrix $\boldsymbol{R}$ must be transformed into node embeddings. We refer to these transformations as *reduction methods*, as they reduce the $n \times n$ matrix $\boldsymbol{R}$ into a lower-dimensional embedding matrix $\boldsymbol{Z} \in \mathbb{R}^{n \times p}$.

The reduction method serves two main purposes. First, it projects nodes into a lower-dimensional space ($p < n$), improving computational efficiency for downstream tasks and alleviating the curse of dimensionality (Hastie et al., 2009, Ch. 2.5). Second, the choice of reduction method encodes specific equivalence relationships into the embeddings (Zhu et al., 2021a). We explore two reduction approaches: proximity embeddings and message-passing embeddings, which capture structural and automorphic equivalence respectively.

### 2.3.1 Proximity embedding reduction and structural equivalence

Proximity embeddings capture mutual node connectivity and closeness, and are often referred to as *positional embeddings* in the context of undirected graphs (Zhu et al., 2021a). However, the asymmetric nature of digraphs makes the concept of "position" in a symmetric, Euclidean sense inappropriate. Instead, we define node proximity asymmetrically using the reachability matrix $\boldsymbol{R}(\boldsymbol{A}_*; \mathrm{P}_{\mathtt{w}})$, where the element $R_{i,j}$ quantifies the closeness of node $j$ to node $i$. The walk length distribution $\mathrm{P}_{\mathtt{w}}$ controls the resolution of this proximity.

To derive proximity embeddings from $\boldsymbol{R}$, we employ Singular Value Decomposition (SVD) (Golub & Van Loan, 2013, Ch. 2.4), a widely used technique in proximity embedding methods (Qiu et al., 2018; Zhu et al., 2021a). Let $\boldsymbol{U}, \boldsymbol{\Sigma}, \boldsymbol{V}^{\mathsf{T}} = \mathtt{SVD}(\boldsymbol{R})$ represent the SVD of $\boldsymbol{R}$, and let $\boldsymbol{U}_{:,:q}$, $\boldsymbol{\Sigma}_{:q,:q}$, and $\boldsymbol{V}_{:,:q}$ denote the $q$-truncation of $\boldsymbol{U}$, $\boldsymbol{\Sigma}$, and $\boldsymbol{V}$, respectively, where $q = p/2$. The proximity embeddings are then defined as:

$$\boldsymbol{Z} = \begin{bmatrix} \boldsymbol{Z_U} & \boldsymbol{Z_V} \end{bmatrix} \in \mathbb{R}^{n \times 2q}, \qquad \boldsymbol{Z_U} = \boldsymbol{U}_{:,:q}\sqrt{\boldsymbol{\Sigma}_{:q,:q}} \in \mathbb{R}^{n \times q}, \qquad \boldsymbol{Z_V} = \boldsymbol{V}_{:,:q}\sqrt{\boldsymbol{\Sigma}_{:q,:q}} \in \mathbb{R}^{n \times q}. \quad (3)$$

The proximity embeddings capture node proximity in two key ways. First, the inner product of the left and right embeddings approximates the reachability matrix: $\boldsymbol{R} \approx \boldsymbol{Z_U}\boldsymbol{Z_V}^\mathsf{T} = \boldsymbol{U}_{:,:q}\boldsymbol{\Sigma}_{:q,:q}\boldsymbol{V}_{:,:q}^\mathsf{T}$. Second, the Euclidean distance between embeddings in $\boldsymbol{Z}$ reflects the overlap of multi-step neighbourhoods. The latter is particularly important for downstream tasks such as clustering, where embedding distances are often utilized.

The following equality (derived in Appendix A.3) formalizes the relationship between reachability similarity and embedding distance:

$$\|\boldsymbol{R}_{i,:} - \boldsymbol{R}_{j,:}\|_2^2 + \|\boldsymbol{R}_{:,i} - \boldsymbol{R}_{:,j}\|_2^2 = \left\|(\boldsymbol{Z}_{i,:} - \boldsymbol{Z}_{j,:})\sqrt{\hat{\boldsymbol{\Sigma}}}\right\|_2^2, \qquad \hat{\boldsymbol{\Sigma}} = \begin{bmatrix} \boldsymbol{\Sigma}_{:q,:q} & \boldsymbol{0} \\ \boldsymbol{0} & \boldsymbol{\Sigma}_{:q,:q} \end{bmatrix}. \qquad (4)$$

The left-hand side measures the difference between rows and columns of $\boldsymbol{R}$ for nodes $i$ and $j$, quantifying the overlap of their multi-step neighbourhoods. The right-hand side represents the distance between their respective proximity embeddings in $\boldsymbol{Z}$. This equivalence implies that, for $q \leq \mathrm{rank}(\boldsymbol{R})$, two nodes have identical embeddings in $\boldsymbol{Z}$ if and only if their multi-step neighbourhoods perfectly overlap.

This property is particularly valuable for distance-based downstream tasks. However, its validity relies on the first term in the definition of $\boldsymbol{R}$ in Equation 2, i.e., $\mathrm{P_w}(k=0)\boldsymbol{I}_n$, being zero. This term introduces a constant, non-neighbourhood-dependent contribution that prevents structurally equivalent nodes from having identical embeddings. To address this, we compute $\boldsymbol{R}$ for proximity embeddings using a shifted walk-length distribution; meaning that the coefficients in Equation 2 are replaced with $\mathrm{P_w}(k-1)$, using $\mathrm{P_w}(-1) = 0$.

Another modification to the definition of $\boldsymbol{R}$ is the application of the elementwise thresholding function $f(R_{i,j}) = \log(\max(nR_{i,j}, 1))$ before performing the SVD. This contrast-enhancing function amplifies the difference between elements where $R_{i,j} < \frac{1}{n}$ and those where $R_{i,j} \in \left[\frac{1}{n}, \frac{1}{\log n}\right]$. This type of contrast enhancement has been shown to yield practical benefits for undirected graphs (Qiu et al., 2018; Chanpuriya & Musco, 2020; Zhu et al., 2021a), and we find that these advantages extend to digraphs as well.

**Structural equivalence.** Two nodes $i$ and $j$ are *structurally equivalent* if they have identical in- and out-neighbours, i.e., $\boldsymbol{A}_{i,:} = \boldsymbol{A}_{j,:}$ and $\boldsymbol{A}_{:,i} = \boldsymbol{A}_{:,j}$ (Borgatti & Everett, 1992). In undirected graphs, this implies that structurally equivalent nodes are at most two steps apart, highlighting the close connection between structural equivalence and node proximity.

Proximity ReachNEs embeddings preserve this equivalence under mild conditions. If nodes $i$ and $j$ are structurally equivalent and have nonzero in- and out-degrees, and if the walk length distribution satisfies $\mathrm{P_w}(k=0) = 0$, then $\boldsymbol{Z}_{i,:} = \boldsymbol{Z}_{j,:}$ holds as long as $q \leq \mathrm{rank}(\boldsymbol{R})$. The reason is straightforward: structural equivalence implies that the $i$-th and $j$-th rows and columns of $\boldsymbol{A}_*^k$ are identical for all $k \geq 1$, and thus also in $\boldsymbol{R} = \sum_{k \geq 1} \mathrm{P_w}(k)\boldsymbol{A}_*^k$. It follows from Equation 4 that the embeddings must be equal, since the diagonal elements of $\hat{\boldsymbol{\Sigma}}$ are positive under the assumption $q \leq \mathrm{rank}(\boldsymbol{R})$.

An exception occurs for sink nodes. When a node lacks outgoing edges, the normalization of $\boldsymbol{A}_0$ sets the corresponding diagonal entry to 1, violating the condition that $\boldsymbol{A}_{*i,:} = \boldsymbol{A}_{*j,:}$. As a result, structurally equivalent sink nodes (with identical incoming edges) can receive different embeddings. The same issue applies under $\boldsymbol{A}_\mathrm{I}$ for nodes without incoming edges.

This limitation is shared by many random-walk-based proximity embeddings (Perozzi et al., 2014; Grover & Leskovec, 2016; Zhou et al., 2017; Qiu et al., 2018; Khosla et al., 2020). A simple workaround is to assign the zero vector $\boldsymbol{0}$ to all zero-degree sink nodes in post-processing. Whether this improves downstream performance remains an open research question.

### 2.3.2 Message-passing embedding reduction and automorphic equivalence

Message-passing embedding reduction requires an initial set of node representations, i.e., a matrix of node attribute vectors $\boldsymbol{X} \in \mathbb{R}^{n \times d}$. These attributes typically supplement the graph structure, such as text embeddings of paper abstracts in citation graphs (Hu et al., 2020a). Alternatively, $\boldsymbol{X}$ may consist of graph-derived properties, such as node degrees or local clustering coefficients.

The ReachNEs message-passing embeddings are smoothed versions of the node attribute vectors, with the columns of the reachability matrix $\boldsymbol{R}$ defining the multi-step smoothing filter over each node's local neighbourhood. Thus, initial $d$-dimensional message-passing embeddings are straightforwardly obtained via the matrix multiplication $\boldsymbol{Z} = \boldsymbol{R}^\mathsf{T} \boldsymbol{X}$. To further reduce the dimensionality to $p$ dimensions, techniques like PCA (Murphy, 2012, Ch. 12.2) can be applied, yielding the embeddings $\boldsymbol{Z} = \text{PCA}(\boldsymbol{R}^\mathsf{T} \boldsymbol{X})$.

**Connection to message-passing.** Message-passing is commonly used in the context of graph neural networks (Gilmer et al., 2017). In typical descriptions, each node receives messages from its neighbours in the form of embedding vectors, which are aggregated and then used to update its own embedding representation. Given this, it may not be immediately clear how the linear model $\boldsymbol{R}^\mathsf{T} \boldsymbol{X}$ relates to message-passing.

To establish this connection, we expand $\boldsymbol{R}^\mathsf{T} \boldsymbol{X}$ by bringing the transpose inside the sum of Equation 2:

$$\boldsymbol{R}^\mathsf{T} \boldsymbol{X} = \left(\sum_{k=0}^{\infty} \mathrm{P}_{\mathtt{w}}(k) \boldsymbol{A}_*^k\right)^\mathsf{T} \boldsymbol{X} = \sum_{k=0}^{\infty} \mathrm{P}_{\mathtt{w}}(k) \left(\boldsymbol{A}_*^k\right)^\mathsf{T} \boldsymbol{X} = \sum_{k=0}^{\infty} \mathrm{P}_{\mathtt{w}}(k) \boldsymbol{A}_*^{\mathsf{T} k} \boldsymbol{X}. \tag{5}$$

Next, we let $\boldsymbol{H}^{(0)} = \boldsymbol{X}$ and $\boldsymbol{Z}^{(0)} = \mathrm{P}_{\mathtt{w}}(0)\boldsymbol{X}$, and express the sum in Equation 5 as an iterative algorithm:

$$\texttt{For } k \in \{1, 2, \dots\}, \quad \begin{cases} \boldsymbol{H}^{(k)} & = \boldsymbol{A}_*^\mathsf{T} \boldsymbol{H}^{(k-1)}, \\ \boldsymbol{Z}^{(k)} & = \boldsymbol{Z}^{(k-1)} + \mathrm{P}_{\mathtt{w}}(k)\boldsymbol{H}^{(k)}. \end{cases} \tag{6}$$

This iterative formulation connects directly to message-passing. To clarify the connection, consider the update $\boldsymbol{H}^{(k)} = \boldsymbol{A}_{\mathtt{U}}^\mathsf{T} \boldsymbol{H}^{(k-1)}$ for the undirected random walk matrix. Assuming no sink nodes and uniform edge weights for simplicity, the process can be expressed as:

$$\boldsymbol{A}_{\mathtt{U}}^\mathsf{T} \boldsymbol{H} = \boldsymbol{D}^{-1} \boldsymbol{A}_{\text{undir}}^\mathsf{T} \boldsymbol{H}, \qquad\qquad [\boldsymbol{A}_{\mathtt{U}}^\mathsf{T} \boldsymbol{H}]_{i,:} = \frac{1}{\deg(i)} \sum_{j:(j,i)\in\mathbb{M}_{\text{undir}}} \boldsymbol{H}_{j,:}. \tag{7}$$

The right-hand side describes *mean neighbourhood aggregation*, where messages (represented by $\boldsymbol{H}$) are averaged over a node's immediate neighbours. This mirrors the aggregation function used in GraphSAGE (Hamilton et al., 2017, Alg. 1). Similarly, $\boldsymbol{A}_{\mathtt{O}}^\mathsf{T}$ and $\boldsymbol{A}_{\mathtt{I}}^\mathsf{T}$ aggregate messages over outgoing and incoming neighbourhoods, respectively, as in the directed GraphSAGE extension proposed by Rossi et al. (2023). Full mathematical expressions for the directed case are provided in Appendix A.2.

**Automorphic equivalence.** Two nodes $i$ and $j$ are *automorphically equivalent* if there exists a permutation matrix $\boldsymbol{P}_\pi \in \{0,1\}^{n \times n}$ such that $\boldsymbol{A} = \boldsymbol{P}_\pi \boldsymbol{A} \boldsymbol{P}_\pi^\mathsf{T}$ and $\boldsymbol{P}_{\pi j,i} = 1$. Intuitively, this means that $i$ and $j$ are indistinguishable in terms of all graph-theoretic properties, such as degree, centrality, and clustering coefficients, and differ only in their labels. Automorphic equivalence is implied by structural equivalence, but not vice versa. Notably, automorphically equivalent nodes can be arbitrarily far apart in the graph, or even belong to disconnected components.

Message-passing ReachNEs preserves automorphic equivalence, provided that the node attributes are also invariant under the automorphism. Specifically, if $\boldsymbol{X} = \boldsymbol{P}_\pi \boldsymbol{X}$, which holds whenever $\boldsymbol{X}$ consists of graph-derived features, then the embeddings satisfy $\boldsymbol{P}_\pi \boldsymbol{Z} = \boldsymbol{Z}$. In this case, automorphically equivalent nodes receive identical message-passing embeddings. A proof and further discussion are provided in Appendix B.

## 3 Local sinks and directed neighbourhood multiplicity

In this section, we use the ReachNEs framework to study local sinks and neighbourhood multiplicity. We examine how local sinks obstruct information propagation and quantify their effects using entropy. We formalize the multiplicity of directed neighbourhoods within ReachNEs and discuss its implications for embedding model expressivity. Finally, we explore how expressivity can be improved through embedding concatenation.

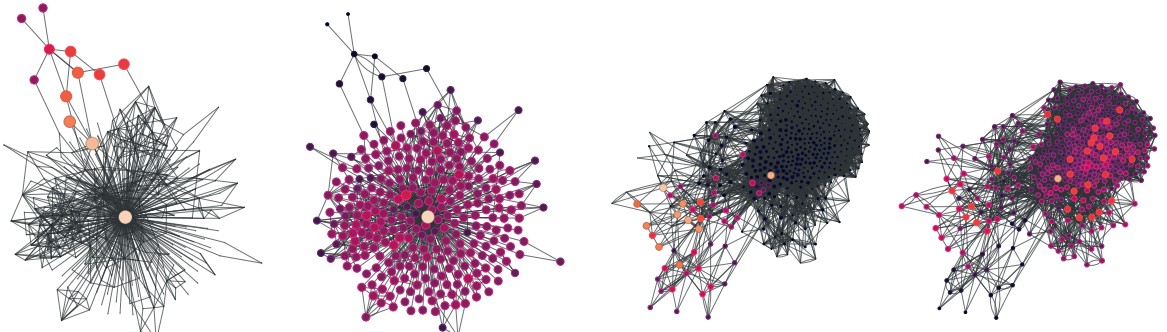

(a) Cora-ML. *Left:* Default (O), *Right:* Undirected (U)    (b) Fly Larva. *Left:* Default (O), *Right:* Undirected (U)

Figure 4: Visualization of transition probability accumulation in local sinks. Node size and colour indicate the magnitude of reachability, with black denoting zero. (a) shows the Cora-ML citation graph using default directed edges (O, left) and undirected edges (U, right). In the directed case, reachability is concentrated in a small set of sink nodes, whereas the undirected version yields more uniform coverage of the multi-step neighbourhood. (b) shows similar behaviour for the Fly Larva dataset: reachability under O accumulates disproportionately in sinks, while U promotes broader dispersal.

## 3.1    Local sink analysis

As seen in Section 2, the elements of the reachability matrix $\boldsymbol{R}$ represent random walk transition probabilities, with each column describing a smoothing over multi-step neighbourhoods for a given node. However, local sinks in digraphs can significantly disrupt this smoothing process.

As highlighted in Figure 1b, local sinks are nodes, or small sets of nodes, without directed paths to the rest of the graph. Random walks initiated at sink nodes remain trapped, preventing them from gathering additional neighbourhood information. Moreover, transition probabilities of walks starting at other nodes tend to accumulate in sinks, as random walkers can enter but cannot escape. This accumulation disproportionately amplifies the influence of sink nodes on the reachability filter, introducing bias into the embedding model.

Figure 4 illustrates these effects using ridiculograms of real-world graphs. In Figure 4a, the Cora-ML citation graph shows strong accumulation in sink nodes when using default edge directions, i.e., $\boldsymbol{R}(\boldsymbol{A_\text{O}}))$, while the undirected variant, i.e., $\boldsymbol{R}(\boldsymbol{A_\text{U}})$, achieves broader coverage over the multi-step neighbourhood. Figure 4b shows a similar pattern in the denser Fly Larva brain connectome (Winding et al., 2023).

The columns of the reachability matrix reflect how local sinks restrict information propagation. Consider a node $j$, whose random walk probabilities are given by the column $\boldsymbol{R}_{:,j}$. If $j$ is a sink node, or if its transition probabilities are largely absorbed by a sink, the reachability will be concentrated in a few nonzero values in $\boldsymbol{R}_{:,j}$; a property we refer to as low *dispersal*. Conversely, high dispersal indicates that probabilities are more evenly distributed across $j$'s multi-step neighbourhood.

This notion of dispersal is naturally captured by *Shannon entropy* (Cover & Thomas, 2005, Ch. 2.1). We define the reachability entropy of node $j$ as $H(j; \boldsymbol{R}) = -\sum_{i=1}^{n} R_{i,j} \log_2 R_{i,j}$. Entropy is always non-negative, reaching zero when all probability is concentrated in a single node. Conversely, it attains its maximum value, $\log_2 n$, when probabilities are uniformly distributed across all $n$ nodes. Thus, higher entropy corresponds to reduced bias toward sink nodes and improved dispersal.

In Figure 5, we plot the entropy for the Cora-ML and Fly Larva graphs. The y-axes represent entropy, $H(j; \boldsymbol{R})$, while the x-axes correspond to nodes $j$, sorted by entropy values. We use $\text{P}_\text{w} = \text{Pois}(\tau)$, with each coloured line representing a distinct $\tau$ value. The dashed line indicates the entropy of the asymptotic reachability matrix as $\tau \to \infty$ for the undirected graph (see Appendix C.1).

Transitioning from the directed to the undirected graph yields a notable increase in entropy for Cora-ML (cf. Figure 5a and Figure 5b), reflecting improved dispersal in the absence of sink nodes. The effect of the neighbourhood scale parameter $\tau$ also differs: in the undirected case, entropy grows steadily toward its asymptotic limit, while in the directed case, it eventually declines as probability mass accumulates in sink nodes.

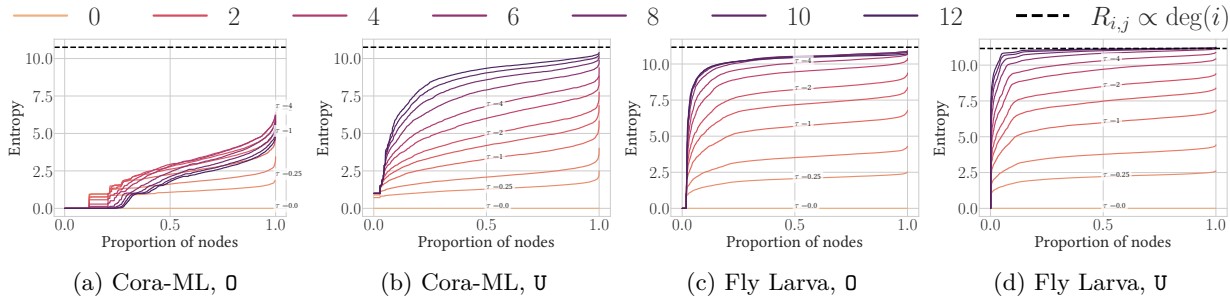

(a) Cora-ML, O      (b) Cora-ML, U      (c) Fly Larva, O      (d) Fly Larva, U

Figure 5: Reachability under $P_{\mathtt{w}} = \mathrm{Pois}(\tau)$, using $\boldsymbol{A}_{\mathtt{O}}$ and $\boldsymbol{A}_{\mathtt{U}}$. The x-axes correspond to nodes sorted by their entropy value. The colours represent various values of $\tau$. The black dashed line indicates the entropy of a reachability distribution proportional to the node degrees, which corresponds to the limiting and uninformative distribution as $\tau \to \infty$. See Appendix C.1 for further details.

For the Fly Larva graph, entropy differences between the directed and undirected cases are smaller, as expected given the graph's higher density. Still, entropy is consistently higher in the undirected case, with the largest gap appearing at the highest $\tau$, as reachability in the directed graph again concentrates in sinks.

It is important to note that high dispersal alone does not guarantee embedding quality. This becomes evident when considering that maximum entropy is achieved when $R_{i,j} = \frac{1}{n}$, meaning smoothing is performed uniformly across the graph. While this maximizes dispersal, it renders the embeddings uninformative. Instead, dispersal must be complemented by another key property, which we discuss in the next section: expressivity.

## 3.2 Embedding model expressivity and neighbourhood multiplicity

As discussed in Section 2.3, ReachNEs respects node equivalence under mild conditions: proximity ReachNEs yields identical embeddings for structurally equivalent nodes, and message-passing ReachNEs does so for automorphically equivalent ones. However, this alone is not sufficient for producing high-quality embeddings. We also require model *expressivity*, which we define as the ability to *distinguish nonequivalent nodes*.

A fully expressive model would assign identical embeddings to equivalent nodes, and distinct embeddings to nonequivalent ones. Achieving this for automorphic equivalence is particularly challenging; in fact, any model that perfectly encodes automorphic equivalence would effectively solve the graph isomorphism problem (Grohe & Schweitzer, 2020). The expressivity of message-passing models is therefore an active area of research (Xu et al., 2019; Morris et al., 2019) (see Section 5.4 for further discussion).

Structural equivalence is simpler to handle. Since structurally equivalent nodes have identical rows and columns in the adjacency matrix $\boldsymbol{A}$, one can construct a fully expressive, albeit impractical, embedding matrix by concatenating $\boldsymbol{A}$ and its transpose: $\boldsymbol{Z} = \begin{bmatrix} \boldsymbol{A} & \boldsymbol{A}^{\mathsf{T}} \end{bmatrix}$. This model captures structural distinctions perfectly but ignores multi-step relationships and produces a prohibitively large embedding space. Consequently, expressivity remains a key concern for proximity ReachNEs, as both the random-walk smoothing in $\boldsymbol{R}$ and the information loss from SVD reduction can limit the model's ability to distinguish between nonequivalent nodes.

Model expressivity is especially important for digraphs due to the existence of *neighbourhood multiplicity*. As illustrated in Figure 1, the neighbourhood of a node in a digraph is not uniquely defined. At each step of a random walk, the walker can follow either incoming or outgoing edges, leading to $2^K$ possible neighbourhoods for a walk of length $K$. Embedding models that fail to capture the distinctions among these directed neighbourhoods inherently suffer from reduced expressivity.

We formalize this idea using a truncated $K$-step reachability matrix. Let the *edge orientation specifier* be a string $\boldsymbol{\sigma}$ of length $K$, where each character $\sigma_l \in \{\mathtt{U}, \mathtt{O}, \mathtt{I}\}$ indicates whether the $l$-th step uses undirected, outgoing, or incoming edges, respectively. Given a specifier $\boldsymbol{\sigma}$, the corresponding reachability matrix is:

$$\boldsymbol{R}^{(K)}(\boldsymbol{A}; P_{\mathtt{w}}, \boldsymbol{\sigma}) = P_{\mathtt{w}}(0)\boldsymbol{I}_n + \sum_{k=1}^{K} P_{\mathtt{w}}(k)\overset{\frown}{\prod_{l=1}^{k}}\boldsymbol{A}_{\sigma_l}, \qquad \overset{\frown}{\prod_{l=1}^{k}}\boldsymbol{A}_{\sigma_l} = \boldsymbol{A}_{\sigma_k}\dots\boldsymbol{A}_{\sigma_2}\boldsymbol{A}_{\sigma_1}. \tag{8}$$

Like the reachability definition in Equation 2, the matrix $\boldsymbol{R}^{(K)}(\boldsymbol{A}; \mathrm{P_w}, \boldsymbol{\sigma})$ is a column-stochastic transition matrix, provided that $\mathrm{P_w}$ is properly normalized over $K$ steps. It captures transition probabilities for multi-directional random walks defined by the edge orientation specifier $\boldsymbol{\sigma}$. For instance, $\boldsymbol{\sigma} = \mathtt{OIO}$ corresponds to walks that follow outgoing, then incoming, then outgoing edges. Our definition also allows walk steps over undirected edges ($\mathtt{U}$), which will be relevant for defining DirSwitch in Section 4.

Equation 8 defines how to compute multi-directional reachability matrices, but it remains to determine how best to incorporate them into node embeddings. A widely used and effective strategy for this purpose is *embedding concatenation* (Jin et al., 2019; Corso et al., 2020). This approach involves extracting multiple sets of embeddings that capture different structural aspects of the graph, e.g., $\boldsymbol{Z}^{(1)} \in \mathbb{R}^{n \times p_1}$ and $\boldsymbol{Z}^{(2)} \in \mathbb{R}^{n \times p_2}$, and combining them into a joint embedding space, $\boldsymbol{Z} = \begin{bmatrix} \boldsymbol{Z}^{(1)} & \boldsymbol{Z}^{(2)} \end{bmatrix} \in \mathbb{R}^{n \times (p_1+p_2)}$. Concatenation is particularly advantageous in unsupervised settings, as it ensures that all constituent embeddings contribute to the final representation.

Concatenation is particularly popular for producing *multi-scale* embeddings (Rozemberczki et al., 2021). The goal of multi-scale embeddings is to represent the inherent hierarchical structure of real-world graphs (Newman, 2018, Ch. 14.7.2) by concatenating embeddings that capture different neighbourhood scales. The ReachNEs framework enables this by incorporating multiple walk length distributions, $\mathrm{P_w}$. For example, using $\mathrm{P_w}^{(1)} = \mathrm{Geom}(\tau = 1)$ and $\mathrm{P_w}^{(2)} = \mathrm{Pois}(\tau = 3)$ results in two reachability matrices, $\boldsymbol{R}(\mathrm{P_w}^{(1)})$ and $\boldsymbol{R}(\mathrm{P_w}^{(2)})$, which focus on immediate and 3-step neighbourhoods, respectively.

The same principle extends to directed neighbourhoods. By concatenating embeddings generated from different edge direction specifiers $\boldsymbol{\sigma}$, we obtain *multi-directional* embeddings. This approach can be further combined with multi-scale embeddings to jointly encode both aspects of graph structure.

However, a key limitation of embedding concatenation is that the embedding dimensionality, $p$, grows with each additional concatenation. This increase can be rapid, given the exponential number of distinct directed neighbourhoods and their possible combinations with multi-scale embeddings. Excessively high dimensionality leads to increased memory and computational costs, as well as potential issues related to the curse of dimensionality. Therefore, it is beneficial to pair concatenation with dimensionality reduction techniques, such as PCA (Murphy, 2012, Ch. 12.2).

## 4 DirSwitch: Switching reachability filters

The previous section highlights a dual challenge in digraph embedding modelling. Local sinks obstruct information propagation, leading to embeddings biased toward sink nodes. This issue can be easily mitigated by treating the digraph as undirected, effectively ignoring edge directions. However, this approach reduces model expressivity by discarding information about the multiplicity of directed neighbourhoods.

To address these intertwined issues, we propose *switching random walks*. The key idea is to generate multi-directional embeddings by following directed edges for the first $r$ steps of a random walk, thereby capturing local directed neighbourhood structure. After $r$ steps, the graph is treated as undirected, allowing the walk to escape sinks and achieve broader dispersal.

We name this approach DirSwitch. To formally define the DirSwitch model, we introduce a shorthand notation for the edge direction specifier $\boldsymbol{\sigma}$. For a $K$-step random walk, we define $\boldsymbol{\sigma}$ as a string of length $r + 1$, where $r < K$. The final character, $\sigma_{r+1}$, determines the direction for the last $K - r$ steps of the walk.

Rewriting Equation 8 using this notation and taking the limit as $K \to \infty$, we obtain:

$$\boldsymbol{R}(\boldsymbol{A}; \mathrm{P_w}, \boldsymbol{\sigma}) = \mathrm{P_w}(0)\boldsymbol{I}_n + \sum_{k=1}^{r} \mathrm{P_w}(k)\overset{\curvearrowleft k}{\prod_{l=1}} \boldsymbol{A}_{\sigma_l} + \sum_{k=r+1}^{\infty} \mathrm{P_w}(k)\boldsymbol{A}_{\sigma_{r+1}}^{k-r}\overset{\curvearrowleft r}{\prod_{l=1}} \boldsymbol{A}_{\sigma_l}. \tag{9}$$

In this formulation, a DirSwitch model is characterized by the first $r$ letters in $\boldsymbol{\sigma}$ being either $\mathtt{O}$ or $\mathtt{I}$, representing directed walk steps, while the final character, $\sigma_{r+1}$, is set to $\mathtt{U}$, ensuring undirected propagation for the remaining steps.

The switch to undirected edges addresses the issue of local sinks, but it remains necessary to incorporate representations of different directed neighbourhoods. To achieve this, we use the multi-directional concatenation approach described in Section 3.2. For a given value of $r$, we define the DirSwitch-$r$ embedding model as the concatenation of all $2^r$ possible configurations of a $\boldsymbol{\sigma}$ sequence of length $r + 1$ ending with $\sigma_{r+1} = \texttt{U}$. For example, DirSwitch-1 concatenates embeddings using $\boldsymbol{\sigma} = \texttt{OU}$ and $\boldsymbol{\sigma} = \texttt{IU}$, while DirSwitch-2 uses $\boldsymbol{\sigma} \in \{\texttt{OOU}, \texttt{OIU}, \texttt{IOU}, \texttt{IIU}\}$, and so on for any DirSwitch-$r$ model.

As a baseline in our experiments, we define MultiDir-$r$ using the same notation as above. Specifically, MultiDir-1 uses $\boldsymbol{\sigma} = \texttt{O}$ and $\boldsymbol{\sigma} = \texttt{I}$, while MultiDir-2 includes all two-step combinations $\boldsymbol{\sigma} \in \{\texttt{O}, \texttt{OI}, \texttt{IO}, \texttt{I}\}$, and so on, covering all $2^r$ possible $r$-step direction sequences. Unlike DirSwitch, however, MultiDir does not transition to undirected edges after the specified sequence. Instead, the walk continues in the last specified direction. For example, with $K$ total steps and $\boldsymbol{\sigma} = \texttt{OI}$, the walk takes one step in the $\texttt{O}$ direction, one in the $\texttt{I}$ direction, and then continues the remaining $K - 2$ steps in the $\texttt{I}$ direction.

As noted in Section 3.2, multi-directional concatenation can be combined with multi-scale concatenation to improve embedding quality. However, this combination exhibits diminishing returns: using $r$ steps of directionality with $s$ walk length distributions for a fixed embedding dimension $p$ means that each $(\boldsymbol{\sigma}, \mathrm{P_w})$-pair contributes only $\frac{p}{2^r s}$ dimensions. We discuss these diminishing returns further in Appendix A.6, where we also present a practical guide to DirSwitch-$r$ hyperparameter selection based on our experiments and experience. Additionally, increasing $r$ and $s$ raises computational costs, which we analyse in Appendix A.5.

## 5 Related Work

### 5.1 Relation to node embedding and graph learning frameworks

The ReachNEs framework is built around the reachability matrix, interpreted as random walk transition probabilities, and two embedding reduction strategies: matrix decomposition and message passing. Similar ideas appear throughout the graph learning literature. Below, we highlight key connections and distinctions.

**Sampled random walks.** The use of random walks for proximity-based embeddings was popularized by Perozzi et al. (2014) and Grover & Leskovec (2016), who combined sampled walks with negative sampling—originally introduced for word embeddings (Mikolov et al., 2013). As shown by Qiu et al. (2018), these methods asymptotically approximate a matrix factorization of $\log(\boldsymbol{R}) + b$, where log is applied elementwise, $b$ is a constant, and $\boldsymbol{R}$ is the random walk transition probability matrix, as in ReachNEs.

This suggests that sampled walk models and proximity ReachNEs converge to similar embeddings in the limit of infinite samples. However, in practice, sampling is expensive, and only a small fraction of possible walks and non-edges are used. As a result, sampled random walk models and proximity ReachNEs typically yield different embeddings in finite regimes.

Additionally, sampled walk methods usually rely on a single walk length distribution; commonly uniform, though Zhou et al. (2017) use the geometric distribution. In contrast, ReachNEs supports multiple walk length distributions, enabling multi-scale embeddings.

**Proximity matrix reduction.** Extending the idea of proximity embeddings via matrix decomposition, Zhu et al. (2021a) introduced the PhUSION framework. Like ReachNEs, PhUSION generates embeddings by applying reduction functions to a node proximity matrix, and can encode either structural or automorphic equivalence depending on the chosen reduction. We compare this aspect of the two frameworks in Section 5.2.

In other aspects, PhUSION differs from ReachNEs in two key ways. First, it is restricted to undirected graphs. Second, it interprets the proximity matrix through the lens of matrix functions (Higham, 2008), treating the scalars $\mathrm{P_w}(k)$ in Equation 2 as Taylor coefficients rather than probabilities. While this perspective is mathematically more general, it limits analysability. For example, our analysis of reachability entropy in Section 3.1 depends on interpreting $\boldsymbol{R}$ as a transition probability matrix.

**Graph convolutional filters.** Like matrix functions, *graph convolutional filters* generalize the reachability matrix (Defferrard et al., 2016; Isufi et al., 2024). They are defined and applied similarly to message-passing ReachNEs, also assuming the presence of initial node features $\boldsymbol{X}$. However, the series coefficients, $\mathrm{P_w}(k)$ in Equation 2, can take arbitrary real values, making random walk interpretations generally inapplicable.

Instead, convolutional filters are viewed as extensions of classical convolutions to graphs and are typically analysed using spectral methods based on eigenfunctions (Bronstein et al., 2017). Since asymmetric matrices lack a standard eigendecomposition, spectral analysis becomes difficult on digraphs, and most research in this area focuses on undirected graphs (Zhang et al., 2021b). In contrast, ReachNEs defines $\boldsymbol{R}$ purely through random walk probabilities, making the framework naturally applicable to both directed and undirected graphs.

**Message-passing graph neural networks.** The dominant paradigm in node embedding learning is message-passing graph neural networks (GNNs) (Gilmer et al., 2017). GNNs are typically nonlinear and parameterized, requiring training via non-convex optimization. While this complicates their analysis compared to ReachNEs, their overall algorithmic structure is similar. In fact, only minor modifications to Equation 6 are needed to recover the GraphSAGE encoder (Hamilton et al., 2017). We exploit this connection in Section 6.5 to show how our results extend to the self-supervised GNN setting.

Interestingly, the binomial walk length distribution in ReachNEs is related to the self-loops added in convolutional GNNs (Kipf & Welling, 2017; Wu et al., 2019). We explore this connection further in Appendix C.5.

**Graph kernels.** While many embedding methods implicitly rely on walk length distributions, few works explicitly analyse or compare their effects as we do. In contrast, the graph kernel literature, particularly *random walk kernels*, has explored related formulations, including both geometric and Poisson walk length distributions (Vishwanathan et al., 2010). However, since graph kernels are designed for graph-level comparison rather than node-level tasks, it remains an open question which insights transfer to ReachNEs.

## 5.2 Structural and automorphic equivalence

Recent work has highlighted the distinction between embedding models that capture structural versus automorphic node equivalence (Jin et al., 2021; Zhu et al., 2021a; Yan et al., 2024). Models targeting structural equivalence are often called *positional embeddings*, while those targeting automorphic equivalence are referred to as *structural embeddings*. The terminology is admittedly confusing, as the term *structural* is used in two different senses.

ReachNEs draws inspiration from the PhUSION framework (Zhu et al., 2021a), which also produces both positional and structural embeddings by applying different reduction methods to a node proximity matrix (analogous to our reachability matrix). For positional embeddings, both ReachNEs and PhUSION use SVD reduction. However, PhUSION assumes undirected graphs and retains only the left singular vectors $\boldsymbol{U}$, while ReachNEs concatenates both $\boldsymbol{U}$ and $\boldsymbol{V}$ to handle directed graphs.

For structural embeddings, PhUSION applies the permutation-invariant empirical characteristic function to each column of $\boldsymbol{R}$, following the approach introduced by Donnat et al. (2018). In contrast, ReachNEs uses message-passing reduction, aligning more directly with the message-passing GNN literature.

Finally, while some argue that positional and structural embeddings represent fundamentally different objectives (Rossi et al., 2020), Srinivasan & Ribeiro (2020) established a statistical equivalence between them. However, the practical consequences of this result remain an open question.

## 5.3 Digraph models

Research on message-passing embedding models, particularly GNNs, has primarily focused on undirected graphs (Rossi et al., 2023). This is especially true in unsupervised and self-supervised learning, where recent state-of-the-art models remain constrained to undirected graphs (Hassani & Khasahmadi, 2020; Thakoor et al., 2022; Zhang et al., 2021a; Hou et al., 2023).

For (semi-)supervised learning, Rossi et al. (2023) proposed digraph extensions for common GNN embedding encoders. Unlike DirSwitch, which uses concatenation to preserve the contributions of different directed neighbourhoods in distinct embedding components, Rossi et al. (2023) employs parameterized mixing of these components. This approach requires the model to learn how to extract and preserve relevant information from multi-directional neighbourhoods. However, this filtering process is more challenging in unsupervised settings due to the weaker learning signal.

In contrast, research on unsupervised proximity embeddings for digraphs has progressed further than its message-passing counterpart. The models HOPE (Ou et al., 2016), APP (Zhou et al., 2017), and NERD (Khosla et al., 2020) follow a similar approach to ReachNEs and DirSwitch, leveraging random walks and matrix factorization of proximity matrices. However, none of these models explicitly address local sinks and neighbourhood multiplicity, which are the primary focus of our work.

Additionally, there are several notable algorithmic differences. Both HOPE and APP use the geometric walk length distribution, while NERD employs the uniform distribution. In contrast, ReachNEs treats the choice of distribution as a parameter. NERD also introduces *alternating walks*, where edge direction specifiers switch between outgoing and incoming edges (e.g., $\sigma = \text{OIOI}...$ or $\sigma = \text{IOIO}...$). While this pattern may help random walks escape some sinks, restricting the model to only these sequences introduces bias by excluding other directed neighbourhood definitions.

Furthermore, HOPE utilizes partial and generalized SVD (Hochstenbach, 2009) for matrix factorization, whereas ReachNEs employs the more scalable single-pass SVD (Yu et al., 2017). Both APP and NERD rely on Monte Carlo sampling of random walks and implicit matrix factorization via gradient descent and negative sampling, while ReachNEs directly computes the reachability matrix using matrix multiplication.

Finally, DGGAN (Zhu et al., 2021b) and BLADE (Virinchi & Saladi, 2023) are unsupervised digraph embedding methods that incorporate neural networks. While neither explicitly addresses the issue of local sinks, BLADE follows an embedding mixing strategy similar to Rossi et al. (2023), where a GNN is responsible for filtering information from distinct directed neighbourhoods.

## 5.4 Expressivity of message-passing models

The expressivity of message-passing models is a central topic in graph representation learning due to its connection with the graph isomorphism problem (Grohe & Schweitzer, 2020). Li & Leskovec (2022, Def. 5.5) offers a formal definition, characterizing model expressivity as the ability to produce distinct representations for non-isomorphic graphs. While their formulation is mathematically rigorous, our definition in Section 3.2 captures the same core idea in a more intuitive manner, with a focus on node equivalence rather than full graph isomorphism.

The foundational works of Xu et al. (2019) and Morris et al. (2019) relate the expressivity of message-passing graph neural networks (MP-GNNs) to the Weisfeiler-Leman (WL) colour refinement algorithm (Grohe et al., 2021), an iterative method used to test graph isomorphism. Rossi et al. (2023) extended these results to digraphs by incorporating both in- and out-neighbourhoods in the message-passing process, in a manner analogous to our MultiDir and DirSwitch approaches.

These studies demonstrate that the expressive power of MP-GNNs is upper-bounded by the WL algorithm. This also applies to message-passing ReachNEs, which can be viewed as a linear MP-GNN. Moreover, they show that MP-GNNs can match the expressivity of the WL test if their message aggregation functions are sufficiently flexible—specifically, if they can form injective mappings from multisets to vectors, mimicking the WL hashing mechanism. These assumptions, however, do not hold for ReachNEs, which is linear and parameterized solely through scalar-valued walk length distributions.

The use of flexible, non-linear aggregation functions is well-suited to the supervised, end-to-end learning context studied by Xu et al. (2019), Morris et al. (2019), and Rossi et al. (2023), where parameters can be optimized to task-specific objectives. In contrast, the unsupervised setting considered in this work demands greater interpretability and analytical tractability, especially for downstream tasks such as anomaly detection.

Table 2: Graph statistics for the datasets. The columns report the number of nodes ($n$) and edges ($m$), the median out- and in-degrees ($|\deg_0|$, $|\deg_I|$), the number of weakly and strongly connected components (#CC and #SCC), the global clustering coefficient ($C_{\mathcal{G}}$), and the average path length ($\langle l_{\text{path}} \rangle$) computed on the undirected version of the graph. Additionally, when available, we list the number of node attributes ($d$), the number of node label classes ($|\mathbb{Y}|$), and the homophily coefficient of the undirected graph ($h_U$). The final column specifies the graph type. We use the shorthand notation with K=$10^3$ and M=$10^6$ for large values.

| Dataset | $n$ | $m$ | $|\deg_0|$ | $|\deg_I|$ | # CC | # SCC | $C_{\mathcal{G}}$ | $\langle l_{\text{path}} \rangle$ | $d$ | $|\mathbb{Y}|$ | $h_U$ | Type |
|---|---|---|---|---|---|---|---|---|---|---|---|---|
| Arxiv[1] | 170K | 1.2M | 4 | 1 | 1 | 141K | 0.017 | 5.7 | 128 | 40 | 0.64 | Citation |
| Arxiv-Year[2] | 170K | 1.2M | 4 | 1 | 1 | 141K | 0.017 | 5.7 | 128 | 5 | 0.29 | Citation |
| Citeseer[3] | 4.2K | 5.4K | 1 | 0 | 515 | 4209 | 0.084 | 7.4 | 602 | 6 | 0.96 | Citation |
| CoCite[4] | 44K | 20K | 2 | 2 | 652 | 44K | 0.081 | 5.5 | – | 15 | 0.42 | Citation |
| Cora[3] | 20K | 65K | 2 | 1 | 364 | 16K | 0.14 | 6.2 | 8710 | 70 | 0.59 | Citation |
| Cora-ML[3] | 3K | 8.4K | 2 | 1 | 61 | 2603 | 0.12 | 5.3 | 2879 | 7 | 0.82 | Citation |
| Cora (Subelj)[4] | 23K | 92K | 3 | 1 | 1 | 18K | 0.12 | 5.8 | – | 70 | 0.56 | Citation |
| Enron[5] | 7.9K | 142K | 3 | 7 | 58 | 861 | 0.16 | 3.1 | – | – | – | Email |
| EU-Email[6] | 1K | 25K | 14 | 17 | 20 | 203 | 0.29 | 2.6 | – | 42 | 0.47 | Email |
| Fly Larva[7] | 3K | 116K | 33 | 33 | 5 | 136 | 0.30 | 2.7 | – | 93 | 0.14 | Connectome |
| Polblogs[8] | 1.5K | 19K | 4 | 2 | 268 | 688 | 0.25 | 2.7 | – | 2 | 0.91 | Hyperlink |
| Pokec[2] | 1.6M | 31M | 8 | 8 | 1 | 326K | 0.058 | 4.7 | 65 | 3 | 0.43 | Dating |
| Pubmed[4] | 20K | 44K | 0 | 1 | 1 | 20K | 0.054 | 6.4 | – | 3 | 0.79 | Citation |
| Roman Empire[2] | 23K | 33K | 1 | 1 | 1 | 23K | 0.28 | 2.4K | 300 | 18 | 0.05 | Text |
| Snap Patents[2] | 2.9M | 14M | 3 | 3 | 181K | 2.9M | 0.066 | 6.8 | 269 | 5 | 0.22 | Citation |
| WikiVote[9] | 7K | 103K | 2 | 0 | 24 | 5816 | 0.14 | 3.2 | – | – | – | Voting |

[1] Hu et al. (2020a)  [2] Lim et al. (2021)  [3] Bojchevski & Günnemann (2018)  [4] Khosla et al. (2020)  [5] Klimt & Yang (2004)
[6] Yin et al. (2017)  [7] Winding et al. (2023)  [8] Adamic & Glance (2005)  [9] Leskovec et al. (2010)

In this regard, our work aligns more closely with Jin et al. (2019) and Corso et al. (2020), who emphasize embedding concatenation as a strategy for increasing expressivity. While their approaches combine embeddings from different aggregation operators (e.g., mean, max, sum), we instead concatenate embeddings derived from multiple edge directions and walk length distributions. This preserves both interpretability and theoretical clarity, as each component of the embedding retains a well-defined meaning grounded in random-walk semantics.

# 6 Experiments

Our experiments focus on evaluating DirSwitch and are divided into three parts. First, we validate that DirSwitch mitigates low dispersal caused by local sinks in digraphs (Section 6.1) while preserving the ability to represent directed neighbourhoods (Section 6.2). Second, in Section 6.3, we demonstrate that these improvements lead to higher-quality embeddings by evaluating DirSwitch in combination with ReachNEs on 14 node classification benchmark datasets.

Third, we evaluate DirSwitch's practical effectiveness by comparing it to recent digraph embedding models on node classification benchmarks. Specifically, we first assess DirSwitch with ReachNEs proximity embeddings against state-of-the-art unsupervised digraph proximity embedding approaches (Section 6.4). Then, in Section 6.5, we demonstrate DirSwitch's flexibility by applying it beyond the ReachNEs framework. We extend self-supervised GNNs to digraphs using ReachNEs and compare this approach to the method proposed by Rossi et al. (2023), which generalizes GNNs to digraphs in the semi-supervised setting.

Our experiments use the graph learning datasets summarized in Table 2, which span a diverse range of graph types and properties, including variations in density, connectivity, and node attributes. Unless otherwise specified, we compute truncated reachability using $K = 12$ steps. All experiments were conducted in a Google Cloud environment with an Nvidia L4 24GB GPU, 32 vCPUs @ 2.20GHz, and 128GB of memory.

## 6.1 Improving dispersal

To verify that DirSwitch mitigates local sinks and improves dispersal, we measure the reachability entropy $H(j; \boldsymbol{R}) = -\sum_{i=1}^{n} R_{i,j} \log_2 R_{i,j}$ (see Section 3.1) for various reachability matrices $\boldsymbol{R}(P_w, \boldsymbol{\sigma})$. Specifically, we

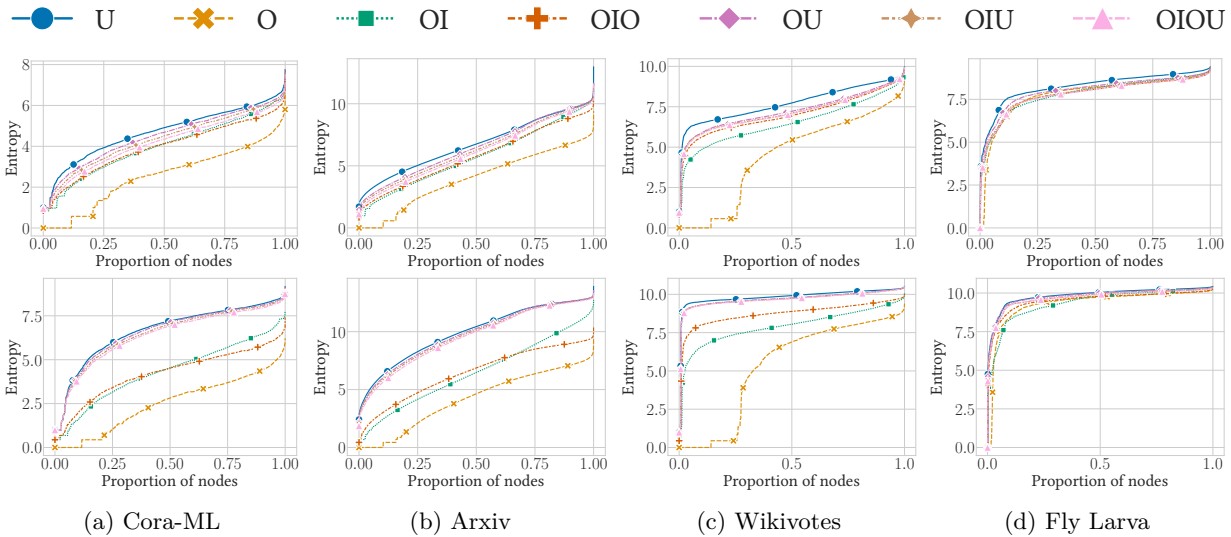

(a) Cora-ML      (b) Arxiv      (c) Wikivotes      (d) Fly Larva

Figure 6: Neighbourhood dispersal evaluation for four graphs, measured via reachability entropy, $-\sum_{i=1}^{n} R_{i,j} \log_2 R_{i,j}$, computed for each node and sorted. Each curve corresponds to a different edge direction specifier $\boldsymbol{\sigma}$, with the top row showing results for $P_{\mathtt{w}}(k; \tau) = \mathrm{Pois}(\tau = 2)$ (local dispersal) and the bottom row for $\mathcal{U}(\tau = 5)$ (long-range dispersal). DirSwitch variants (e.g., $\mathtt{OU}$, $\mathtt{OIU}$, $\mathtt{OIOU}$) demonstrate high dispersal, comparable to $\mathtt{U}$, while purely directed specifiers ($\mathtt{O}$, $\mathtt{OI}$, $\mathtt{OIO}$) exhibit lower entropy due to sink effects.

compare DirSwitch edge direction specifiers $\boldsymbol{\sigma} \in \{\mathtt{OU}, \mathtt{OIU}, \mathtt{OIOU}\}$ against MultiDir specifiers $\boldsymbol{\sigma} \in \{\mathtt{O}, \mathtt{OI}, \mathtt{OIO}\}$ and purely undirected edges, $\boldsymbol{\sigma} = \mathtt{U}$. We use two walk length distributions: $\mathrm{Pois}(\tau = 2)$, which highlights short-range differences, and $\mathcal{U}(\tau = 5)$, which captures long-range behaviour.

Figure 6 presents results for four representative graphs, with additional datasets provided in Appendix E. The top row shows results for $\mathrm{Pois}(\tau = 2)$, while the bottom row corresponds to $\mathcal{U}(\tau = 5)$.

The entropy associated with DirSwitch specifiers closely approaches that of undirected edges, which consistently achieve the highest entropy due to the absence of sinks. This outcome is both expected and desirable, highlighting how the switch to undirected edges mitigates the dispersal-limiting effects of local sinks.

These effects are most pronounced for long-range walks using $\mathcal{U}(\tau = 5)$, shown in the bottom row. In Figures 6a, 6b, and 6c, there is a significant entropy gap between the DirSwitch curves and the MultiDir curves ($\mathtt{O}, \mathtt{OI}, \mathtt{OIO}$). The Cora-ML, Arxiv, and Wikivote graphs are sparsely connected (see Table 2), leading to increased reachability concentration in local sinks and lower entropy. In contrast, the densely connected Fly Larva graph in Figure 6d exhibits a less pronounced but still noticeable entropy difference.

For local smoothing with $\mathrm{Pois}(\tau = 2)$ (top row), the entropy increase for DirSwitch over $\mathtt{OI}$ and $\mathtt{OIO}$ is visible but small. This is because alternating $\mathtt{O}$- and $\mathtt{I}$-steps allows short-range walkers to partially escape sinks. Moreover, we observe a discernible gap between $\mathtt{U}$ and the DirSwitch directions. This gap reflects the reduced dispersal caused by using edge directions in the initial steps of DirSwitch random walks. As walk length increases, both effects vanish as the undirected regime of DirSwitch dominates, as seen in the bottom row.

## 6.2 Directed neighbourhood expressivity

We use node embedding graph alignment (Heimann et al., 2018) to verify that DirSwitch can represent the diversity of directed neighborhoods. Graph alignment is a generalized version of the graph isomorphism problem, where nodes in two graphs are matched based on structural similarity (Skitsas et al., 2023).

Following the embedding benchmark protocols of Heimann et al. (2018) and Jin et al. (2021), we construct a second graph $\mathcal{G}_2 = (\mathbb{N}, \mathbb{M}_2)$ from a given graph $\mathcal{G}_1 = (\mathbb{N}, \mathbb{M}_1)$ by removing 15% of the edges from $\mathbb{M}_1$ and randomly permuting the node indices. Node embeddings are then computed for both graphs, and each node

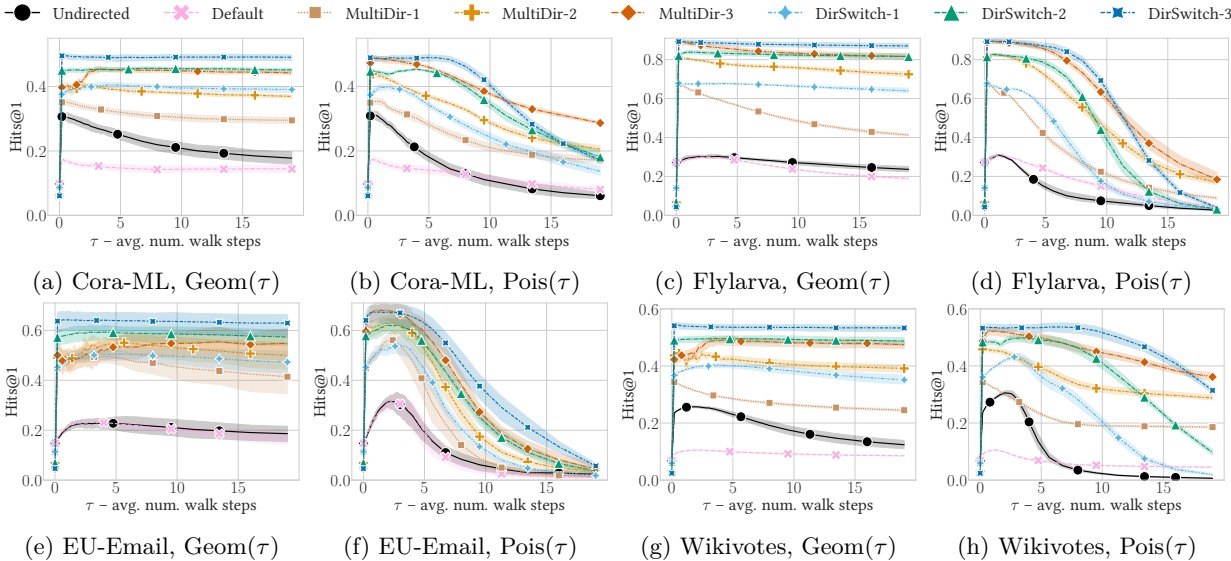

Figure 7: Evaluation of edge direction expressivity for four graphs using the geometric, $\text{Geom}(\tau)$, and Poisson, $\text{Pois}(\tau)$, walk length distributions. The y-axes represent graph alignment accuracy under 15% edge removal, while the x-axes correspond to $\tau$, the average walk length. The curve colours and styles denote different sets of edge direction specifiers, $\boldsymbol{\sigma}$.

in $\mathcal{G}_2$ is matched to a node in $\mathcal{G}_1$ based on the shortest Euclidean distance between embeddings. This process is repeated five times with different random seeds.

The graph alignment task benefits significantly from embedding expressivity, particularly the ability to distinguish distinct directed neighbourhoods, as this reduces the risk of erroneous matches. However, for matching to be feasible, embeddings must also capture local graph structure that generalizes across separate graph components. To achieve this, we use ReachNEs message-passing embeddings combined with node attributes derived from graph structure, including out- and in-degrees and the four digraph local clustering coefficients—*out*, *in*, *cycle*, and *middleman*—as defined by Fagiolo (2007).

We compare the multi-directional DirSwitch-$r$ against MultiDir-$r$ for $r \leq 3$, as well as undirected ($\boldsymbol{\sigma} = \mathtt{U}$) and default directed ($\boldsymbol{\sigma} = \mathtt{O}$) edges. To account for the influence of walk length distribution on embedding expressivity, we evaluate both $\mathrm{P_w} = \text{Geom}(\tau)$ and $\mathrm{P_w} = \text{Pois}(\tau)$ for $\tau \in (0, 20]$, using a reachability truncation of $K = 30$ steps.

Figure 7 presents results for four datasets, with $\tau$ values along the x-axes and graph alignment accuracy on the y-axes. Additional results for other datasets and walk length distributions are provided in Appendix F. DirSwitch-$r$ and MultiDir-$r$ achieve comparable accuracies for each $r$, with accuracy increasing as $r$ grows. This similarity is expected, as both methods use concatenation to represent $2^r$ directed neighbourhoods.

A closer comparison reveals that DirSwitch-$r$ consistently outperforms MultiDir-$r$, provided $\tau$ is not too large. This improvement stems from DirSwitch mitigating sink bias. When reachability is concentrated in small node sets, embeddings become overly sensitive to edge removal, as this can then radically change the reachability distribution. This sensitivity reduces the alignment accuracy. By alleviating this bias, DirSwitch provides more robust representations.

For the same reason, the undirected edges perform better than the default directed edges. However, overall, both these single-neighbourhood baselines achieve the lowest overall accuracies due to their inability to capture directed neighbourhood multiplicity.

Another notable observation is the impact of the walk length distribution $\mathrm{P_w}(\tau)$. In all cases, accuracy collapses as $\tau \to 0$, corresponding to the use of node attributes without smoothing. Additionally, for the Poisson distribution, accuracy decreases as $\tau$ increases, whereas for the geometric distribution, it plateaus. This behaviour aligns with the expected asymptotic properties of both distributions.

Table 3: Abbreviations used when reporting results for multi-scale reachability embeddings. Visualizations are available in Figure 11 in the Appendix.

| Abbreviation | Geom | Pois | Geom-$\mathcal{U}$ | Binom-3 | Geom-4 |
|---|---|---|---|---|---|
| Distributions, $P_w$ | $\text{Geom}(k; \tau = 1)$ | $\text{Pois}(k; \tau = 2)$ | $\text{Geom}(k; \tau = 1)$, $\mathcal{U}(k-1; \tau = 2)$ | $\text{Binom}(k; \tau = 1)$, $\text{Binom}(k-2; \tau = 2)$, $\text{Binom}(k-5; \tau = 3)$ | $\text{Geom}(k; \tau = 1)$, $\text{Geom}(k-1; \tau = 2)$, $\text{Geom}(k-2; \tau = 3)$, $\text{Geom}(k-3; \tau = 4)$ |

In short, as $\tau$ increases, the mode of $\text{Pois}(\tau)$ shifts continuously, emphasizing long-range walks over local smoothing. This results in less distinguishable embeddings and lower alignment accuracy. In contrast, the geometric distribution maintains local neighbourhood smoothing, preventing significant accuracy drops at higher $\tau$ values. See Appendix C for further details.

The difference between Pois and Geom is also evident in the comparison between DirSwitch-$r$ and MultiDir-$r$. With the geometric distribution, DirSwitch consistently achieves higher alignment accuracy across varying $\tau$ values, as discussed above. However, for the Poisson distribution, DirSwitch's accuracy declines more rapidly than that of MultiDir as $\tau$ increases in some datasets. For instance, this effect is observed for DirSwitch-3 and MultiDir-3 in Figures 7b and 7d.

This behaviour stems from the shifting mode of Pois. In the limit $\tau \to \infty$, DirSwitch embeddings converge to identical representations within a weakly connected component. In contrast, MultiDir embeddings retain finer granularity, as equivalence groups in the directed case remain more distinct, see our discussion at the end of Appendix C.1. This explains why MultiDir can surpass DirSwitch in alignment accuracy at large $\tau$ values.

### 6.3 DirSwitch embedding quality evaluation

In this section, we demonstrate that DirSwitch's increased dispersal and multi-directionality lead to higher-quality embeddings. To assess this, we use node classification as a representative downstream task, comparing ReachNEs embeddings generated with DirSwitch-$r$, MultiDir-$r$, undirected edges (U), and default directed edges (O).

**Setup** Table 2 lists the node classification benchmark datasets. For datasets with node attributes, we evaluate message-passing ReachNEs, while for those without attributes, we use proximity ReachNEs.

Node embeddings are computed in a fully unsupervised manner, after which a logistic regression classifier is trained on frozen embeddings. We employ a 3x repeated 5-fold cross-validation strategy to obtain mean performance and standard deviations.

To assess the impact of both multi-directional and multi-scale concatenation, we include both single- and multi-scale embeddings in our analysis. Table 3 details the walk length distributions used for single- and multi-scale embeddings. We consider two single-scale distributions and three multi-scale distributions, incorporating two, three, and four scales, respectively. The embedding dimensions are set to $p = 1024$ and $p = 512$.

**Results Overview** Table 4 reports classification accuracies for message-passing ReachNEs using datasets with node attributes, while Table 6 presents results for proximity ReachNEs for datasets without attributes, both using $p = 1024$. Across all datasets and embedding types, DirSwitch-$r$ consistently achieves the highest accuracies or performs within one to two standard deviations of the best results. In the tables, we highlight the best results for each scale in bold blue and results within one standard deviation in light blue. The consistent top performance of DirSwitch across datasets and multi-scale choices underscores its superior embedding quality compared to non-switching approaches. Similar trends are observed for $p = 512$, with further analysis provided in Appendix G.

A notable trend is that $\sigma = $ O (default) yields some of the lowest accuracies across all datasets and scales, as highlighted in orange. This suggests that embedding quality improves both by using undirected edges (U) and by concatenating $\sigma = $ I embeddings, as done in MultiDir-1. These results further support the claim that mitigating local sinks and capturing different directed neighbourhoods enhances embedding quality.

Next, we conduct a deeper analysis of these results. To facilitate this, we use the relative accuracy improvements in Tables 5 and 7. These values represent the highest accuracy in each row of Tables 4 and 6, relative to the accuracy of the default. For default edge directions themselves, absolute accuracy values are reported.

Table 4: Node classification accuracy for message-passing ReachNEs with $p = 1024$ embedding dimensions. Columns correspond to different datasets and multi-scale walk length distributions, while rows represent various edge direction specifiers. Each entry reports the mean accuracy and standard deviation. Bold blue highlights the highest accuracy in each column, with light blue indicating results within one standard deviation of the best. Similarly, bold orange denotes the lowest accuracy, and light orange represents values within one standard deviation of the worst.

| Edge directions | ARXIV | | | | | ARXIV YEAR | | | | |
| --- | --- | --- | --- | --- | --- | --- | --- | --- | --- | --- |
| | GEOM | POIS | GEOM-$\mathcal{U}$ | BINOM-3 | GEOM-4 | GEOM | POIS | GEOM-$\mathcal{U}$ | BINOM-3 | GEOM-4 |
| DEFAULT | 55.0±0.2 | 42.4±0.3 | 55.5±0.3 | 54.2±0.2 | 55.4±0.3 | 36.5±0.3 | 35.4±0.5 | 36.7±0.2 | 37.8±0.2 | 38.2±0.2 |
| UNDIRECTED | 61.6±0.2 | 64.9±0.3 | 67.0±0.2 | 69.8±0.2 | 69.1±0.2 | 36.4±0.2 | 36.5±0.2 | 37.1±0.2 | 37.6±0.3 | 38.3±0.3 |
| MULTIDIR-1 | 59.8±0.3 | 59.4±0.3 | 60.2±0.3 | 60.5±0.2 | 60.3±0.2 | 37.0±0.3 | 35.3±0.8 | 38.1±0.2 | 39.3±0.2 | 39.7±0.3 |
| MULTIDIR-2 | 61.2±0.3 | 65.3±0.3 | 65.2±0.3 | 64.9±0.3 | 65.3±0.3 | 38.5±0.3 | 37.9±0.4 | 40.9±0.2 | 42.0±0.2 | 41.7±0.2 |
| MULTIDIR-3 | 61.5±0.3 | 65.5±0.3 | 65.9±0.3 | 65.9±0.3 | 65.7±0.3 | 40.0±0.3 | 40.6±0.3 | 42.3±0.2 | 43.2±0.3 | 42.4±0.3 |
| DIRSWITCH-1 | 61.5±0.3 | 65.9±0.2 | 67.3±0.3 | 69.7±0.2 | 68.4±0.3 | 38.2±0.3 | 37.8±0.2 | 39.9±0.3 | 40.2±0.3 | 41.5±0.2 |
| DIRSWITCH-2 | 61.5±0.3 | 65.8±0.3 | 66.9±0.3 | 69.6±0.3 | 68.4±0.2 | 39.6±0.2 | 39.5±0.3 | 41.4±0.3 | 41.9±0.3 | 42.4±0.2 |
| DIRSWITCH-3 | 61.5±0.3 | 65.8±0.3 | 66.7±0.3 | 69.0±0.3 | 67.8±0.3 | 40.0±0.3 | 40.9±0.4 | 42.4±0.3 | 43.3±0.3 | 43.3±0.3 |

| Edge directions | CORA-ML | | | | | CITESEER | | | | | CORA | | | | |
| --- | --- | --- | --- | --- | --- | --- | --- | --- | --- | --- | --- | --- | --- | --- | --- |
| | GEOM | POIS | GEOM-$\mathcal{U}$ | BINOM-3 | GEOM-4 | GEOM | POIS | GEOM-$\mathcal{U}$ | BINOM-3 | GEOM-4 | GEOM | POIS | GEOM-$\mathcal{U}$ | BINOM-3 | GEOM-4 |
| DEFAULT | 31.5±1.6 | 35.8±2.4 | 46.1±2.1 | 58.4±2.2 | 63.2±2.3 | 67.5±2.0 | 56.5±1.9 | 60.8±2.7 | 70.0±2.0 | 72.9±2.1 | 52.4±0.6 | 51.2±0.7 | 57.6±0.7 | 59.3±0.7 | 60.2±0.8 |
| UNDIRECTED | 31.2±1.9 | 41.0±2.2 | 54.5±1.7 | 72.1±2.2 | 76.1±1.8 | 76.2±1.8 | 83.7±1.4 | 72.6±1.8 | 88.0±1.7 | 89.0±1.2 | 59.4±0.6 | 64.9±0.6 | 67.5±0.7 | 69.3±0.7 | 69.6±0.7 |
| MULTIDIR-1 | 55.1±2.1 | 53.2±2.0 | 66.9±2.1 | 72.2±2.1 | 75.4±2.3 | 66.1±2.0 | 60.8±1.8 | 73.8±1.3 | 78.6±1.7 | 81.0±1.6 | 64.3±0.6 | 63.3±0.6 | 65.8±0.8 | 65.3±1.0 | 64.8±1.0 |
| MULTIDIR-2 | 71.3±1.6 | 71.1±2.3 | 78.4±1.8 | 79.8±2.1 | 82.3±1.4 | 79.4±1.2 | 78.4±1.2 | 84.3±1.4 | 85.0±1.2 | 86.0±1.4 | 67.1±0.7 | 68.4±0.7 | 67.8±0.9 | 66.5±0.9 | 66.0±0.9 |
| MULTIDIR-3 | 78.9±1.9 | 80.0±1.0 | 84.0±1.4 | 83.9±1.4 | 84.5±1.5 | 84.7±1.7 | 87.2±1.2 | 87.2±1.2 | 87.3±1.3 | 87.5±1.2 | 66.7±0.9 | 68.5±0.9 | 66.8±0.9 | 65.3±0.8 | 64.4±0.9 |
| DIRSWITCH-1 | 56.9±2.9 | 57.5±2.4 | 72.0±2.0 | 79.4±1.6 | 82.3±2.0 | 73.9±1.2 | 76.5±1.9 | 85.6±1.5 | 91.4±0.8 | 91.3±1.5 | 65.6±0.7 | 67.5±0.8 | 69.5±0.8 | 70.1±0.9 | 69.7±0.8 |
| DIRSWITCH-2 | 71.6±1.8 | 72.4±2.4 | 80.7±1.6 | 84.3±1.6 | 84.4±1.9 | 80.4±1.3 | 85.1±1.4 | 88.6±1.0 | 90.9±1.0 | 90.1±1.0 | 67.5±0.6 | 68.9±0.7 | 68.9±0.9 | 69.0±0.7 | 68.0±0.7 |
| DIRSWITCH-3 | 79.1±2.0 | 80.6±1.6 | 84.0±1.5 | 85.6±1.6 | 85.7±1.4 | 84.7±1.3 | 88.0±0.9 | 87.7±1.3 | 89.6±0.8 | 88.5±1.2 | 66.4±0.8 | 68.7±0.8 | 67.0±0.8 | 67.1±0.6 | 65.6±0.9 |

| Edge directions | ROMAN EMPIRE | | | | | POKEC | | | | | SNAP PATENTS | | | | |
| --- | --- | --- | --- | --- | --- | --- | --- | --- | --- | --- | --- | --- | --- | --- | --- |
| | GEOM | POIS | GEOM-$\mathcal{U}$ | BINOM-3 | GEOM-4 | GEOM | POIS | GEOM-$\mathcal{U}$ | BINOM-3 | GEOM-4 | GEOM | POIS | GEOM-$\mathcal{U}$ | BINOM-3 | GEOM-4 |
| DEFAULT | 69.3±0.9 | 32.5±0.4 | 68.5±0.6 | 51.2±0.6 | 67.2±0.8 | 60.1±0.2 | 59.0±0.8 | 61.7±0.1 | 60.7±0.1 | 65.4±0.1 | 35.7±0.1 | 41.7±0.1 | 39.2±0.1 | 42.2±0.2 | 43.1±0.3 |
| UNDIRECTED | 67.2±0.8 | 46.9±0.7 | 70.9±0.7 | 65.9±0.8 | 70.0±0.4 | 60.3±0.1 | 59.6±0.5 | 62.0±0.1 | 62.3±0.1 | 71.7±0.1 | 30.8±0.1 | 29.9±0.1 | 31.8±0.1 | 33.2±0.1 | 34.7±0.4 |
| MULTIDIR-1 | 71.8±0.8 | 42.0±0.8 | 66.7±0.8 | 56.6±0.7 | 69.9±0.9 | 61.3±0.1 | 60.1±0.1 | 62.2±0.1 | 61.7±0.1 | 67.3±0.1 | 41.0±0.1 | 43.9±0.4 | 45.7±0.1 | 47.8±0.2 | 46.9±0.2 |
| MULTIDIR-2 | 71.9±0.6 | 60.4±0.7 | 71.1±0.5 | 65.2±0.6 | 75.5±0.6 | 61.5±0.1 | 61.1±0.1 | 63.0±0.1 | | 69.1±0.1 | 45.2±0.1 | 46.5±0.2 | 46.8±0.2 | 49.8±0.2 | 48.1±0.2 |
| MULTIDIR-3 | 73.1±0.6 | 68.0±0.6 | 73.7±0.7 | 69.2±0.7 | 76.0±0.5 | 61.8±0.1 | 63.5±0.0 | 64.8±0.1 | 64.8±0.1 | 70.7±0.1 | 45.9±0.1 | 49.1±0.1 | 47.4±0.1 | 49.8±0.7 | 46.5±0.1 |
| DIRSWITCH-1 | 73.0±0.6 | 58.4±0.8 | 72.4±0.9 | 67.5±0.5 | 74.6±0.6 | 61.2±0.1 | 60.3±0.3 | 62.8±0.1 | 62.8±0.1 | 70.0±0.1 | 40.8±0.1 | 40.8±0.1 | 42.7±0.1 | 43.8±0.1 | 44.8±0.3 |
| DIRSWITCH-2 | 72.0±0.5 | 62.8±0.7 | 74.2±0.5 | 70.8±0.6 | 76.4±0.7 | 61.4±0.1 | 60.9±0.1 | 63.5±0.1 | 64.1±0.1 | 71.5±0.1 | 46.6±0.1 | 48.1±0.2 | 46.8±0.1 | 47.2±0.2 | 47.6±0.4 |
| DIRSWITCH-3 | 72.8±0.6 | 67.9±0.6 | 75.1±0.6 | 72.5±0.6 | 76.4±0.6 | 61.6±0.1 | 62.7±0.1 | 64.7±0.1 | 64.4±0.1 | 70.3±0.1 | 46.3±0.1 | 50.8±0.3 | 49.2±0.1 | 49.4±0.3 | 48.8±0.3 |

Table 5: Relative improvements in node classification accuracy for message-passing ReachNEs with $p = 1024$. The values reflect the maximum accuracy for per row and dataset in Table 4. The top row displays absolute accuracies for the default edge directions, with standard deviations expressed as percentages. The subsequent rows present the relative improvements compared to the top row. The table is structured with the four homophilic datasets on the left and the four heterophilic datasets on the right.

| Edge directions | CORA-ML | CITESEER | CORA | ARXIV | ROMAN EMPIRE | ARXIV YEAR | POKEC | SNAP PATENTS |
| --- | --- | --- | --- | --- | --- | --- | --- | --- |
| DEFAULT | 63.2±3.4% | 72.9±3.0% | 60.2±1.1% | 55.5±0.5% | 69.3±0.9% | 38.2±0.8% | 65.4±0.4% | 43.1±0.4% |
| UNDIRECTED | +20.5% | +22.0% | +15.5% | +25.8% | +2.2% | +0.2% | +9.5% | -19.3% |
| MULTIDIR-1 | +19.4% | +11.1% | +9.2% | +9.0% | +3.5% | +3.9% | +2.8% | +11.0% |
| MULTIDIR-2 | +30.3% | +18.0% | +13.5% | +17.7% | +8.9% | +9.9% | +5.6% | +15.6% |
| MULTIDIR-3 | +33.7% | +20.0% | +13.6% | +18.9% | +9.6% | +12.9% | +8.0% | +15.6% |
| DIRSWITCH-1 | +30.3% | +25.4% | +16.3% | +25.7% | +7.5% | +8.5% | +6.9% | +4.1% |
| DIRSWITCH-2 | +33.6% | +24.7% | +14.5% | +25.4% | +10.2% | +10.9% | +9.3% | +11.7% |
| DIRSWITCH-3 | +35.7% | +22.9% | +14.0% | +24.5% | +10.2% | +13.4% | +7.5% | +17.9% |

**Homophilic vs heterophilic datasets for message-passing embeddings**  The node classification datasets in Table 2 exhibit varying levels of *homophily* (Zhu et al., 2020), i.e., the tendency of nodes with the same class label to connect. The homophily level is quantified in the $h_{\mathrm{U}}$ column, representing the average fraction of a node's neighbours that share its label (Rossi et al., 2023, Eq. 1).

Table 6: Node classification accuracy for proximity ReachNEs with $p = 1024$ embedding dimensions. Columns correspond to different datasets and multi-scale walk length distributions, while rows represent various edge direction specifiers. The values denote average accuracies with standard deviations. Bold blue highlights the best results in each column, with light blue indicating results within one standard deviation. Similarly, bold orange marks the worst results, with light orange showing values within one standard deviation of the lowest performance.

| EDGE DIRECTIONS | FLY LARVA | | | | | EU-EMAIL | | | | | POLBLOGS | | | | |
|---|---|---|---|---|---|---|---|---|---|---|---|---|---|---|---|
| | GEOM | POIS | GEOM-$\mathcal{U}$ | BINOM-3 | GEOM-4 | GEOM | POIS | GEOM-$\mathcal{U}$ | BINOM-3 | GEOM-4 | GEOM | POIS | GEOM-$\mathcal{U}$ | BINOM-3 | GEOM-4 |
| DEFAULT | 42.2±2.0 | 45.2±2.0 | 49.0±1.9 | 49.5±1.7 | 50.1±2.0 | 30.3±3.3 | 36.6±2.9 | 54.0±3.0 | 65.0±2.8 | 66.0±3.2 | 61.8±2.7 | 70.2±2.4 | 73.0±2.7 | 79.1±1.9 | 82.0±2.1 |
| UNDIRECTED | 45.0±1.7 | 50.9±2.0 | 53.7±2.1 | 54.6±1.7 | 55.4±1.7 | 29.8±3.0 | 35.4±3.7 | 57.2±3.5 | 70.6±2.9 | 72.5±3.1 | 64.4±3.3 | 76.8±2.0 | 79.1±2.4 | 84.3±2.2 | 85.7±1.9 |
| MULTIDIR-1 | 47.3±1.6 | 49.1±1.8 | 50.8±1.8 | 51.9±1.9 | 52.0±1.9 | 50.2±3.1 | 59.7±3.7 | 68.1±2.9 | 72.2±2.8 | 73.5±3.0 | 73.7±2.6 | 74.7±2.6 | 83.6±1.9 | 85.2±1.7 | 86.7±2.1 |
| MULTIDIR-2 | 50.3±2.3 | 50.7±2.1 | 52.4±2.2 | 54.2±2.0 | 53.6±1.7 | 66.8±3.2 | 69.5±3.3 | 74.9±3.0 | 75.2±2.8 | 74.7±2.6 | 83.4±2.3 | 84.4±2.1 | 88.2±2.0 | 88.7±1.6 | 89.2±1.5 |
| MULTIDIR-3 | 51.8±2.6 | 52.6±2.2 | 53.8±2.1 | 53.6±1.8 | 53.0±1.5 | 73.5±3.4 | 74.2±2.4 | 74.9±2.3 | 73.1±3.3 | 70.4±2.9 | 88.0±1.7 | 88.0±2.0 | 89.2±1.4 | 89.7±1.5 | 89.8±1.6 |
| DIRSWITCH-1 | 48.8±1.7 | 51.6±1.6 | 53.8±2.4 | 55.8±1.7 | 55.8±2.4 | 50.9±3.2 | 59.8±3.7 | 70.5±3.1 | 74.8±2.8 | 75.6±2.4 | 75.6±2.6 | 75.3±2.2 | 85.5±2.4 | 86.6±2.1 | 86.2±2.2 |
| DIRSWITCH-2 | 50.3±1.7 | 51.9±2.1 | 54.1±1.5 | 56.7±2.0 | 56.2±1.9 | 66.8±2.7 | 69.8±3.5 | 74.6±2.9 | 75.7±2.5 | 74.8±2.5 | 83.8±2.2 | 85.2±1.7 | 88.4±2.1 | 87.1±2.1 | 89.7±1.6 |
| DIRSWITCH-3 | 52.4±1.7 | 52.8±1.6 | 55.6±1.8 | 55.6±1.8 | 55.6±1.8 | 73.3±2.6 | 73.8±2.7 | 75.3±2.6 | 72.2±2.9 | 70.6±2.8 | 87.9±1.8 | 88.4±1.6 | 89.4±1.6 | 87.1±2.0 | 89.8±1.5 |

| EDGE DIRECTIONS | COCITE | | | | | PUBMED | | | | | CORA (SUBELJ) | | | | |
|---|---|---|---|---|---|---|---|---|---|---|---|---|---|---|---|
| | GEOM | POIS | GEOM-$\mathcal{U}$ | BINOM-3 | GEOM-4 | GEOM | POIS | GEOM-$\mathcal{U}$ | BINOM-3 | GEOM-4 | GEOM | POIS | GEOM-$\mathcal{U}$ | BINOM-3 | GEOM-4 |
| DEFAULT | 38.6±0.4 | 39.0±0.4 | 39.9±0.5 | 39.8±0.4 | 40.2±0.6 | 72.4±0.7 | 72.6±0.7 | 73.3±0.8 | 73.1±0.7 | 72.6±0.6 | 58.2±0.7 | 58.2±1.0 | 59.0±0.8 | 59.2±0.9 | 59.3±0.8 |
| UNDIRECTED | 45.3±0.5 | 45.8±0.6 | 46.6±0.4 | 47.9±0.5 | 47.6±0.5 | 81.9±0.5 | 82.2±0.6 | 82.2±0.5 | 82.4±0.6 | 82.2±0.6 | 65.7±0.7 | 66.0±0.8 | 66.7±0.7 | 67.1±0.8 | 67.0±0.8 |
| MULTIDIR-1 | 39.8±0.5 | 40.0±0.5 | 40.8±0.5 | 41.3±0.6 | 41.3±0.5 | 73.9±0.7 | 73.5±0.8 | 72.9±0.6 | 72.7±0.8 | 71.5±0.7 | 58.7±0.8 | 58.5±0.8 | 58.7±0.7 | 58.3±0.8 | 58.3±0.7 |
| MULTIDIR-2 | 42.5±0.6 | 43.2±0.6 | 43.9±0.6 | 45.0±0.6 | 45.0±0.5 | 77.8±0.6 | 77.4±0.7 | 77.0±0.6 | 76.3±0.6 | 75.7±0.9 | 61.3±0.6 | 61.5±0.7 | 61.7±0.6 | 61.6±0.7 | 61.8±0.8 |
| MULTIDIR-3 | 44.2±0.5 | 44.7±0.5 | 44.6±0.4 | 46.4±0.4 | 46.1±0.5 | 79.3±0.6 | 79.0±0.6 | 78.7±0.6 | 78.0±0.6 | 77.7±0.6 | 61.6±0.8 | 62.1±0.7 | 62.4±0.7 | 62.2±0.7 | 62.5±0.7 |
| DIRSWITCH-1 | 45.3±0.4 | 45.9±0.5 | 46.9±0.5 | 48.4±0.4 | 47.9±0.5 | 81.6±0.7 | 81.9±0.6 | 82.0±0.7 | 82.3±0.6 | 81.9±0.7 | 65.2±0.7 | 65.7±0.6 | 66.0±0.8 | 66.4±0.7 | 66.4±0.7 |
| DIRSWITCH-2 | 44.9±0.5 | 45.9±0.5 | 46.8±0.5 | 48.3±0.5 | 48.0±0.4 | 81.8±0.7 | 81.8±0.7 | 81.8±0.6 | 81.9±0.7 | 81.1±0.6 | 64.0±0.9 | 64.7±0.8 | 65.1±0.8 | 65.6±0.9 | 65.4±0.7 |
| DIRSWITCH-3 | 43.8±0.5 | 45.1±0.5 | 45.8±0.5 | 48.3±0.5 | 47.5±0.5 | 81.3±0.6 | 81.4±0.6 | 80.9±0.7 | 81.2±0.7 | 80.4±0.6 | 61.8±0.8 | 62.9±0.8 | 63.1±0.9 | 64.4±0.8 | 63.9±0.9 |

We observe distinct differences between homophilic and heterophilic datasets in the message-passing results in Table 5. The three homophilic datasets—Cora-ML, Citeseer, and Cora (leftmost in the table)—experience a 15–26% accuracy improvement when using undirected edges. In contrast, the heterophilic Roman Empire and Arxiv Year datasets show no significant improvement over default directions, while Snap Patents exhibits a 19% accuracy drop. Pokec is a partial exception, where undirected edges yield benefits with Geom-4, a case analysed in Appendix G.

On the other hand, heterophilic datasets such as Arxiv Year and Snap Patents see substantial accuracy gains as $r$ increases in DirSwitch-$r$ and MultiDir-$r$. For instance, DirSwitch-3 improves accuracy by +17.9% on Snap Patents, compared to +4.1% for DirSwitch-1. Meanwhile, in homophilic datasets, DirSwitch's accuracy remains stable across different $r$ values.

Together, these results indicate that multi-directional representations are more important for heterophilic datasets, whereas using undirected edges is beneficial in homophilic cases. These findings align with prior observations in supervised graph neural networks for digraphs (Rossi et al., 2023).

This contrasting behaviour can be explained through the lens of reachability. In homophilic datasets, nodes with the same label tend to cluster locally. Removing edge directions improves dispersal and promotes local smoothing, resulting in more similar embeddings for same-label nodes and improved classification performance. This effect is illustrated in Figure 2.

In heterophilic graphs, however, capturing local connectivity patterns that repeat across different regions is essential for distinguishing node roles. Here, multi-directional approaches play a crucial role, as discarding edge direction information can introduce spurious similarities between nodes with distinct, direction-dependent neighbourhoods.

**Sparse vs dense graphs for proximity embeddings** We next analyse classification accuracies for proximity embeddings in Table 7, observing distinct trends based on graph density. Denser graphs (Fly Larva, EU-Email, and Polblogs) benefit more from multi-directional embeddings, whereas sparser graphs (CoCite, Pubmed, and Cora (subelj)) achieve higher accuracies when dispersal is increased using undirected edges. For instance, the sparse CoCite graph improves by 19% with undirected edges, with a similar 20% improvement

Table 7: Relative improvements in node classification accuracy for proximity ReachNEs with $p = 1024$. The values reflect the maximum accuracy for per row and dataset in Table 6. The top row displays absolute accuracies for the default edge directions, with standard deviations expressed as percentages. The subsequent rows present the relative improvements compared to the top row. The table is structured with the three denser graphs on the left and the three sparser graphs on the right.

| EDGE DIRECTIONS | FLY LARVA | EU-EMAIL | POLBLOGS | CoCITE | PUBMED | CORA (SUBELJ) |
|---|---|---|---|---|---|---|
| DEFAULT | 50.1±3.9% | 66.0±4.6% | 82.0±2.9% | 40.2±1.2% | 73.3±1.0% | 59.3±1.4% |
| UNDIRECTED | +10.6% | +9.8% | +4.5% | +19.0% | **+12.4%** | **+13.2%** |
| MULTIDIR-1 | +3.9% | +11.3% | +5.8% | +2.7% | +0.8% | -1.0% |
| MULTIDIR-2 | +8.2% | +14.0% | +8.8% | +11.9% | +6.1% | +4.3% |
| MULTIDIR-3 | +7.5% | +13.4% | **+9.5%** | +15.5% | +8.2% | +5.4% |
| DIRSWITCH-1 | +11.5% | +14.5% | +5.6% | **+20.2%** | +12.3% | +12.1% |
| DIRSWITCH-2 | **+13.2%** | **+14.7%** | +9.4% | **+20.2%** | +11.7% | +10.7% |
| DIRSWITCH-3 | +11.1% | +14.0% | **+9.5%** | **+20.2%** | +11.0% | +8.6% |

for each DirSwitch model. Conversely, the dense Polblogs graph gains only 4.5% from undirected edges but sees a 9.5% boost with DirSwitch-3.

This difference between dense and sparse graphs can be explained by reachability entropy. Even with purely directed edges, entropy remains high for the dense Fly Larva graph (see Figure 6d), indicating that local sinks do not significantly hinder dispersal. A similar pattern is observed for EU-Email and Polblogs in Appendix E, explaining why undirected edges provide less benefit in these graphs.

Instead, increasing embedding distinguishability by representing directed neighbourhoods becomes more important. Comparing default edges and DirSwitch-1 in Table 6, we observe significant accuracy gains when transitioning from single-scale to multi-scale walk length distributions. For example, Polblogs achieves only 61% accuracy with single-scale Geom but improves to 82% with multi-scale Geom-4. Conversely, sparser datasets show only minor improvements with multi-scale embeddings, with dispersal remaining the primary factor influencing embedding quality.

**Further analysis** Additional discussion of these results is provided in Appendix G, where we examine the accuracy improvements from multi-scale embeddings, the diminishing returns of repeated concatenation and perform further analysis of the results for the Pokec dataset. We also present evaluation results using unsupervised community detection as an alternative to node classification.

## 6.4 Comparison to state-of-the-art digraph proximity embeddings

We compare DirSwitch with proximity ReachNEs embeddings against state-of-the-art unsupervised digraph proximity embedding approaches discussed in Section 5.3. Hyperparameters for each model are optimized via cross-validation grid search, with the search grids and best values reported in Appendix D.2.

Table 8: Node classification accuracies for proximity node embedding models. Bold indicate the top accuracy, and results within one standard deviation, for each dataset. Average and standard deviations are calculated over 3x repeated 5-fold cross validations, and 5 different random seeds.

| MODEL | FLYLARVA | EU-EMAIL | POLBLOGS | CoCITE | PUBMED | CORA (SUBELJ) |
|---|---|---|---|---|---|---|
| HOPE[1] | 35.8 ± 1.9 | 49.3 ± 3.6 | 83.2 ± 3.1 | 30.4 ± 1.2 | 65.3 ± 0.9 | 34.0 ± 0.8 |
| APP[2] | 42.8 ± 2.3 | **76.3 ± 2.8** | **88.9 ± 1.7** | 46.7 ± 0.5 | 81.2 ± 0.6 | 65.6 ± 0.7 |
| NERD[3] | 40.3 ± 2.2 | 71.3 ± 3.2 | **89.4 ± 1.7** | 28.1 ± 0.5 | 76.5 ± 0.7 | 45.7 ± 0.6 |
| DGGAN[4] | 16.5 ± 1.4 | 11.7 ± 2.5 | 54.6 ± 2.9 | 17.7 ± 0.4 | 41.5 ± 0.9 | 6.2 ± 0.3 |
| BLADE[5] | 49.2 ± 2.2 | 66.7 ± 3.0 | **89.3 ± 1.9** | 24.2 ± 0.5 | 60.2 ± 1.2 | 27.7 ± 1.0 |
| DIRSWITCH | **56.8 ± 2.2** | **75.6 ± 2.7** | **90.0 ± 1.7** | **48.3 ± 0.5** | **82.2 ± 0.6** | **66.6 ± 0.7** |

[1] OU ET AL. (2016)  [2] ZHOU ET AL. (2017)  [3] KHOSLA ET AL. (2020)  [4] ZHU ET AL. (2021B)  [5] VIRINCHI & SALADI (2023)

Table 9: Node classification accuracy for self-supervised GraphSAGE embeddings using training via Graph-MAEv2 and CCA-SSG losses. Bold indicate the top accuracy for each dataset. Average and standard deviations are calculated over 3x repeated 5-fold cross validations and 5 different seeds.

(a) GraphMAEv2

| EDGE DIRECTIONS | ROMAN EMPIRE | ARXIV YEAR | POKEC | SNAP PATENTS | CORA-ML | CITESEER | CORA | ARXIV |
|---|---|---|---|---|---|---|---|---|
| DEFAULT | $66.09_{\pm 0.79}$ | $44.49_{\pm 0.24}$ | $64.62_{\pm 0.20}$ | $45.68_{\pm 0.26}$ | $79.61_{\pm 1.76}$ | $80.13_{\pm 1.28}$ | $59.48_{\pm 0.61}$ | $59.81_{\pm 0.25}$ |
| UNDIRECTED | $68.17_{\pm 0.74}$ | $40.08_{\pm 0.26}$ | $65.99_{\pm 0.52}$ | $33.82_{\pm 0.07}$ | $\mathbf{86.03_{\pm 1.42}}$ | $\mathbf{90.71_{\pm 0.92}}$ | $\mathbf{68.54_{\pm 0.74}}$ | $\mathbf{70.49_{\pm 0.23}}$ |
| ROSSI ET AL. (2023) | $67.05_{\pm 1.85}$ | $46.38_{\pm 0.34}$ | $57.93_{\pm 0.55}$ | $\mathbf{61.34_{\pm 0.28}}$ | $81.88_{\pm 3.94}$ | $87.53_{\pm 2.30}$ | $34.35_{\pm 4.42}$ | $65.53_{\pm 0.52}$ |
| DIRSWITCH-1 | $71.76_{\pm 0.76}$ | $41.28_{\pm 0.28}$ | $67.16_{\pm 0.34}$ | $39.49_{\pm 0.30}$ | $85.79_{\pm 1.48}$ | $89.03_{\pm 1.19}$ | $68.01_{\pm 0.83}$ | $69.95_{\pm 0.21}$ |
| DIRSWITCH-3 | $\mathbf{71.90_{\pm 0.78}}$ | $\mathbf{46.65_{\pm 0.31}}$ | $\mathbf{67.57_{\pm 0.25}}$ | $57.99_{\pm 0.21}$ | $85.24_{\pm 1.37}$ | $86.67_{\pm 1.06}$ | $66.31_{\pm 0.80}$ | $68.40_{\pm 0.23}$ |

(b) CCA-SSG

| EDGE DIRECTIONS | ROMAN EMPIRE | ARXIV YEAR | POKEC | SNAP PATENTS | CORA-ML | CITESEER | CORA | ARXIV |
|---|---|---|---|---|---|---|---|---|
| DEFAULT | $53.73_{\pm 0.94}$ | $45.01_{\pm 0.26}$ | $58.07_{\pm 0.27}$ | $46.37_{\pm 0.25}$ | $61.70_{\pm 3.82}$ | $69.72_{\pm 1.77}$ | $39.70_{\pm 0.94}$ | $36.73_{\pm 1.46}$ |
| UNDIRECTED | $53.85_{\pm 2.45}$ | $35.82_{\pm 0.28}$ | $60.38_{\pm 0.74}$ | $38.60_{\pm 0.13}$ | $\mathbf{82.64_{\pm 1.59}}$ | $80.71_{\pm 1.36}$ | $52.35_{\pm 1.30}$ | $45.93_{\pm 1.18}$ |
| ROSSI ET AL. (2023) | $53.74_{\pm 0.75}$ | $46.83_{\pm 0.44}$ | $60.54_{\pm 0.50}$ | $\mathbf{65.55_{\pm 0.28}}$ | $60.44_{\pm 2.94}$ | $64.64_{\pm 2.41}$ | $31.28_{\pm 2.21}$ | $34.92_{\pm 0.54}$ |
| DIRSWITCH-1 | $56.77_{\pm 2.36}$ | $45.19_{\pm 0.35}$ | $66.23_{\pm 0.70}$ | $49.84_{\pm 0.60}$ | $82.00_{\pm 1.76}$ | $77.35_{\pm 2.53}$ | $52.70_{\pm 1.28}$ | $61.63_{\pm 0.38}$ |
| DIRSWITCH-3 | $\mathbf{66.39_{\pm 1.24}}$ | $\mathbf{52.64_{\pm 0.30}}$ | $\mathbf{67.25_{\pm 0.21}}$ | $64.08_{\pm 0.19}$ | $82.27_{\pm 1.42}$ | $\mathbf{85.26_{\pm 1.32}}$ | $\mathbf{61.07_{\pm 0.80}}$ | $\mathbf{63.58_{\pm 0.30}}$ |

Following hyperparameter tuning, test accuracy is evaluated using the best hyperparameter setting on new 3x repeated 5-fold CV splits, with five different random seeds per model. We use the official implementations for all baselines except BLADE, which we reimplement as its source code is not publicly available.

Table 8 presents the average test accuracies and standard deviations for each proximity embedding dataset. DirSwitch consistently achieves the highest average accuracy across datasets or performs within one standard deviation of the best model (EU-Email). Notably, DirSwitch outperforms competing approaches by a significant margin on Fly Larva, achieving 57% accuracy compared to 49% for BLADE. These results highlight DirSwitch's ability to leverage both multi-directional and multi-scale embeddings while avoiding local sinks, distinguishing it from prior proximity embedding models.

## 6.5 Applying DirSwitch to self-supervised message-passing graph neural networks

DirSwitch is not limited to the ReachNEs framework and can also be applied to message-passing graph neural networks (GNNs). The most natural integration is with the GraphSAGE model (Hamilton et al., 2017), given its structural similarity to ReachNEs. By defining $\boldsymbol{H}^{(0)} = \boldsymbol{Z}^{(0)} = \boldsymbol{X}$, we can represent the DirSwitch-GraphSAGE model using the following iterative formulation, analogous to Equation 6:

$$\text{For } k \in \{1, 2, \dots\}, \quad \begin{cases} \boldsymbol{H}^{(k)} &= \boldsymbol{A}_*^{\mathsf{T}} \boldsymbol{H}^{(k-1)}, \\ \boldsymbol{Z}^{(k)} &= \sigma\left(\boldsymbol{Z}^{(k-1)} \boldsymbol{W}_1^{(k-1)} + \boldsymbol{H}^{(k)} \boldsymbol{W}_2^{(k-1)}\right), \end{cases} \quad (10)$$

The key differences are the weight matrices $\boldsymbol{W}_1$ and $\boldsymbol{W}_2$, which replace the walk length probability coefficients, and the non-linear activation function $\sigma$.

For training the weight matrices, we employ self-supervised learning, the current state-of-the-art approach for unsupervised GNN training. These loss functions typically combine reconstruction and contrastive learning objectives. In our experiments, we use two recent and efficient methods: GraphMAEv2 (Hou et al., 2023) and CCA-SSG (Zhang et al., 2021a). We adopt the default hyperparameters for both loss functions and optimizers, as recommended by the original implementations. We train DirSwitch-1 and DirSwitch-3 models but exclude DirSwitch-2 due to the high computational cost of GNN training.

As baselines, we compare against GraphSAGE with default (O) and undirected (U) edges, as well as the digraph extension of GraphSAGE proposed by Rossi et al. (2023). Like DirSwitch, Rossi et al. (2023) incorporates multi-directional neighbourhoods but does not include a switch to undirected edges.

Unlike ReachNEs, GNNs typically use a small number of aggregation steps to balance computational efficiency and mitigate over-smoothing (Chen et al., 2020). We set $K = 4$ aggregation steps, following the default in GraphMAEv2 (Hou et al., 2023).

Table 9 presents the results for GraphMAEv2 and CCA-SSG. As observed in the ReachNEs experiments (Section 6.3), performance varies qualitatively between heterophilic and homophilic datasets, with undirected edges generally performing better in homophilic cases. This is particularly evident in Cora, where DirSwitch and undirected edges significantly outperform both the default edges and Rossi et al. (2023).

The only dataset where Rossi et al. (2023) surpasses DirSwitch in accuracy is Snap Patents. As shown in Table 5, using undirected edges causes a substantial accuracy drop for Snap Patents in ReachNEs, and the same trend is observed with GraphSAGE.

## 7 Limitations and future work

This work addresses two key challenges in digraph embedding: the dispersal-obstructing effects of local sinks and the need for expressive embeddings that capture multiple directed neighbourhoods. A limitation of our approach is its reliance on embedding concatenation to achieve both multi-directionality and multi-scale representation, which incurs notable computational overhead. While our results suggest that few directed neighbourhoods and scales are often sufficient—due to the typically short characteristic path lengths of real-world networks (Watts & Strogatz, 1998)—further work is needed to better quantify diminishing returns and to explore more efficient alternatives to concatenation. Additionally, other structural properties of digraphs, such as the role of cycles and self-loops, may be equally important and deserve further investigation.

For tractability, our analysis considered graphs with a single type of supplementary information: node attributes. In reality, graphs are often far more complex, incorporating multiple edge types, attributed edges (as in knowledge graphs (Wang et al., 2017a)), or temporal dynamics where edges evolve over time. While we expect the DirSwitch principle, i.e., separating short-range and long-range behaviours, to generalize to these cases, addressing expressivity in multi-modal graphs will likely require specialized techniques beyond our current concatenation approach.

Another limitation lies in the embedding reduction methods explored. We focused on two widely used techniques: node attribute smoothing for message-passing embeddings and SVD for proximity embeddings. However, other reduction methods, such as structural embeddings, remain unexplored in our analysis. Investigating how local sinks and multi-directional embeddings affect alternative reduction techniques is an important avenue for future research, especially since much of the existing work on structural embeddings is limited to undirected graphs (Donnat et al., 2018; Zhu et al., 2021a; Jin et al., 2021).

Finally, while our benchmark evaluation demonstrates DirSwitch's effectiveness, its real-world utility in unsupervised tasks such as node clustering or anomaly detection remains underexplored. These tasks often lack well-defined ground-truth labels, making evaluation challenging and necessitating further research. Expanding DirSwitch to these domains presents exciting opportunities for future work.

## 8 Conclusion

In this paper, we analysed unsupervised node embedding learning on digraphs through our reachability-based random walk filter framework, ReachNEs. Our analysis identified two key challenges: local sinks obstruct information propagation, while embeddings must also represent directed neighbourhoods to be expressive.

To address these issues, we introduced DirSwitch. By decoupling local and global smoothing, DirSwitch preserves fine-grained directed structure through multi-directional embedding concatenation, while treating the graph as undirected for long-range interactions to mitigate the impact of local sinks.

We demonstrated that DirSwitch significantly enhances embedding quality, achieving higher accuracy on standard node classification benchmarks. Additionally, we showcased its practical effectiveness by outperforming state-of-the-art unsupervised proximity embedding models. Finally, we highlighted DirSwitch's broad applicability by using it to generalize self-supervised graph neural networks to digraphs, illustrating its flexibility across different embedding paradigms.

**Acknowledgments**

This work was partially supported by the Wallenberg AI, Autonomous Systems and Software Program (WASP) funded by the Knut and Alice Wallenberg Foundation. We would also like to thank Petra Poklukar, Marcus Klasson, Aniss Medbouhi, and Miguel Vasco for their valuable feedback and support in improving this paper.

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

# A  Additional details on ReachNEs and DirSwitch

## A.1  Verification of column stochasticity of the random walk normalized adjacency matrices

In Section 2.1, we defined the random walk normalized adjacency matrices as follows:

$$
A_{\mathtt{O}i,j} = \begin{cases} 1 & \text{if } i = j \text{ and } \deg_{\mathtt{O}}(j) = 0, \\ \frac{A_{i,j}}{\deg_{\mathtt{O}}(j)} & \text{otherwise}, \end{cases}
\qquad
A_{\mathtt{I}i,j} = \begin{cases} 1 & \text{if } i = j \text{ and } \deg_{\mathtt{I}}(j) = 0, \\ \frac{A_{j,i}}{\deg_{\mathtt{I}}(j)} & \text{otherwise}, \end{cases}
$$

$$
A_{\mathtt{U}i,j} = \begin{cases} 1 & \text{if } i = j \text{ and } \deg(j) = 0, \\ \frac{A_{\mathrm{undir}\,i,j}}{\deg(j)} & \text{otherwise}. \end{cases}
$$

These definitions ensure that each matrix is column stochastic, meaning that each column sums to one. We verify this property below:

$$
\sum_{k=1}^{n} A_{\mathtt{O}k,j} = \begin{cases} 1 & \text{if } \deg_{\mathtt{O}}(j) = 0 \\ \frac{1}{\deg_{\mathtt{O}}(j)} \sum_{k=1}^{n} A_{k,j} & \text{if } \deg_{\mathtt{O}}(j) > 0 \end{cases} = \begin{cases} 1 & \text{if } \deg_{\mathtt{O}}(j) = 0 \\ \frac{\deg_{\mathtt{O}}(j)}{\deg_{\mathtt{O}}(j)} & \text{if } \deg_{\mathtt{O}}(j) > 0 \end{cases} = 1,
$$

$$
\sum_{k=1}^{n} A_{\mathtt{I}k,j} = \begin{cases} 1 & \text{if } \deg_{\mathtt{I}}(j) = 0 \\ \frac{1}{\deg_{\mathtt{I}}(j)} \sum_{k=1}^{n} A_{j,k} & \text{if } \deg_{\mathtt{I}}(j) > 0 \end{cases} = \begin{cases} 1 & \text{if } \deg_{\mathtt{I}}(j) = 0 \\ \frac{\deg_{\mathtt{I}}(j)}{\deg_{\mathtt{I}}(j)} & \text{if } \deg_{\mathtt{I}}(j) > 0 \end{cases} = 1,
$$

$$
\sum_{k=1}^{n} A_{\mathtt{U}k,j} = \begin{cases} 1 & \text{if } \deg(j) = 0 \\ \frac{1}{\deg(j)} \sum_{k=1}^{n} A_{\mathrm{undir}\,k,j} & \text{if } \deg(j) > 0 \end{cases} = \begin{cases} 1 & \text{if } \deg(j) = 0 \\ \frac{\deg(j)}{\deg(j)} & \text{if } \deg(j) > 0 \end{cases} = 1.
$$

## A.2  Message-passing embeddings for directed edges

In Section 2.3.2, we highlighted the connection between the reachability message-passing reduction using $\boldsymbol{R}(\boldsymbol{A_{\mathtt{U}}})$ and the undirected message-passing graph neural network GraphSAGE (Hamilton et al., 2017) in Equation 7. Here, we present the corresponding formulas for directed cases:

$$
\boldsymbol{A_{\mathtt{O}}^{\mathsf{T}} H} = \boldsymbol{D_{\mathtt{O}}^{-1} A^{\mathsf{T}} H}, \qquad [\boldsymbol{A_{\mathtt{O}}^{\mathsf{T}} H}]_{i,:} = \sum_{j=1}^{n} A_{\mathtt{O}j,i} \boldsymbol{H}_{j,:} = \frac{1}{\deg_{\mathtt{O}}(i)} \sum_{j:(i,j)\in\mathbb{M}} \boldsymbol{H}_{j,:}, \tag{11}
$$

$$
\boldsymbol{A_{\mathtt{I}}^{\mathsf{T}} H} = \boldsymbol{D_{\mathtt{I}}^{-1} A H}, \qquad [\boldsymbol{A_{\mathtt{I}}^{\mathsf{T}} H}]_{i,:} = \sum_{j=1}^{n} A_{\mathtt{I}j,i} \boldsymbol{H}_{j,:} = \frac{1}{\deg_{\mathtt{I}}(i)} \sum_{j:(j,i)\in\mathbb{M}} \boldsymbol{H}_{j,:}. \tag{12}
$$

Note that in Equation 11, messages are aggregated over *outgoing* edges of node $i$, meaning that they flow opposite to edge directions. Conversely, in Equation 12, messages are aggregated along the *incoming*, following the edge directions.

## A.3  Proximity embeddings and structural equivalence

We extend Section 2.3.1 by deriving Equation 4, which relates the distances between reachability vectors in $\boldsymbol{R}$ to distances between embedding vectors in $\boldsymbol{Z} = \begin{bmatrix} \boldsymbol{Z_U} & \boldsymbol{Z_V} \end{bmatrix}$. For convenience, we restate the equation:

$$
\|\boldsymbol{R}_{i,:} - \boldsymbol{R}_{j,:}\|_2^2 + \|\boldsymbol{R}_{:,i} - \boldsymbol{R}_{:,j}\|_2^2 = \left\| (\boldsymbol{Z}_{i,:} - \boldsymbol{Z}_{j,:}) \sqrt{\hat{\boldsymbol{\Sigma}}} \right\|_2^2, \qquad \hat{\boldsymbol{\Sigma}} = \begin{bmatrix} \boldsymbol{\Sigma}_{:q,:q} & \boldsymbol{0} \\ \boldsymbol{0} & \boldsymbol{\Sigma}_{:q,:q} \end{bmatrix}. \tag{13}
$$

Assuming $\mathrm{rank}(\boldsymbol{R}) = q$, the $n - q$ smallest singular values vanish, leading to:

$$
\boldsymbol{R} = \boldsymbol{U}_{:,:q} \boldsymbol{\Sigma}_{:q,:q} (\boldsymbol{V}_{:,:q})^{\mathsf{T}} = \boldsymbol{U}_{:,:q} \sqrt{\boldsymbol{\Sigma}_{:q,:q}} \left( \boldsymbol{V}_{:,:q} \sqrt{\boldsymbol{\Sigma}_{:q,:q}} \right)^{\mathsf{T}} = \boldsymbol{Z_U} \boldsymbol{Z_V}^{\mathsf{T}}.
$$

We can then express each row of $\boldsymbol{R}$ as the inner product between the corresponding row in $\boldsymbol{Z_U}$ and $\boldsymbol{Z_V}$, and similarly, each column of $\boldsymbol{R}$ can be expressed as the inner product between a row in $\boldsymbol{Z_V}$ and $\boldsymbol{Z_U}$:

$$\boldsymbol{R}_{i,:} = \boldsymbol{Z_U}_{i,:}\boldsymbol{Z_V}^\intercal, \qquad\qquad \boldsymbol{R}_{:,i} = \boldsymbol{Z_U}(\boldsymbol{Z_V}_{i,:})^\intercal.$$

We now derive Equation 13, using the simplified notation $\boldsymbol{U} = \boldsymbol{U}_{:,:q}$, $\boldsymbol{\Sigma} = \boldsymbol{\Sigma}_{:q,:q}$, and $\boldsymbol{V} = \boldsymbol{V}_{:,:q}$:

$$\|\boldsymbol{R}_{i,:} - \boldsymbol{R}_{j,:}\|_2^2 + \|\boldsymbol{R}_{:,i} - \boldsymbol{R}_{:,j}\|_2^2 = \left\|\boldsymbol{Z_U}_{i,:}\boldsymbol{Z_V}^\intercal - \boldsymbol{Z_U}_{j,:}\boldsymbol{Z_V}^\intercal\right\|_2^2 + \left\|\boldsymbol{Z_U}(\boldsymbol{Z_V}_{i,:})^\intercal - \boldsymbol{Z_U}(\boldsymbol{Z_V}_{j,:})^\intercal\right\|_2^2 \tag{14}$$

$$= \left\|\left(\boldsymbol{Z_U}_{i,:} - \boldsymbol{Z_U}_{j,:}\right)\boldsymbol{Z_V}^\intercal\right\|_2^2 + \left\|\boldsymbol{Z_U}\left(\boldsymbol{Z_V}_{i,:} - \boldsymbol{Z_V}_{j,:}\right)^\intercal\right\|_2^2 \tag{15}$$

$$= \left\|\left(\boldsymbol{Z_U}_{i,:} - \boldsymbol{Z_U}_{j,:}\right)\sqrt{\boldsymbol{\Sigma}}\boldsymbol{V}^\intercal\right\|_2^2 + \left\|\boldsymbol{U}\sqrt{\boldsymbol{\Sigma}}\left(\boldsymbol{Z_V}_{i,:} - \boldsymbol{Z_V}_{j,:}\right)^\intercal\right\|_2^2 \tag{16}$$

$$= \left\|\left(\boldsymbol{Z_U}_{i,:} - \boldsymbol{Z_U}_{j,:}\right)\sqrt{\boldsymbol{\Sigma}}\right\|_2^2 + \left\|\sqrt{\boldsymbol{\Sigma}}\left(\boldsymbol{Z_V}_{i,:} - \boldsymbol{Z_V}_{j,:}\right)^\intercal\right\|_2^2 \tag{17}$$

$$= \sum_{k=1}^{q}\sqrt{\Sigma_{k,k}}\left(Z_{\boldsymbol{U}\,i,k} - Z_{\boldsymbol{U}\,j,k}\right)^2 + \sum_{k=1}^{q}\sqrt{\Sigma_{k,k}}\left(Z_{\boldsymbol{V}\,i,k} - Z_{\boldsymbol{V}\,j,k}\right)^2 \tag{18}$$

$$= \sum_{k=1}^{2q}\sqrt{\hat{\Sigma}_{k,k}}\left(Z_{i,k} - Z_{j,k}\right)^2 \tag{19}$$

$$= \left\|\left(\boldsymbol{Z}_{i,:} - \boldsymbol{Z}_{j,:}\right)\sqrt{\hat{\boldsymbol{\Sigma}}}\right\|_2^2 \tag{20}$$

The transition from Equation 16 to Equation 17 follows from the orthogonality of $\boldsymbol{U}$ and $\boldsymbol{V}$, i.e., $\boldsymbol{U}^\intercal\boldsymbol{U} = \boldsymbol{I}_q$ and $\boldsymbol{V}^\intercal\boldsymbol{V} = \boldsymbol{I}_q$.

### A.4 Computational complexity of ReachNEs

In this section, we discuss the time and memory complexity of ReachNEs and how to efficiently implement it for embedding learning on large graphs. The primary computational bottleneck is the $n \times n$ reachability matrix $\boldsymbol{R}$, which is generally dense. Storing $\boldsymbol{R}$ explicitly is infeasible for even moderately large graphs—for instance, a graph with one million nodes would require over 3TB of memory.

For message-passing embeddings, the iterative algorithm described in Equation 6 offers a practical way to bypass the need to store $\boldsymbol{R}$ explicitly. For convenience, we restate it below:

$$\text{For } k \in \{1, 2, \dots\}, \quad \begin{cases} \boldsymbol{H}^{(k)} &= \boldsymbol{A}_*^\intercal \boldsymbol{H}^{(k-1)}, \\ \boldsymbol{Z}^{(k)} &= \boldsymbol{Z}^{(k-1)} + \mathrm{P}_{\mathtt{w}}(k)\boldsymbol{H}^{(k)}, \end{cases} \qquad \boldsymbol{H}^{(0)} = \boldsymbol{X}, \quad \boldsymbol{Z}^{(0)} = \mathrm{P}_{\mathtt{w}}(0)\boldsymbol{X}. \tag{21}$$

Rather than computing $\boldsymbol{R}$ in full and multiplying it with the node feature matrix $\boldsymbol{X} \in \mathbb{R}^{n \times d}$, this iterative formulation directly constructs the embedding matrix $\boldsymbol{Z}^{(K)} = \boldsymbol{R}^\intercal\boldsymbol{X}$ through sequential summation. This reduces the memory complexity from quadratic to linear, specifically $\mathcal{O}(nd+m)$, where $nd$ accounts for the embedding matrices $\boldsymbol{H}^{(k)}$ and $\boldsymbol{Z}^{(k)}$, and $m$ is the number of non-zero entries in the sparse adjacency matrix $\boldsymbol{A}_*$.

The time complexity of the iterative message-passing algorithm in Equation 21 is $\mathcal{O}(Kmd+Knd)$, where $md$ arises from the sparse-dense matrix multiplication $\boldsymbol{A}_*^\intercal\boldsymbol{H}^{(k-1)}$, and $nd$ comes from the embedding summation update. Both operations are efficiently supported by existing GPU libraries (Paszke et al., 2019; Fey & Lenssen, 2019), making the method highly practical in large-scale settings.

By contrast, proximity ReachNEs presents greater memory challenges. Both standard and randomized SVD methods (Halko et al., 2011) require multiple passes over all elements of the input matrix, which is problematic since storing $\boldsymbol{R}$ in memory is intractable, and recomputing it on-the-fly is computationally prohibitive.

To address this, we adopt the *single-pass randomized SVD* (SP-rSVD) algorithm proposed by Yu et al. (2017). SP-rSVD follows the general structure of randomized SVD (rSVD) (Halko et al., 2011), but avoids multiple passes over the input. First, the reachability matrix $\boldsymbol{R}$ is projected into a lower-dimensional subspace via a

random matrix $\boldsymbol{\Omega} \in \mathbb{R}^{n \times p}$ to compute $\boldsymbol{G} = \boldsymbol{R}\boldsymbol{\Omega}$. Then, $\boldsymbol{G}$ is used as a proxy to approximate the left singular subspace of $\boldsymbol{R}$.

Unlike standard rSVD, SP-rSVD also computes a second projection $\boldsymbol{F} = \boldsymbol{R}^\mathsf{T}\boldsymbol{G} \in \mathbb{R}^{n \times p}$ in the same pass, enabling accurate SVD approximation without re-accessing $\boldsymbol{R}$. Once $\boldsymbol{G}$ and $\boldsymbol{F}$ are computed, the remaining SVD steps proceed efficiently on these compressed matrices. We refer the reader to the original paper by Yu et al. (2017) for implementation details and theoretical guarantees.

To compute $\boldsymbol{G}$ and $\boldsymbol{F}$ efficiently in a single pass over $\boldsymbol{R}$, we combine the SP-rSVD algorithm with the sequential summation strategy described in Equation 21. In particular, a single row of $\boldsymbol{R}$ can be obtained by setting $\boldsymbol{X} = \boldsymbol{I}_{n:,i}$, yielding $\boldsymbol{R}_{i,:}^\mathsf{T} = \boldsymbol{R}^\mathsf{T}\boldsymbol{I}_{n:,i}$, where $\boldsymbol{I}_{n:,i}$ is the $i$th column on $\boldsymbol{I}_n$.

Using this approach, the matrices $\boldsymbol{G}$ and $\boldsymbol{F}$ can be constructed incrementally, as shown in Equation 22, starting with initializations $\boldsymbol{G} = \boldsymbol{0}_{n \times p}$ and $\boldsymbol{F} = \boldsymbol{0}_{n \times p}$:

$$
\begin{aligned}
&\texttt{For } i \in \{1, 2, \ldots, n\} : \\
&\quad \text{Compute } \boldsymbol{R}_{i,:} \text{ via Equation 21 with } \boldsymbol{X} = \boldsymbol{I}_{n:,i} \\
&\quad \boldsymbol{g} = \boldsymbol{R}_{i,:}\boldsymbol{\Omega} \\
&\quad \boldsymbol{G}_{i,:} = \boldsymbol{g} \\
&\quad \boldsymbol{F} = \boldsymbol{F} + \boldsymbol{R}_{i,:}^\mathsf{T}\boldsymbol{g}.
\end{aligned}
\tag{22}
$$

In practice, it is advantageous to process multiple rows of $\boldsymbol{R}$ in parallel batches to maximize GPU throughput. Letting $b$ denote the batch size, the memory complexity of the algorithm becomes $\mathcal{O}(np + nb + m)$, where $m$ is the number of non-zero entries in the adjacency matrix.

The total time complexity of proximity ReachNEs is $\mathcal{O}(nm + n^2p + np^2)$. The first term, $nm$, accounts for computing all $n$ rows of $\boldsymbol{R}$ using sparse matrix multiplications. The $n^2p$ term arises from evaluating the remaining steps in Equation 22, and $np^2$ corresponds to the final SVD step in SP-rSVD after computing $\boldsymbol{G}$ and $\boldsymbol{F}$ (Yu et al., 2017).

To accelerate this process, our implementation leverages the PyTorch framework, with built-in support for both single- and multi-GPU environments. A further advantage over the original MATLAB implementation of SP-rSVD is PyTorch's support for automatic differentiation, which opens up new research directions for self-supervised proximity embedding methods based on differentiable reachability matrices.

### A.5 Computational complexity of DirSwitch and run time results

### A.5.1 Computational complexity

We begin by analysing the time and space complexity of DirSwitch-$r$, followed by empirical run time measurements of our GPU-accelerated implementation.

The computational complexity of DirSwitch depends on the following variables:

- $n$: number of nodes.

- $m$: number of edges.

- $p$: embedding dimensionality.

- $r$: number of multi-directional steps.

- $s$: number of walk length distributions.

- $K$: number of walk steps (i.e., reachability terms).

- $b$: batch size used in SP-rSVD (proximity version only; see Section A.4).

- $d$: number of node features (message-passing version only).

We separately analyse the complexity of proximity and message-passing variants. We begin with the message-passing version, which is simpler due to the absence of batching.

**Message-passing DirSwitch.**  Message-passing DirSwitch-$r$ is implemented using the algorithm in Equation 6, which performs $K$ sparse-dense matrix multiplications, each with time complexity $\mathcal{O}(md)$. This results in an overall cost of $\mathcal{O}(Kmd)$ per walk.

Each resulting $n \times d$ matrix is then reduced to $\frac{p}{s2^r}$ dimensions using PCA via SVD, with time complexity $\mathcal{O}(nd^2)$ (Golub & Van Loan, 2013, Ch. 5.5.6). This reduction is performed for all $s \cdot 2^r$ combinations of walk length distributions and directed neighbourhoods, resulting in a total time complexity of $\mathcal{O}(s2^r(Kmd+nd^2))$.

In terms of space complexity, we must store the $m$ nonzero entries of the normalized adjacency matrix, the intermediate $s$ reachability representations of shape $n \times d$, and the final embedding matrix $\boldsymbol{Z} \in \mathbb{R}^{n \times p}$. Our implementation computes the $s$ multi-scale walks in parallel using tensors of shape $s \times n \times d$, while the $2^r$ directed neighbourhoods are processed sequentially. The resulting space complexity is $\mathcal{O}(m + snd + np)$.

**Proximity DirSwitch.**  Proximity DirSwitch-$r$ is implemented using the algorithm in Equation 22. Assuming a batch size $b$, the algorithm runs in $\frac{n}{b}$ iterations. Each iteration computes a slice of the reachability matrix via $K$ sparse-dense matrix multiplications with cost $\mathcal{O}(Kmb)$, followed by projection onto $\boldsymbol{\Omega} \in \mathbb{R}^{n \times p}$ and the computation of $\boldsymbol{F}$, both requiring $\mathcal{O}(bnp)$. Repeating for all $\frac{n}{b}$ batches yields a total cost of $\mathcal{O}(Kmn + n^2p)$.

After constructing the sketch matrices, SP-rSVD computes the final embeddings in time $\mathcal{O}(np^2)$ (Yu et al., 2017), which is negligible when $p \ll n$. As with message-passing DirSwitch, this process is repeated for all $s \cdot 2^r$ combinations of walk length distributions and edge direction patterns, resulting in total time complexity $\mathcal{O}(s2^r(Kmn + n^2p))$.

In terms of space, the algorithm requires storage of the $m$ edges, one $n \times b$ slice of the reachability matrix at a time, and the $n \times p$ matrices $\boldsymbol{\Omega}$, $\boldsymbol{G}$, and $\boldsymbol{F}$ used by SP-rSVD. Our implementation stores $s$ copies of $\boldsymbol{\Omega}$, one per walk length distribution, computed in parallel. The total space complexity is $\mathcal{O}(snp + snb + m)$.

**Discussion.**  The use of SP-rSVD assumes $p \ll n$, and its numerical accuracy may degrade as $p$ approaches $n$. For small graphs, we recommend computing the full reachability matrix directly and applying SVD without batching. This yields a time complexity of $\mathcal{O}(s2^r(Kmn + n^3))$ and space complexity of $\mathcal{O}(sn^2 + m)$.

While SP-rSVD helps improve the scalability of proximity DirSwitch-$r$ by mitigating memory usage (see Section A.4), the method still incurs a quadratic time cost in $n$. In contrast, message-passing DirSwitch-$r$ has linear time complexity in $n$, making it more scalable for large graphs.

A potential alternative to improve proximity scalability is a hybrid approach: apply SVD to the sparse adjacency matrix to generate initial proximity embeddings, then use these as node attributes in a message-passing DirSwitch-$r$ model. Investigating the effectiveness and scalability of this hybrid method is a promising direction for future work.

### A.5.2  Run time results

To evaluate practical run times, we conducted a small-scale scalability experiment. We generated synthetic undirected graphs using the Barabási–Albert model (Barabási & Albert, 1999), with an average degree of 2. We then measured the run time of DirSwitch while varying one parameter at a time—$n$, $r$, $s$, $p$, or $d$—while keeping the others fixed. For proximity DirSwitch, we fixed $n = 2^{16} = 65,536$, $r = 2$, $s = 3$, and $p = 256$. For message-passing DirSwitch, we used $n = 2^{20} = 1,048,576$, $r = 2$, $s = 3$, $p = 512$, and $d = 128$. Each experiment was repeated three times.

We tested proximity DirSwitch on three computational environments in Google Cloud: (i) 2 Nvidia L4 (24GB) GPUs, 24 vCPUs @ 2.20GHz, and 96GB RAM; (ii) 1 Nvidia L4 (24GB) GPU, 16 vCPUs @ 2.20GHz, and 64GB RAM; and (iii) CPU-only with 16 vCPUs @ 2.20GHz and 64GB RAM. These are denoted as GPU-2, GPU-1, and CPU-0, respectively. As our current implementation of message-passing DirSwitch does not support multi-GPU computation, we only evaluated it in the GPU-1 and CPU-0 environments.

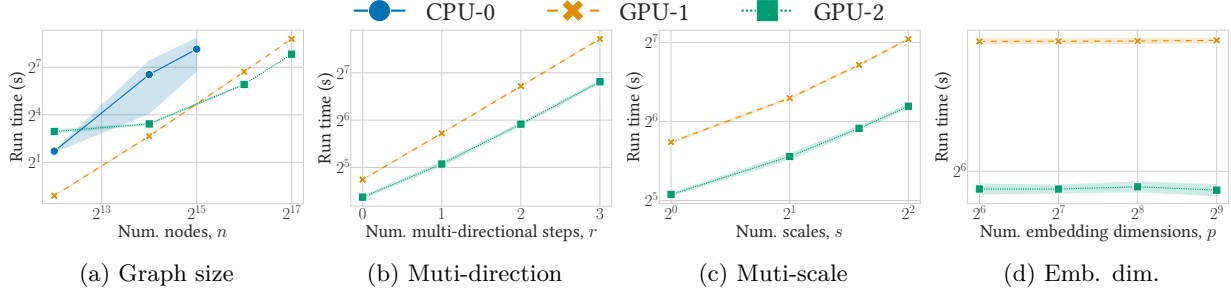

(a) Graph size      (b) Muti-direction      (c) Muti-scale      (d) Emb. dim.

Figure 8: Scalability results for proximity DirSwitch-$r$. Each subfigure plots run time (in seconds) against a key parameter in the DirSwitch time complexity. Markers indicate the mean over three runs, with shaded areas showing standard deviation. Line colour and style correspond to the number of GPUs used: 0, 1, or 2.

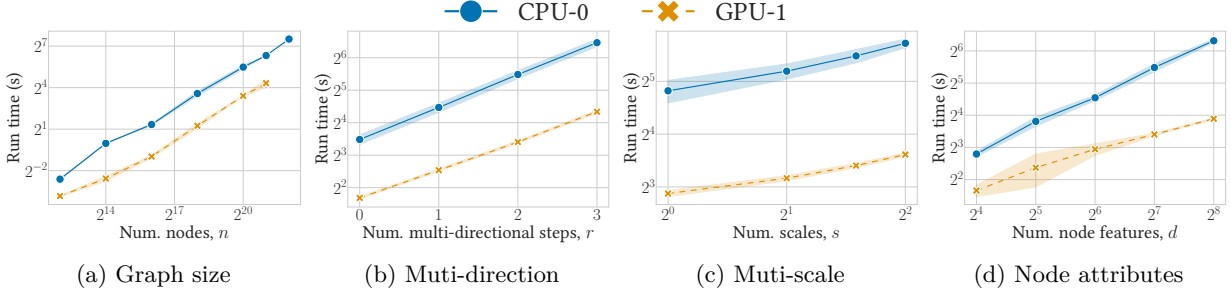

(a) Graph size      (b) Muti-direction      (c) Muti-scale      (d) Node attributes

Figure 9: Scalability results for message-passing DirSwitch-$r$. Each subfigure plots run time (in seconds) against a key parameter in the DirSwitch time complexity. Markers show the mean over three runs; shaded areas indicate standard deviation. Line colour and style correspond to the number of GPUs used: 0 or 1.

Figures 8 and 9 show the average run times for proximity and message-passing DirSwitch, respectively, with line colours indicating the compute environment. These results clearly demonstrate the advantage of GPU acceleration, as both models run significantly faster on GPU than on CPU. For proximity DirSwitch, CPU runtimes were prohibitively slow at $n = 2^{16}$, and we therefore aborted those runs—hence the missing data points. Although multi-GPU setups introduce some overhead, they become advantageous for larger graphs.

Most trends align with theoretical expectations. Figure 8a confirms the quadratic time complexity in $n$ for proximity DirSwitch, while Figure 9a shows linear scaling for the message-passing variant. The exponential growth in the number of directed neighbourhoods, $2^r$, is reflected in the roughly exponential growth in run times shown in Figures 8b and 9b. In Figures 8c and 9c, we observe sublinear growth in run time with respect to $s$, due to parallelization across the multi-scale computations.

One unexpected result appears in Figure 8d: the run time of proximity DirSwitch remains largely unaffected by the embedding dimensionality $p$. This is counterintuitive, as the theoretical time complexity includes an $n^2 p$ term. However, this term is dominated in practice by the $Kmn$ cost of computing reachability. A similar pattern is observed in Figure 9d, where run time increases linearly with $d$ for message-passing DirSwitch—indicating that the $Kmd$ term outweighs the $nd^2$ term. These findings suggest that optimizing the reachability computation is the most promising avenue for improving the overall efficiency of DirSwitch.

Tables 10 and 11 report measured run times for the experiments in Section 6.3. These were run on varying computational environments, so runtimes may differ slightly across datasets. We only report results for the slowest datasets. On the others, runtime typically falls between 1–3 seconds.

The results confirm the trends observed in the scalability experiments. Since the implementation parallelizes over the number of scales ($s$) while computing different direction sequences ($r$) sequentially, $r$ has a significantly greater impact on runtime than $s$.

The Cora datasets yield the longest runtimes, primarily because they contain the largest number of node features ($d = 8710$; see Table 2).

Table 10: Embedding computation duration for message-passing ReachNEs with $p = 1024$ embedding dimensions. Columns correspond to different datasets and multi-scale walk length distributions, while rows represent various edge direction specifiers.

| Edge directions | Cora-ML | | | | | Cora | | | | |
|---|---|---|---|---|---|---|---|---|---|---|
| | Geom | Pois | Geom-$\mathcal{U}$ | Binom-3 | Geom-4 | Geom | Pois | Geom-$\mathcal{U}$ | Binom-3 | Geom-4 |
| Default | 2s | 2s | 4s | 6s | 8s | 1m 24s | 1m 25s | 2m 49s | 5m 33s | 6m 2s |
| Undirected | 2s | 4s | 4s | 7s | 8s | 1m 23s | 1m 24s | 2m 49s | 4m 26s | 5m 40s |
| MultiDir-1 | 3s | 4s | 8s | 13s | 17s | 2m 47s | 2m 49s | 5m 36s | 9m 43s | 11m 36s |
| MultiDir-2 | 7s | 7s | 16s | 27s | 35s | 5m 33s | 5m 37s | 11m 10s | 19m 13s | 22m 56s |
| MultiDir-3 | 19s | 19s | 43s | 1m 9s | 1m 30s | 10m 54s | 11m 2s | 21m 57s | 37m 11s | 44m 48s |
| DirSwitch-1 | 3s | 3s | 8s | 14s | 18s | 2m 47s | 2m 47s | 5m 35s | 8m 52s | 11m 18s |
| DirSwitch-2 | 7s | 7s | 17s | 29s | 38s | 5m 35s | 5m 37s | 11m 10s | 17m 58s | 22m 40s |
| DirSwitch-3 | 14s | 15s | 33s | 59s | 1m 15s | 11m 6s | 11m 13s | 22m 16s | 36m 14s | 45m 21s |

| Edge directions | Pokec | | | | | Snap-Patents | | | | |
|---|---|---|---|---|---|---|---|---|---|---|
| | Geom | Pois | Geom-$\mathcal{U}$ | Binom-3 | Geom-4 | Geom | Pois | Geom-$\mathcal{U}$ | Binom-3 | Geom-4 |
| Default | 18s | 22s | 23s | 21s | 21s | 26s | 26s | 45s | 57s | 56s |
| Undirected | 22s | 18s | 19s | 21s | 21s | 31s | 37s | 42s | 53s | 1m 39s |
| MultiDir-1 | 44s | 44s | 47s | 57s | 1m 3s | 59s | 1m 24s | 2m 8s | 2m 45s | 2m 2s |
| MultiDir-2 | 1m 48s | 1m 30s | 1m 54s | 1m 41s | 1m 46s | 1m 58s | 1m 59s | 4m 57s | 6m 27s | 7m 34s |
| MultiDir-3 | 4m 10s | 4m 11s | 4m 21s | 4m 37s | 4m 48s | 4m 56s | 4m 56s | 6m 39s | 8m 23s | 9m 52s |
| DirSwitch-1 | 34s | 43s | 46s | 48s | 42s | 1m 54s | 1m 1s | 1m 23s | 2m 53s | 3m 34s |
| DirSwitch-2 | 1m 26s | 1m 22s | 1m 28s | 1m 17s | 1m 22s | 2m 4s | 2m 2s | 5m 7s | 3m 59s | 7m 53s |
| DirSwitch-3 | 2m 54s | 2m 54s | 3m 0s | 3m 8s | 3m 26s | 6m 32s | 7m 18s | 5m 29s | 11m 40s | 8m 15s |

Table 11: Embedding computation duration for proximity ReachNEs with $p = 1024$ embedding dimensions. Columns correspond to different datasets and multi-scale walk length distributions, while rows represent various edge direction specifiers.

| Edge directions | CoCite | | | | | PubMed | | | | | Cora (Subelj) | | | | |
|---|---|---|---|---|---|---|---|---|---|---|---|---|---|---|---|
| | Geom | Pois | Geom-$\mathcal{U}$ | Binom-3 | Geom-4 | Geom | Pois | Geom-$\mathcal{U}$ | Binom-3 | Geom-4 | Geom | Pois | Geom-$\mathcal{U}$ | Binom-3 | Geom-4 |
| Default | 3s | 3s | 4s | 5s | 6s | 1s | 1s | 1s | 1s | 1s | 1s | 1s | 1s | 1s | 2s |
| Undirected | 4s | 5s | 6s | 7s | 8s | 1s | 1s | 2s | 3s | 3s | 1s | 1s | 1s | 2s | 2s |
| MultiDir-1 | 6s | 6s | 8s | 9s | 11s | 1s | 1s | 2s | 2s | 3s | 1s | 1s | 2s | 2s | 3s |
| MultiDir-2 | 12s | 12s | 15s | 19s | 22s | 3s | 3s | 4s | 4s | 5s | 3s | 3s | 4s | 5s | 6s |
| MultiDir-3 | 37s | 37s | 57s | 1m 17s | 1m 37s | 7s | 7s | 10s | 14s | 18s | 10s | 10s | 15s | 21s | 26s |
| DirSwitch-1 | 8s | 8s | 10s | 13s | 15s | 2s | 2s | 3s | 4s | 4s | 2s | 2s | 2s | 3s | 3s |
| DirSwitch-2 | 14s | 15s | 19s | 25s | 28s | 3s | 3s | 4s | 6s | 7s | 3s | 3s | 4s | 6s | 7s |
| DirSwitch-3 | 28s | 28s | 36s | 47s | 53s | 6s | 6s | 8s | 11s | 13s | 6s | 6s | 9s | 11s | 13s |

Finally, Table 12 shows the run times of the models used in the node classification benchmark in Section 6.4. DirSwitch ranks among the fastest methods, alongside BLADE and NERD. Note that graph size and hyperparameter settings vary across these results.

## A.6   Guide to hyperparameter selection for DirSwitch

DirSwitch-$r$ embeddings are governed by four key hyperparameters: the embedding dimensionality $p$, the multi-directional neighbourhood radius $r$, the number of embedding scales $s$, and the choice of walk length distributions $P_w(k; \tau)$. Each of these influences both embedding quality and computational cost. This section offers practical guidelines for selecting these hyperparameters, based on our experimental findings and experience with DirSwitch.

**Walk length distribution.**  We begin with the walk length distribution $P_w(k - l_o; \tau)$, where $l_o$ denotes an offset. For a single-scale setup ($s = 1$), we recommend using $\text{Pois}(k - l_o; \tau = 2)$, with $l_o = 0$ for message-passing and $l_o = 1$ for proximity embeddings, as explained in Section 2.3.1. This setting emphasizes the $k = 1$ and $k = 2$ neighbourhoods, which typically offer the most structural diversity, while still incorporating

Table 12: Embedding computation duration for proximity node embedding models. Average and standard deviations are computed using 5 random seeds.

| MODEL | FLY LARVA | EU-EMAIL | POLBLOGS | CoCite | PUBMED | CORA (SUBELJ) |
|---|---|---|---|---|---|---|
| HOPE[1] | 58s | 9s | 7s | 41M 47s | 10M 46s | 4M 45s |
| APP[2] | 15s | 3s | 1s | 8M 41s | 1M 54s | 2M 14s |
| NERD[3] | 9s | 28s | 4s | 12s | 18s | 11s |
| DGGAN[4] | 29M 10s | 6M 19s | 2M 24s | 1H 33M | 7M 25s | 51M 47s |
| BLADE[5] | 5s | 1s | 1s | 23s | 11s | 21s |
| DIRSWITCH | 7s | 677MS | 287MS | 2M 31s | 36s | 34s |

[1] OU ET AL. (2016)   [2] ZHOU ET AL. (2017)   [3] KHOSLA ET AL. (2020)   [4] ZHU ET AL. (2021B)
[5] VIRINCHI & SALADI (2023)

information from $k = 0$, 3, and 4 (see Figure 11a). Moreover, our graph alignment results suggest that $\tau \approx 2$ tends to correspond to peak expressivity (see Figures 7 and 15). Lower values of $\tau$ risk overly localized embeddings, while higher values may lead to over-smoothing.

When using multiple distributions ($s > 1$), it is important to avoid excessive overlap between them, as this leads to redundant embedding dimensions. We find that Geometric distributions with small $\tau$ and varying offsets are particularly effective, as they place their modes at distinct distances from the source node. This configuration performed especially well on the Pokec dataset (Section G).

As a general rule for $s \geq 3$, we recommend the following family of walk distributions:

$$\{\text{Geom}(k - s_i; \tau = 1 + 0.5s_i) \mid s_i \in \{0, 1, \ldots, s - 1\}\}. \tag{23}$$

For $s = 2$, it is preferable to combine one short-range and one long-range distribution, such as $\text{Geom}(k; \tau = 2)$ and $\text{Pois}(k - 2; \tau = 3)$.

**Expressivity and computation trade-offs.** We now discuss the parameters $p$, $r$, and $s$, which govern the trade-off between model expressivity and computational cost. Increasing any of these parameters typically enhances embedding expressivity and improves downstream task performance, as demonstrated in Section 6. However, this comes at the cost of increased computation, as analysed in Section A.5.

Notably, increasing $r$ and $s$ leads to diminishing returns for two key reasons. First, larger values result in embeddings that capture increasingly broader neighbourhoods. Since most real-world graphs exhibit the small-world property (Watts & Strogatz, 1998)—i.e., short diameters and small average path lengths (see Table 2)—larger neighbourhoods quickly converge to covering the full graph, reducing the embeddings' ability to distinguish local structure.

Our graph alignment experiments support this: using the Poisson distribution, the most discriminative and noise-robust embeddings arise from walks of 1 to 5 steps (see Figures 7 and 15). We therefore recommend keeping $r$ and the support of the walk length distribution within 5–6 steps, unless the graph's structure specifically warrants longer ranges. Nonetheless, further research is needed to quantify these diminishing returns and understand how they vary across graph types and downstream tasks.

A second source of diminishing returns arises from the constraint of fixed embedding dimensionality $p$. Since DirSwitch-$r$ concatenates embeddings across $s$ scales and $2^r$ directed neighbourhoods, each $(\boldsymbol{\sigma}, P_w)$ pair contributes only $\frac{p}{s2^r}$ dimensions. If $s$ or $r$ are set too high relative to $p$, each component becomes overly compressed, degrading the overall embedding quality.

Increasing $p$ generally improves expressivity, but it also raises computational demands. In particular, excessive values may exceed GPU memory limits, forcing computations to fall back to CPUs—significantly increasing run time, as shown in Section A.5.

Based on our experimental findings with $p = 512$ in Section G, where we observed diminishing returns, we recommend ensuring $p \geq 32s2^r$ as a practical guideline. For instance, with $p = 1024$, we suggest $r = 3$ and $s = 3$ or $s = 4$, which matches the best-performing models in Section 6.3. If hardware permits, $p$, $r$, and $s$ can be increased proportionally while respecting the $p \geq 32s2^r$ rule. For resource-constrained settings, we recommend $r = 1$, $s = 2$, and $p = 256$ as an efficient configuration that still benefits from both multi-directionality and multi-scale expressivity.

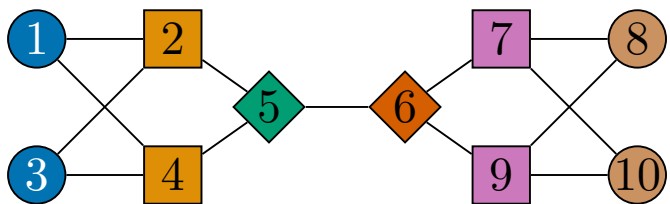

Figure 10: Illustration of structural and automorphic node equivalence. Nodes with the same colour (e.g., nodes 1 and 3) are structurally equivalent, meaning they share identical connections to the same set of nodes. Nodes with the same shape (e.g., nodes 2, 4, 7, and 9) are automorphically equivalent, meaning they are indistinguishable based solely on graph structure without node labels.

## B    Message-passing embeddings and automorphic node equivalence

This section provides additional detail on automorphic node equivalence and proves that message-passing ReachNEs embeddings assign identical representations to automorphic nodes.

Figure 10 illustrates both structural and automorphic node equivalence. Nodes with the same colour are structurally equivalent, while nodes with the same shape are automorphically equivalent. As shown, structural equivalence always implies automorphic equivalence, but not vice versa. Figure 10 also highlights how structural equivalence implies node proximity, whereas automorphically equivalent nodes can be far apart, such as nodes 1 and 10. In fact, automorphic nodes can even exist in separate weakly connected components.

To prove that message-passing ReachNEs embeddings preserve automorphic equivalence, we begin by formalizing the definition. Let $\pi : \mathbb{N} \to \mathbb{N}$ be a permutation of the node set in a graph $\mathcal{G} = (\mathbb{N}, \mathbb{M})$, and let $\boldsymbol{P}_\pi \in \{0,1\}^{n \times n}$ denote the corresponding permutation matrix, where $\boldsymbol{P}_\pi(\pi(i), i) = 1$ for all $i$, and all other entries are zero. An *automorphism* is any such permutation $\pi$ that preserves the edge structure of the graph, i.e., $(i,j) \in \mathbb{M} \iff (\pi(i), \pi(j)) \in \mathbb{M}$. Equivalently, this condition holds if the adjacency matrix satisfies $\boldsymbol{A} = \boldsymbol{P}_\pi \boldsymbol{A} \boldsymbol{P}_\pi^\mathsf{T}$. Two nodes $i$ and $j$ are *automorphically equivalent* if there exists an automorphism $\pi$ such that $\pi(i) = j$, i.e., $\boldsymbol{P}_\pi(j, i) = 1$.

Next, we prove that message-passing ReachNEs produces identical embeddings for automorphic nodes, i.e., $\boldsymbol{P}_\pi \boldsymbol{Z} = \boldsymbol{Z}$, under the *node attribute equivalence* assumption, $\boldsymbol{X} = \boldsymbol{P}_\pi \boldsymbol{X}$, which is guaranteed if $\boldsymbol{X}$ consists of graph-derived features.

We start by showing $\boldsymbol{R} = \boldsymbol{P}_\pi \boldsymbol{R} \boldsymbol{P}_\pi^\mathsf{T}$. To see this, note that by the definition of automorphism, we have $\boldsymbol{A} = \boldsymbol{P}_\pi \boldsymbol{A} \boldsymbol{P}_\pi^\mathsf{T}$. The same holds for the normalized adjacency matrices, $\boldsymbol{A}_\sigma = \boldsymbol{P}_\pi \boldsymbol{A}_\sigma \boldsymbol{P}_\pi^\mathsf{T}$, and their powers, $\boldsymbol{A}_\sigma^k = \boldsymbol{P}_\pi \boldsymbol{A}_\sigma^k \boldsymbol{P}_\pi^\mathsf{T}$. This follows from:

$$\overset{\frown}{\prod_{l=1}^{k}} \boldsymbol{A}_{\sigma_l} = \overset{\frown}{\prod_{l=1}^{k}} \boldsymbol{P}_\pi \boldsymbol{A}_{\sigma_l} \boldsymbol{P}_\pi^\mathsf{T} = \boldsymbol{P}_\pi \boldsymbol{A}_{\sigma_k} \ldots \boldsymbol{A}_{\sigma_2} \underbrace{\boldsymbol{P}_\pi^\mathsf{T} \boldsymbol{P}_\pi}_{\boldsymbol{I}_n} \boldsymbol{A}_{\sigma_1} \boldsymbol{P}_\pi^\mathsf{T} = \boldsymbol{P}_\pi \overset{\frown}{\prod_{l=1}^{k}} \boldsymbol{A}_{\sigma_l} \boldsymbol{P}_\pi^\mathsf{T}.$$

Since the reachability matrix $\boldsymbol{R}$ is a sum of power matrices, see Equation 8, it follows that $\boldsymbol{R}$ is also invariant under automorphism $\boldsymbol{R} = \boldsymbol{P}_\pi \boldsymbol{R} \boldsymbol{P}_\pi^\mathsf{T}$.

To complete the proof, recall that $\boldsymbol{Z} = \boldsymbol{R}^\mathsf{T} \boldsymbol{X}$ represents message-passing embeddings (see Section 2.3.2). We then simply insert $\boldsymbol{R} = \boldsymbol{P}_\pi \boldsymbol{R} \boldsymbol{P}_\pi^\mathsf{T}$ and $\boldsymbol{X} = \boldsymbol{P}_\pi \boldsymbol{X}$:

$$\boldsymbol{P}_\pi \boldsymbol{Z} = \boldsymbol{P}_\pi \boldsymbol{R}^\mathsf{T} \boldsymbol{X} = \boldsymbol{P}_\pi \boldsymbol{R}^\mathsf{T} \underbrace{\boldsymbol{I}_n}_{\boldsymbol{P}_\pi^\mathsf{T} \boldsymbol{P}_\pi} \boldsymbol{X} = \underbrace{\boldsymbol{P}_\pi \boldsymbol{R}^\mathsf{T} \boldsymbol{P}_\pi^\mathsf{T}}_{\boldsymbol{R}^\mathsf{T}} \underbrace{\boldsymbol{P}_\pi \boldsymbol{X}}_{\boldsymbol{X}} = \boldsymbol{R}^\mathsf{T} \boldsymbol{X} = \boldsymbol{Z}.$$

Unlike message-passing ReachNEs, proximity embeddings are obtained via SVD reduction, a global operation that does not necessarily preserve automorphism invariance. Thus, it is generally the case for proximity embeddings that $\boldsymbol{P}_\pi \boldsymbol{Z} \neq \boldsymbol{Z}$. A formal proof for undirected graphs is provided by Zhu et al. (2021a).

## C  Asymptotic behaviour of random walk length distributions

This section expands on the analysis in Section 2.2, focusing on the asymptotic behaviour of different random walk length distributions and their implications for the reachability matrix $\boldsymbol{R}$ as $\tau \to \infty$.

We begin with the Poisson distribution, deriving its associated diffusion differential equation and establishing its well-known connection to the normalized Laplacian. Next, we derive a similar equation for the geometric distribution, demonstrating that, unlike the Poisson case, it does not necessarily converge to a non-informative stationary distribution as $\tau \to \infty$.

We then examine the relationships between distributions in the $K$-truncated setting, proving that:

$$\boldsymbol{R}^{(K)}(\boldsymbol{A}_*; \mathrm{Geom}(\tau)) \to \boldsymbol{R}^{(K)}(\boldsymbol{A}_*; \mathcal{U}(\tau)), \quad \text{as } \tau \to \infty,$$
$$\boldsymbol{R}^{(K)}(\boldsymbol{A}_*; \mathrm{Binom}(\tau)) \to \boldsymbol{R}^{(K)}(\boldsymbol{A}_*; \mathrm{Pois}(\tau)), \quad \text{as } K \to \infty.$$

Finally, we highlight the relation between the binomial distribution and random walks on graphs with self-loops added at each node.

### C.1  The Poisson Distribution

We now establish the identity:

$$\boldsymbol{R}(\boldsymbol{A}_*; \mathrm{Pois}(\tau)) = e^{-\tau \boldsymbol{L}_*}, \tag{24}$$

where $\boldsymbol{L}_* = \boldsymbol{I}_n - \boldsymbol{A}_*$ is the normalized graph Laplacian.

Starting from the definition in Equation 2, we rewrite the reachability matrix as:

$$\boldsymbol{R}(\boldsymbol{A}_*; \mathrm{Pois}(\tau)) = \sum_{k=0}^{\infty} e^{-\tau} \frac{\tau^k}{k!} \boldsymbol{A}_*^k = e^{-\tau} e^{\tau \boldsymbol{A}_*} = e^{-\tau \boldsymbol{L}_*},$$

using properties of the matrix exponential (Hall, 2015).

Next, we show that for an initial probability distribution $\boldsymbol{p}^{(0)} \in [0,1]^n$, the solution:

$$\boldsymbol{p}(\tau) = \boldsymbol{R}(\boldsymbol{A}_*, \mathrm{Pois}(\tau)) \boldsymbol{p}^{(0)} \tag{25}$$

satisfies the differential equation:

$$\frac{\mathrm{d}\boldsymbol{p}}{\mathrm{d}\tau} = -\boldsymbol{L}_* \boldsymbol{p}(\tau). \tag{26}$$

*Proof.* From Equation 24, we recall that:

$$\boldsymbol{R}(\boldsymbol{A}_*, \mathrm{Pois}(\tau)) = e^{-\tau \boldsymbol{L}_*}.$$

Setting $\tau = 0$ gives $\boldsymbol{R}(\boldsymbol{A}_*, \mathrm{Pois}(0)) = \boldsymbol{I}_n$, ensuring that the initial condition is satisfied:

$$\boldsymbol{p}(0) = \boldsymbol{p}^{(0)}.$$

Differentiating Equation 25 with respect to $\tau$:

$$\frac{\mathrm{d}\boldsymbol{p}(\tau)}{\mathrm{d}\tau} = \frac{\mathrm{d}}{\mathrm{d}\tau} e^{-\tau \boldsymbol{L}_*} \boldsymbol{p}^{(0)} = -\boldsymbol{L}_* e^{-\tau \boldsymbol{L}_*} \boldsymbol{p}^{(0)} = -\boldsymbol{L}_* \boldsymbol{p}(\tau).$$

Thus, $\boldsymbol{p}(\tau)$ satisfies the differential equation as required. $\qquad \square$

The differential relation in Equation 26 is useful for analysing the dynamics of $\boldsymbol{R}(\boldsymbol{A}_*, \mathrm{Pois}(\tau))$ with respect to $\tau$, allowing us to establish its asymptotic behavior.

Focusing first on the undirected case, the eigenvalue multiplicity of 0 in the normalized Laplacian $\boldsymbol{L}_{\mathtt{U}}$ equals the number of weakly connected components in the graph (Chung, 1997; von Luxburg, 2007). The corresponding eigenvectors $\boldsymbol{u}$ take values:

$$u_i = \frac{\deg(i)}{\sum_{k \in \mathcal{C}(j)} \deg(k)}$$

for each node $i$ within a weakly connected component $\mathcal{C}(j)$. These eigenvectors are the only stationary solutions to Equation 26, as all other eigenvalues of $\boldsymbol{L}_{\mathtt{U}}$ are strictly positive.

Thus, as $\tau \to \infty$, the reachability matrix converges to:

$$R(\boldsymbol{A}_{\mathtt{U}}, \mathrm{Pois}(\tau))_{i,j} \to \frac{\deg(i)}{\sum_{k \in \mathcal{C}(j)} \deg(k)}. \tag{27}$$

Importantly, all nodes within the same weakly connected component have identical asymptotic reachability values, making them indistinguishable under the Poisson walk length distribution for undirected graphs. Consequently, downstream tasks relying on embedding distinguishability, such as graph alignment, are expected to degrade as $\tau$ increases. This effect was also observed in Figures 7d and 7b.

The above analysis provides a good first-order approximation of the Poisson distribution's behaviour on digraphs. However, a precise theoretical analysis using $\boldsymbol{L}_0$ and $\boldsymbol{L}_{\mathtt{I}}$ is more involved and beyond the scope of this paper. Instead, we refer to Veerman & Lyons (2020) for valuable insights. Specifically, the multiplicity of the eigenvalue 0 corresponds to the number of *reaches* in the graph (Veerman & Lyons, 2020, Theorem 4.6). Unlike weakly connected components, multiple reaches can overlap. Consequently, the eigenvectors exhibit a more complex structure (Veerman & Lyons, 2020, Theorem 5.1), and nodes within a reach do not necessarily converge to the same reachability vector as $\tau \to \infty$.

### C.2   The geometric distribution

We now analyse the asymptotic behavior of walk lengths following the geometric distribution. We start with the parameterization $\alpha \in [0, 1)$, related to $\tau$ by $\alpha = \frac{\tau}{1+\tau}$, so that as $\tau \to \infty$, we have $\alpha \to 1$.

As before, we express the reachability matrix as a matrix function:

$$\boldsymbol{R}(\boldsymbol{A}_*; \mathrm{Geom}(\alpha)) = (1 - \alpha) \sum_{k=0}^{\infty} \alpha^k \boldsymbol{A}_*^k = (1 - \alpha) \left( \boldsymbol{I}_n - \alpha \boldsymbol{A}_* \right)^{-1}. \tag{28}$$

Substituting $\alpha = \frac{\tau}{1+\tau}$ and simplifying, we obtain:

$$\begin{aligned}
\boldsymbol{R}(\boldsymbol{A}_*; \mathrm{Geom}(\tau)) &= \frac{1}{1+\tau} \left( \boldsymbol{I}_n - \frac{\tau}{1+\tau} \boldsymbol{A}_* \right)^{-1} \\
&= \left( \boldsymbol{I}_n + \tau(\boldsymbol{I}_n - \boldsymbol{A}_*) \right)^{-1} \\
&= \left( \boldsymbol{I}_n + \tau \boldsymbol{L}_* \right)^{-1}.
\end{aligned}$$

Using this identity, we show that $\boldsymbol{p}(\tau) = \boldsymbol{R}(\boldsymbol{A}_*; \mathrm{Geom}(\tau))\boldsymbol{p}^{(0)}$ satisfies the differential equation:

$$\frac{\mathrm{d}\boldsymbol{p}(\tau)}{\mathrm{d}\tau} = -\left( \boldsymbol{I}_n + \tau \boldsymbol{L}_* \right)^{-1} \boldsymbol{L}_* \boldsymbol{p}(\tau), \quad \boldsymbol{p}(0) = \boldsymbol{p}^{(0)}. \tag{29}$$

*Proof.* The initial condition is satisfied since $\boldsymbol{R}(\boldsymbol{A}_*; \mathrm{Geom}(0)) = \boldsymbol{I}_n$, so $\boldsymbol{p}(0) = \boldsymbol{p}^{(0)}$. To compute the derivative $\frac{\mathrm{d}\boldsymbol{p}(\tau)}{\mathrm{d}\tau}$, we apply the matrix inversion derivative rule (Petersen & Pedersen, 2012, Sec. 2.2):

$$\begin{aligned}
\frac{\mathrm{d}\boldsymbol{p}(\tau)}{\mathrm{d}\tau} &= \frac{\mathrm{d}}{\mathrm{d}\tau} \left( \left( \boldsymbol{I}_n + \tau \boldsymbol{L}_* \right)^{-1} \right) \boldsymbol{p}^{(0)} \\
&= -\left( \boldsymbol{I}_n + \tau \boldsymbol{L}_* \right)^{-1} \boldsymbol{L}_* \left( \boldsymbol{I}_n + \tau \boldsymbol{L}_* \right)^{-1} \boldsymbol{p}^{(0)} \\
&= -\left( \boldsymbol{I}_n + \tau \boldsymbol{L}_* \right)^{-1} \boldsymbol{L}_* \boldsymbol{p}(\tau).
\end{aligned}$$

Thus, the differential equation is verified. $\qquad\square$

The differential equation in Equation 29 offers insight into the similarities and differences between the dynamics of the geometric and Poisson distributions. Both equations contain the term $\boldsymbol{L}_*\boldsymbol{p}(\tau)$, implying that any stationary distribution under Poisson dynamics is also stationary for the geometric case. However, the presence of the prefactor $(\boldsymbol{I}_n + \tau\boldsymbol{L}_*)^{-1}$ in Equation 29 suggests that this stationary distribution may not be reached under geometric dynamics.

To understand this, let $\lambda$ be an eigenvalue of $\boldsymbol{L}_*$. Then $\frac{1}{1+\tau\lambda}$ is an eigenvalue of $(\boldsymbol{I}_n + \tau\boldsymbol{L}_*)^{-1}$. For any $\lambda > 0$, we see that $\frac{1}{1+\tau\lambda} \to 0$ as $\tau \to \infty$. Consequently, many eigenvectors will yield near-zero gradients as $\tau$ increases, effectively stalling the dynamics.

As a result, $\boldsymbol{p}(\tau)$ is likely to become "stuck" before converging to the non-informative stationary distribution that characterizes Poisson dynamics. This phenomenon is also observed empirically in Figures 5 and 7, where both dispersal entropy and graph alignment accuracy exhibit this same saturation effect.

### C.3 Relation between the geometric and the uniform distribution

An interesting asymptotic behavior of the geometric distribution is that its $K$-truncation approaches the uniform distribution as $\tau \to \infty$, or equivalently, as $\alpha \to 1$.

To demonstrate this, we begin with the expression for the $K$-truncated reachability of the geometric distribution using the $\alpha$ parameterization:

$$\boldsymbol{R}^{(K)}(\boldsymbol{A}_*; \text{Geom}(\alpha)) = \frac{(1-\alpha)}{Z(\alpha, K)} \sum_{k=0}^{\infty} \alpha^k \boldsymbol{A}_*^k, \tag{30}$$

where $Z(\alpha, K)$ is the normalization factor required due to the truncation.

Next, we reparameterize using $\varepsilon = 1 - \alpha$, yielding:

$$\boldsymbol{R}^{(K)}(\boldsymbol{A}_*; \varepsilon) = \frac{\varepsilon}{Z(\varepsilon, K)} \sum_{k=0}^{K} (1-\varepsilon)^k \boldsymbol{A}_*^k. \tag{31}$$

We can compute the normalization factor $Z(\varepsilon, K)$ as:

$$Z(\varepsilon, K) = \sum_{k=0}^{K} \varepsilon(1-\varepsilon)^k = \varepsilon\frac{1 - (1-\varepsilon)^{K+1}}{1 - (1-\varepsilon)} = 1 - (1-\varepsilon)^{K+1}.$$

As $\varepsilon \to 0$, we can use the approximation $(1-\varepsilon)^k = 1 - k\varepsilon + \mathcal{O}(\varepsilon^2)$ for both $Z(\varepsilon, K)$ and the summands in Equation 31 to show that $\boldsymbol{R}^{(K)}(\boldsymbol{A}_*; \varepsilon)$ approaches the reachability matrix under the uniform distribution:

$$\lim_{\varepsilon \to 0} \boldsymbol{R}^{(K)}(\boldsymbol{A}_*; \varepsilon) = \lim_{\varepsilon \to 0} \frac{\varepsilon}{(K+1)\varepsilon} \sum_{k=0}^{K} (1 - k\varepsilon)\boldsymbol{A}_*^k$$

$$= \lim_{\varepsilon \to 0} \frac{1}{K+1} \sum_{k=0}^{K} \boldsymbol{A}_*^k - \frac{\varepsilon}{K+1} \sum_{k=0}^{K} k\boldsymbol{A}_*^k$$

$$= \frac{1}{K+1} \sum_{k=0}^{K} \boldsymbol{A}_*^k.$$

### C.4 Relation between the binomial and the Poisson distribution

The final asymptotic behavior we highlight is that the reachability under the binomial walk length distribution, $\boldsymbol{R}^{(K)}(\boldsymbol{A}_*; \text{Binom}(\tau))$, approaches that of the Poisson distribution $\boldsymbol{R}(\boldsymbol{A}_*; \text{Pois}(\tau))$ as $K \to \infty$.

We begin by expressing $\boldsymbol{R}^{(K)}(\boldsymbol{A}_*; \mathrm{Binom}(\tau))$ as a matrix function using the Binomial Theorem:

$$
\begin{aligned}
\boldsymbol{R}^{(K)}(\boldsymbol{A}_*; \mathrm{Binom}(\tau)) &= \sum_{k=0}^{K} \binom{K}{k} \left(1 - \frac{\tau}{K}\right)^{K-k} \left(\frac{\tau}{K}\right)^k \boldsymbol{A}_*^k \\
&= \left(\left(1 - \frac{\tau}{K}\right) \boldsymbol{I}_n + \frac{\tau}{K} \boldsymbol{A}_*\right)^K \\
&= \left(\boldsymbol{I}_n - \frac{\tau}{K} \boldsymbol{L}_*\right)^K.
\end{aligned}
$$

Next, we perform the substitution $K_\tau = \frac{K}{\tau}$, and as we take the limit $K_\tau \to \infty$, we obtain:

$$
\lim_{K_\tau \to \infty} \left(\boldsymbol{I}_n - \frac{1}{K_\tau} \boldsymbol{L}_*\right)^{\tau K_\tau} = e^{-\tau \boldsymbol{L}_*} = \boldsymbol{R}(\boldsymbol{A}_*; \mathrm{Pois}(\tau)). \tag{32}
$$

Here, we recover the matrix exponential using its limit definition (Hall, 2015), which, together with Equation 24, proves the asymptotic equality to the Poisson-based reachability.

### C.5 Relation between the binomial distribution and self-loops

Graph Convolutional Networks (GCNs) (Kipf & Welling, 2017) popularized the common pre-processing step of adding self-loops to each node, a practice that has since been widely adopted in self-supervised GNN methods (Wu et al., 2019; Zhang et al., 2021a; Thakoor et al., 2022). As discussed in Section 2.2, there is a close connection between the binomial walk length distribution and graphs with added self-loops, which we explore in this section.

This connection becomes evident when we consider the random surfer interpretation of the binomial distribution. In this view, a random walker makes a total of $K$ decisions. At each step, the walker moves to a neighbouring node with probability $\alpha$, or remains at its current node (i.e., performs a self-loop) with probability $1 - \alpha$. The probability of taking exactly $k$ steps to neighbouring nodes (and $K - k$ self-loops) then follows the binomial distribution:

$$
\mathrm{P}_{\mathtt{w}}(k; \alpha) = \binom{K}{k} (1 - \alpha)^{K-k} \alpha^k.
$$

The corresponding reachability matrix is given by:

$$
\boldsymbol{R}^{(K)}(\boldsymbol{A}_*; \mathrm{Binom}(\alpha)) = \sum_{k=0}^{K} \binom{K}{k} (1 - \alpha)^{K-k} \alpha^k \boldsymbol{A}_*^k. \tag{33}
$$

We can rewrite this expression using the Binomial Theorem and the fact that the diagonal matrix $(1 - \alpha) \boldsymbol{I}_n$ commutes with any matrix under multiplication:

$$
\begin{aligned}
\boldsymbol{R}^{(K)}(\boldsymbol{A}_*; \mathrm{Binom}(\alpha)) &= \sum_{k=0}^{K} \binom{K}{k} (1 - \alpha)^{K-k} \alpha^k \boldsymbol{A}_*^k \\
&= \sum_{k=0}^{K} \binom{K}{k} (1 - \alpha)^{K-k} \boldsymbol{I}_n^{K-k} \alpha^k \boldsymbol{A}_*^k \\
&= \sum_{k=0}^{K} \binom{K}{k} ((1 - \alpha) \boldsymbol{I}_n)^{K-k} (\alpha \boldsymbol{A}_*)^k \\
&= ((1 - \alpha) \boldsymbol{I}_n + \alpha \boldsymbol{A}_*)^K \\
&= \boldsymbol{A}_{*+}^K.
\end{aligned}
$$

Here, we define $\boldsymbol{A}_{*+} = (1-\alpha)\boldsymbol{I}_n + \alpha\boldsymbol{A}_*$ as a self-loop enhanced normalized adjacency matrix. We see that the binomial reachability matrix is simply the $K$th power of $\boldsymbol{A}_{*+}$.

This matrix consists of two parts: a diagonal component $(1-\alpha)\boldsymbol{I}_n$, representing the probability of staying at the current node (i.e., self-loops), and a weighted adjacency component $\alpha\boldsymbol{A}_*$, representing the transition probabilities to neighbouring nodes.

A very similar construction arises in message-passing models that explicitly add self-loops. In what follows, we formalize this for random walks using the outgoing edge transition matrix $\boldsymbol{A}_\mathtt{O}$. The analysis extends directly to the incoming edge matrix $\boldsymbol{A}_\mathtt{I}$ and the undirected matrix $\boldsymbol{A}_\mathtt{U}$.

First, let $\boldsymbol{S} = \boldsymbol{A} + \boldsymbol{I}_n$ be the adjacency matrix with added self-loops, and let $\boldsymbol{D}_{\mathtt{O}+} = \boldsymbol{D}_\mathtt{O} + \boldsymbol{I}_n$ denote the corresponding out-degree matrix. Then, the normalized transition matrix for the self-loop-enhanced graph is given by $\boldsymbol{S}_\mathtt{O} = \boldsymbol{S}\boldsymbol{D}_{\mathtt{O}+}^{-1}$.

We decompose $\boldsymbol{S}_\mathtt{O}$ into two components: $\boldsymbol{S}_\mathtt{O} = \boldsymbol{P}_\circlearrowleft + \boldsymbol{P}_\rightarrow$, where $\boldsymbol{P}_\circlearrowleft = \mathrm{diag}(\boldsymbol{S}_\mathtt{O})$ contains the self-loop (stay-in-place) probabilities, and $\boldsymbol{P}_\rightarrow$ contains the off-diagonal transition probabilities to neighbouring nodes.

This structure closely mirrors that of $\boldsymbol{A}_{*+} = (1-\alpha)\boldsymbol{I}_n + \alpha\boldsymbol{A}_*$. In both cases, the stay-in-place behavior is captured by a diagonal matrix—$\boldsymbol{P}_\circlearrowleft$ for $\boldsymbol{S}_\mathtt{O}$ and $(1-\alpha)\boldsymbol{I}_n$ for $\boldsymbol{A}_{*+}$—while the remaining transitions are modelled via $\boldsymbol{P}_\rightarrow$ and $\alpha\boldsymbol{A}_*$, respectively.

Consequently, the $K$th power of $\boldsymbol{S}_\mathtt{O}$ produces a matrix polynomial with binomial coefficients, analogous to the binomial reachability matrix in Equation 33. This observation aligns with practice: in a $K$-step linear GCN, node embeddings are computed as $\boldsymbol{Z} = (\boldsymbol{S}_\mathtt{O}^K)^\intercal \boldsymbol{X}$ (Wu et al., 2019), which matches the message-passing form of the binomial reachability matrix $\boldsymbol{R}^{(K)}(\boldsymbol{A}_*; \mathrm{Binom}(\alpha)) = \boldsymbol{A}_{*+}^K$ described in Section 2.3.2.

Thus, both approaches share the same underlying random surfer interpretation and exhibit similar mathematical structures. However, there is one key difference: in the binomial reachability model, the stay-in-place probability is uniform across all nodes, controlled by the scalar $(1-\alpha)$. In contrast, for GCNs with added self-loops, the stay-in-place probability varies by node and is given by $\boldsymbol{P}_\circlearrowleft$, where $P_{\circlearrowleft i,i} = \frac{1}{1+\deg_\mathtt{O}(i)}$. This means that nodes with lower out-degree have higher probability of remaining in place.

Whether this degree-dependent behavior improves embedding quality is ultimately an empirical question and may depend on the characteristics of the graph and the downstream task.

# D  Additional experiment setup information

## D.1  Single- and multi-scale ReachNEs embedding distributions

This section provides further details related to the experimental setup described in Section 6. Figure 11 shows the random walk length distributions used in the ReachNEs experiments, as listed in Table 3.

The first plot, Figure 11a, displays the two single-scale distributions, referred to as **Geom** and **Pois**. The Geom distribution corresponds to $\mathrm{Geom}(k; \tau = 1)$, emphasizing the node itself ($k = 0$) and short-range walks (1–2 steps). In contrast, Pois uses $\mathrm{Pois}(k; \tau = 2)$, which down-weights self-loops and focuses on medium-range neighbourhoods (2–4 steps).

The **Geom-$\mathcal{U}$** setting creates two-scale embeddings by combining two distributions designed to capture both short- and medium-range structure. The first is $\mathrm{Geom}(k; \tau = 1)$ (same as in Geom), while the second is the shifted uniform distribution $\mathcal{U}(k - 1; \tau = 2)$, which excludes the $k = 0$ term and concentrates probability mass over $k \in [1, 5]$.

The **Binom-3** and **Geom-4** settings are multi-scale distributions designed to capture short-, medium-, and long-range reachability. Binom-3 is inspired by Gaussian mixture models (Murphy, 2012, Ch. 11.2.1) and combines three binomial components: $\mathrm{Binom}(k; \tau = 1)$ centred at $k = 0$, $\mathrm{Binom}(k - 2; \tau = 1)$ centred at $k = 2$, and $\mathrm{Binom}(k - 5; \tau = 1)$ centred at $k = 5$. To account for the increasing range, we increase the parameter $\tau$ for longer walks, thereby widening the support of the distribution.

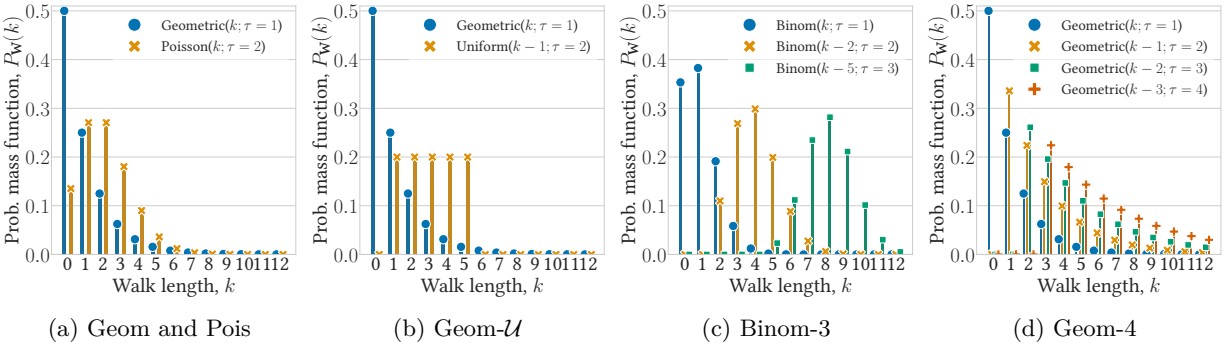

(a) Geom and Pois     (b) Geom-$\mathcal{U}$     (c) Binom-3     (d) Geom-4

Figure 11: Plots of the walk length distributions $P_w$ listed in Table 3. (a) show the two distributions used to create single-scale ReachNEs embeddings, while (b), (c) and (d) show the distributions used for multi-scale embeddings, with 2, 3 and 4 components respectively.

In contrast, Geom-4 uses four geometric components with progressively increasing mode and tail width: $\text{Geom}(k; \tau = 1)$, $\text{Geom}(k - 1; \tau = 2)$, $\text{Geom}(k - 2; \tau = 3)$, and $\text{Geom}(k - 3; \tau = 4)$. Unlike Binom-3, these distributions are not centred on long-range steps but exhibit heavy tails, meaning they still provide smoothing over distant nodes. The advantage of Geom-4 is that it places distinct modes over steps $k \in \{0, 1, 2, 3\}$, a range where most node-distinguishing information typically resides. This focus on short- and mid-range steps aligns with the goal of preserving local structure while retaining some long-range awareness.

## D.2 Hyperparameters for proximity embedding comparison

Table 13 shows the hyperparameter search grids for each digraph proximity method compared in Section 6.4. For all models, we tune the number of embedding dimensions using $p \in [64, 1024]$. Note that for HOPE, we report halved values, as the implementation internally doubles the embedding size.

For DirSwitch-$r$, we tune both the number of directed steps $r$ and the choice of walk length distributions $P_w$. For the other baselines, we tune their respective hyperparameters that influence proximity scale. These include:

- the $\beta$ parameter for HOPE,

- the number of walk steps and jump factor for APP,

- the number of walk steps for NERD,

- the number of layers for BLADE.

We also tune selected learning-related hyperparameters: the number of sampled walks for NERD, the loss trade-off parameter $\lambda$ for DGGAN, and the learning rate and number of epochs for BLADE.

With this setup, the total grid sizes are approximately comparable across methods. DirSwitch uses 75 different hyperparameter combinations. Among the baselines, APP and BLADE have the largest grids, with 120 and 300 combinations respectively. DGGAN has the smallest grid (45 combinations), as it is significantly more computationally expensive to train, making extensive tuning infeasible.

The best-performing hyperparameter settings for each model and dataset, based on cross-validation accuracy, are listed in Table 14. For DirSwitch-$r$, we observe that multi-scale embeddings with three or four components consistently yield the best results, aligning with prior findings on the benefits of multi-scale embedding strategies (Rozemberczki et al., 2021).

Table 13: Hyperparameter grids used for the comparison of proximity embedding models in Section 6.4.

| MODEL | HYPERPARAMETER | VALUES |
|---|---|---|
| DIRSWITCH-$r$ | EMB. DIM. $p$ | 64, 128, 256, 512, 1024 |
| | $r$ | 1, 2, 3 |
| | $P_w$ | GEOM, POIS, GEOM-$\mathcal{U}$, BINOM-3, GEOM-4 |
| HOPE | $\beta$ | 0.001, 0.01, 0.1, 0.2, 0.35, 0.5, 0.65, 0.8, 0.9, 1.0 |
| | EMB. DIM. $p$ | 32, 64, 128, 256, 512 |
| APP | EMB. DIM. $p$ | 64, 128, 256, 512, 1024 |
| | JUMP FACTOR | 0.1, 0.25, 0.5, 0.75 |
| | WALK STEPS | 2, 3, 4, 5, 6, 7 |
| NERD | EMB. DIM. $p$ | 64, 128, 256, 512, 1024 |
| | NUM. SAMPLES | 1, 2, 3 |
| | WALK STEPS | 2, 3, 4, 5, 6 |
| DGGAN | EMB. DIM. $p$ | 64, 128, 256, 512, 1024 |
| | $\lambda$ | 5E-06, 1E-05, 5E-05 |
| | LEARNING RATE | 5E-05, 0.0001, 0.0005 |
| BLADE | EMB. DIM. $p$ | 64, 128, 256, 512, 1024 |
| | LEARNING RATE | 0.0001, 0.001, 0.01 |
| | NUM. EPOCHS | 10, 30, 50, 100 |
| | NUM. LAYERS | 2, 3, 4, 5, 6 |

Table 14: The best hyperparameter values for each proximity embedding model and dataset.

| MODEL | | FLY LARVA | | EU-EMAIL | | POLBLOGS | | COCITE | | PUBMED | | CORA (SUBELJ) | |
|---|---|---|---|---|---|---|---|---|---|---|---|---|---|
| DIRSWITCH | EMB. DIM. $p$ | 256 | EMB. DIM. $p$ | 1024 | EMB. DIM. $p$ | 128 | EMB. DIM. $p$ | 1024 | EMB. DIM. $p$ | 1024 | EMB. DIM. $p$ | 1024 |
| | $r$ | 3 | $r$ | 2 | $r$ | 1 | $r$ | 1 | $r$ | 1 | $r$ | 1 |
| | $P_w$ | GEOM-4 | $P_w$ | BINOM-3 | $P_w$ | BINOM-3 | $P_w$ | BINOM-3 | $P_w$ | BINOM-3 | $P_w$ | BINOM-3 |
| HOPE | $\beta$ | 1 | $\beta$ | 1 | $\beta$ | 0.9 | $\beta$ | 0.9 | $\beta$ | 0.9 | $\beta$ | 1 |
| | EMB. DIM. $p$ | 512 | EMB. DIM. $p$ | 128 | EMB. DIM. $p$ | 32 | EMB. DIM. $p$ | 512 | EMB. DIM. $p$ | 512 | EMB. DIM. $p$ | 512 |
| APP | EMB. DIM. $p$ | 1024 | EMB. DIM. $p$ | 512 | EMB. DIM. $p$ | 64 | EMB. DIM. $p$ | 512 | EMB. DIM. $p$ | 1024 | EMB. DIM. $p$ | 1024 |
| | JUMP FACTOR | 0.75 | JUMP FACTOR | 0.75 | JUMP FACTOR | 0.5 | JUMP FACTOR | 0.25 | JUMP FACTOR | 0.75 | JUMP FACTOR | 0.75 |
| | WALK STEPS | 5 | WALK STEPS | 4 | WALK STEPS | 5 | WALK STEPS | 5 | WALK STEPS | 5 | WALK STEPS | 7 |
| NERD | EMB. DIM. $p$ | 256 | EMB. DIM. $p$ | 1024 | EMB. DIM. $p$ | 128 | EMB. DIM. $p$ | 256 | EMB. DIM. $p$ | 512 | EMB. DIM. $p$ | 256 |
| | NUM. SAMPLES | 3 | NUM. SAMPLES | 3 | NUM. SAMPLES | 2 | NUM. SAMPLES | 3 | NUM. SAMPLES | 3 | NUM. SAMPLES | 3 |
| | WALK STEPS | 5 | WALK STEPS | 5 | WALK STEPS | 5 | WALK STEPS | 6 | WALK STEPS | 6 | WALK STEPS | 6 |
| DGGAN | EMB. DIM. $p$ | 256 | EMB. DIM. $p$ | 512 | EMB. DIM. $p$ | 128 | EMB. DIM. $p$ | 128 | EMB. DIM. $p$ | 128 | EMB. DIM. $p$ | 256 |
| | $\lambda$ | 5E-06 | $\lambda$ | 1E-05 | $\lambda$ | 5E-05 | $\lambda$ | 1E-05 | $\lambda$ | 5E-05 | $\lambda$ | 5E-06 |
| | LEARNING RATE | 0.0005 | LEARNING RATE | 0.0005 | LEARNING RATE | 5E-05 | LEARNING RATE | 5E-05 | LEARNING RATE | 0.0001 | LEARNING RATE | 5E-05 |
| BLADE | EMB. DIM. $p$ | 1024 | EMB. DIM. $p$ | 1024 | EMB. DIM. $p$ | 512 | EMB. DIM. $p$ | 1024 | EMB. DIM. $p$ | 1024 | EMB. DIM. $p$ | 1024 |
| | LEARNING RATE | 0.0001 | LEARNING RATE | 0.0001 | LEARNING RATE | 0.001 | LEARNING RATE | 0.0001 | LEARNING RATE | 0.0001 | LEARNING RATE | 0.0001 |
| | NUM. EPOCHS | 10 | NUM. EPOCHS | 10 | NUM. EPOCHS | 10 | NUM. EPOCHS | 10 | NUM. EPOCHS | 10 | NUM. EPOCHS | 30 |
| | NUM. LAYERS | 3 | NUM. LAYERS | 2 | NUM. LAYERS | 6 | NUM. LAYERS | 4 | NUM. LAYERS | 6 | NUM. LAYERS | 2 |

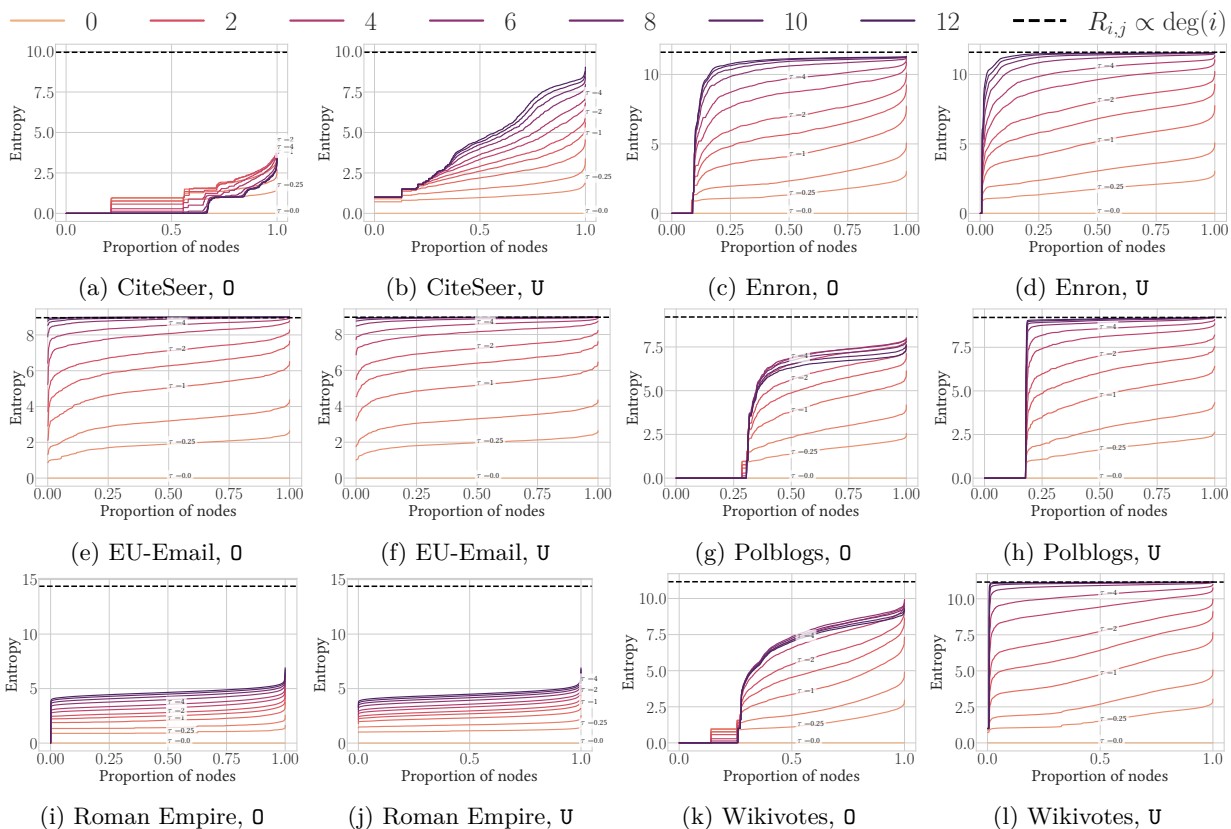

Figure 12: Reachability under $\mathrm{P_w} = \mathrm{Pois}(\tau)$, using $\boldsymbol{A_0}$ and $\boldsymbol{A_U}$. The x-axes correspond to nodes sorted by their entropy value. The colours represent various values of $\tau$. The black dashed line indicates the entropy of a reachability distribution proportional to the node degrees, which corresponds to the limiting and uninformative distribution as $\tau \to \infty$. See Appendix C.1 for further details.

## E  Additional analysis of local sinks and dispersal

This section provides additional analysis of local sinks and reachability dispersal for graph datasets not included in the main paper due to space constraints. As before, we use *reachability entropy* to quantify dispersal. Recall that the reachability entropy of node $j$ is defined as:

$$H(j; \boldsymbol{R}) = -\sum_{i=1}^{n} R_{i,j} \log_2 R_{i,j}, \tag{34}$$

where $R_{i,j}$ is the $(i,j)$-th element of the reachability matrix $\boldsymbol{R}(\mathrm{P_w}, \boldsymbol{\sigma})$, computed under a walk length distribution $\mathrm{P_w}$ and edge direction specifier $\boldsymbol{\sigma}$.

Figure 12 complements Figure 5 by presenting results for six additional datasets, using the Poisson walk length distribution $\mathrm{P_w} = \mathrm{Pois}(\tau)$. Each coloured curve corresponds to a different value of $\tau$. The figure focuses on comparing the dispersal effects of default directed edges ($\boldsymbol{\sigma} = \mathtt{O}$) versus undirected edges ($\boldsymbol{\sigma} = \mathtt{U}$).

Consistent with the examples in Section 3.1, we observe substantial entropy gains when using undirected edges on sparse and moderately dense graphs. For example, the sparse CiteSeer graph shows a pronounced increase in entropy, while medium-density graphs such as Enron, Polblogs, and Wikivotes also exhibit clear improvements. These gains become more pronounced at larger $\tau$ values, as reachability under directed edges increasingly concentrates in local sinks.

In contrast, the dense EU-Email graph, with an average node degree of 25, shows only minor differences between the directed and undirected cases. While small increases in entropy are still visible (cf. Figure 12e and 12f), the graph's high density mitigates sink formation and enables more uniform information propagation.

Figure 13 extends Figure 6 by presenting results for eleven additional datasets. We again use $\mathrm{P}_{\mathtt{w}} = \mathrm{Pois}(\tau = 2)$ to model short-range walks and $\mathrm{P}_{\mathtt{w}} = \mathcal{U}(\tau = 5)$ for longer-range behaviour.

The figure compares DirSwitch edge direction specifiers with those of MultiDir. The results closely follow the trends observed in Figure 5, with DirSwitch consistently achieving higher entropy than MultiDir—especially under the long-range distribution $\mathrm{P}_{\mathtt{w}} = \mathcal{U}(\tau)$. As before, the differences are most pronounced on sparse graphs, including Arxiv, CoCite, Pubmed, and Snap Patents.

The Roman Empire graph constitutes a counterexample to the trends described above. As shown in Figure 12i and 12j, there is virtually no difference in entropy between $\boldsymbol{\sigma} = \mathtt{O}$ and $\boldsymbol{\sigma} = \mathtt{U}$, despite the graph's sparsity. In fact, close inspection reveals that entropy is slightly *lower* for the undirected case, particularly for large values of $\tau$. This effect is even more pronounced in Figure 13j, where $\boldsymbol{\sigma} = \mathtt{O}$ yields the highest entropy under the uniform walk length distribution. This is the only dataset where such behaviour is observed.

To understand this counterintuitive result, we must examine the specific structural properties of the Roman Empire graph. As shown in Table 2, this graph stands out with an unusually long average path length—around 2400 steps. The underlying reason is its chain-like topology: the graph primarily consists of a long linear sequence of nodes, where each node typically has an in-degree and out-degree of one. Occasionally, nodes branch off from the main chain and later reconnect.

Figure 14 visualizes a subgraph of the Roman Empire graph, clearly illustrating this structure. It also highlights why entropy may decrease in the undirected case. The node colours represent reachability values from the same starting node under $\boldsymbol{\sigma} = \mathtt{O}$ on the left, and $\boldsymbol{\sigma} = \mathtt{U}$ on the right, both using the uniform walk length distribution. As shown in Figure 14a, reachability in the directed case is spread more uniformly along the chain. In contrast, Figure 14b shows that the undirected case concentrates reachability around the starting node.

This is a consequence of the chain structure: in the directed case, random walks proceed almost exclusively forward along the chain. In the undirected case, however, each step introduces a 33–50% chance of stepping backward. This backward drift increases the likelihood of returning to or remaining near the starting node, reducing entropy.

## F   Additional graph alignment results

Figure 15 presents graph alignment results for three additional datasets, complementing the four datasets shown in Figure 7. As in the main paper, the y-axes indicate graph alignment accuracy, while the x-axes show values of $\tau$. The top row corresponds to $\mathrm{P}_{\mathtt{w}} = \mathrm{Geom}(\tau)$, and the bottom row to $\mathrm{P}_{\mathtt{w}} = \mathrm{Pois}(\tau)$.

The results in Figure 15 are consistent with the findings from the main paper. For both DirSwitch-$r$ and MultiDir-$r$, alignment accuracy increases with $r$, reflecting the benefit of representing a larger number of directed neighbourhoods. In contrast, models using $\boldsymbol{\sigma} = \mathtt{O}$ and $\boldsymbol{\sigma} = \mathtt{U}$ consistently achieve the lowest accuracies, as they lack the capacity to capture multiple directed neighbourhoods. DirSwitch-$r$ generally outperforms MultiDir-$r$, highlighting that mitigating the effects of local sinks contributes to improved alignment accuracy.

We again observe the distinct behaviours of the Poisson and geometric walk length distributions: increasing $\tau$ tends to degrade accuracy under $\mathrm{Pois}(\tau)$, whereas accuracy plateaus under $\mathrm{Geom}(\tau)$, as previously discussed.

A notable exception is the Pubmed dataset, where DirSwitch-$r$ with $\mathrm{P}_{\mathtt{w}} = \mathrm{Pois}(\tau)$ reaches peak accuracy around $\tau \approx 9$, indicating that a relatively large receptive field is required for optimal performance. This contrasts with the other datasets, which generally attain their highest accuracy for $\tau < 5$.

This behaviour can be attributed to the structure of the Pubmed graph. As shown in Table 2, Pubmed has a very sparse local structure-its median out-degree is 0, and its median in-degree is 1—meaning that the short-range neighbourhoods of many nodes are nearly indistinguishable. Additionally, Pubmed exhibits the longest average path length among the alignment datasets. These factors necessitate a broader receptive field to generate more expressive and distinguishable embeddings.

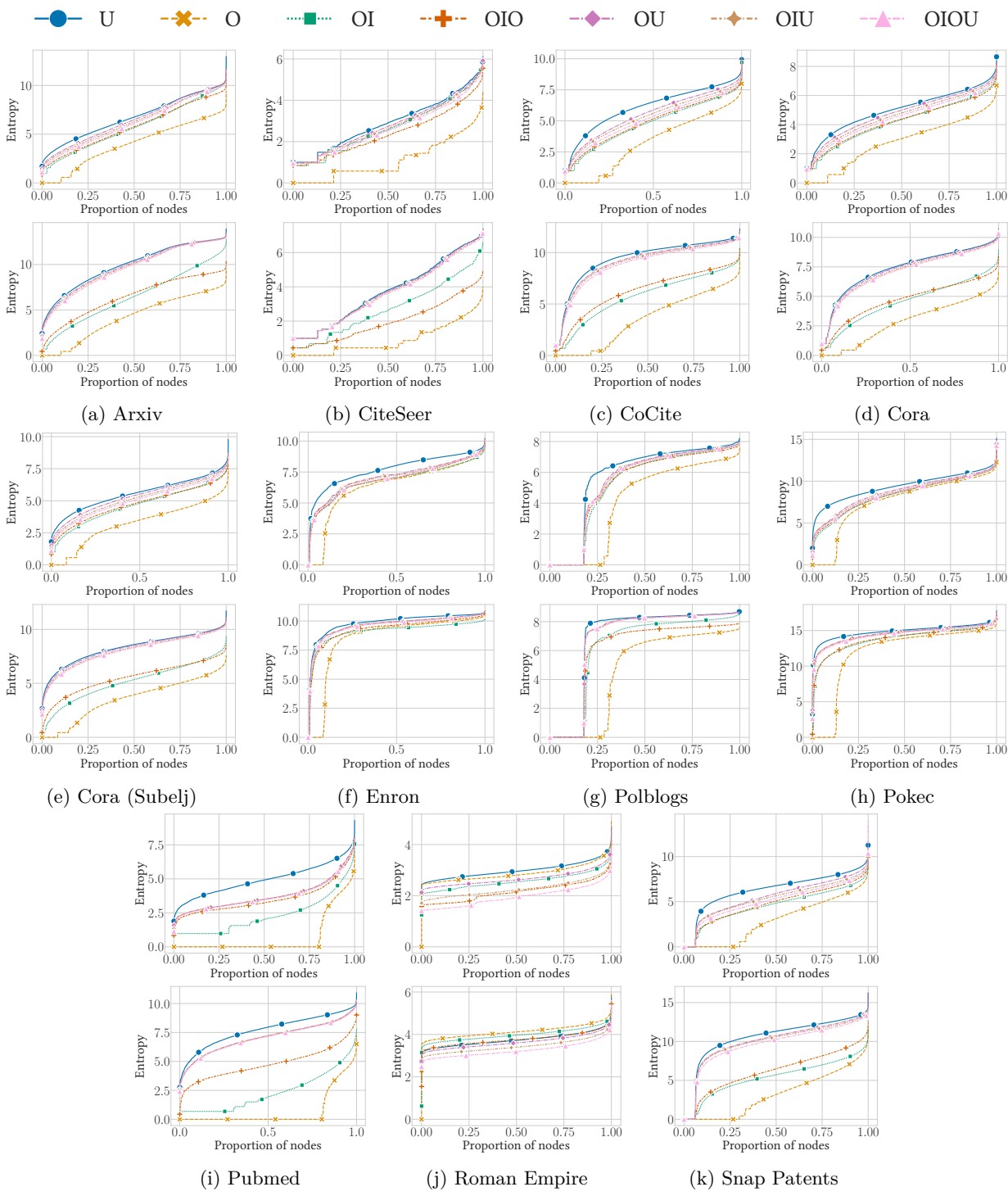

Figure 13: Neighbourhood dispersal evaluation for 11 graphs, measured via reachability entropy, $-\sum_{i=1}^{n} R_{i,j} \log_2 R_{i,j}$, computed for each node and sorted. Each curve corresponds to a different edge direction specifier $\boldsymbol{\sigma}$, with the top row showing results for $P_{\mathtt{w}}(k; \tau) = \mathrm{Pois}(\tau = 2)$ (local dispersal) and the bottom row for $\mathcal{U}(\tau = 5)$ (long-range dispersal). DirSwitch variants (e.g., OU, OIU, OIOU) demonstrate high dispersal, comparable to U, while purely directed specifiers (O, OI, OIO) exhibit lower entropy due to sink effects.

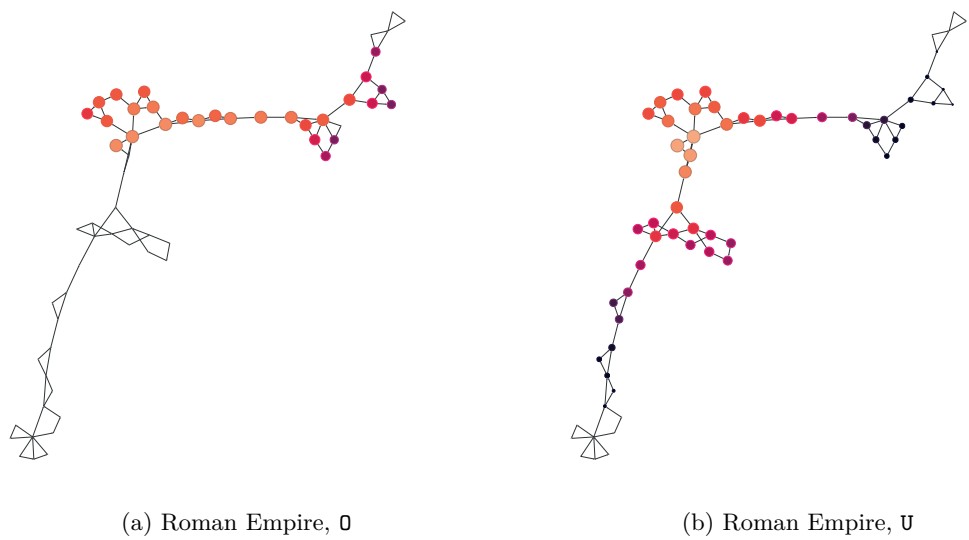

(a) Roman Empire, O
    (b) Roman Empire, U

Figure 14: Both figures show the same subgraph of the Roman Empire graph. The node colours reflect the reachability vector for the same node under the uniform walk length distribution. (a) uses default directed edges ($\boldsymbol{\sigma} = $ O) while (b) uses undirected edge ($\boldsymbol{\sigma} = $ U).

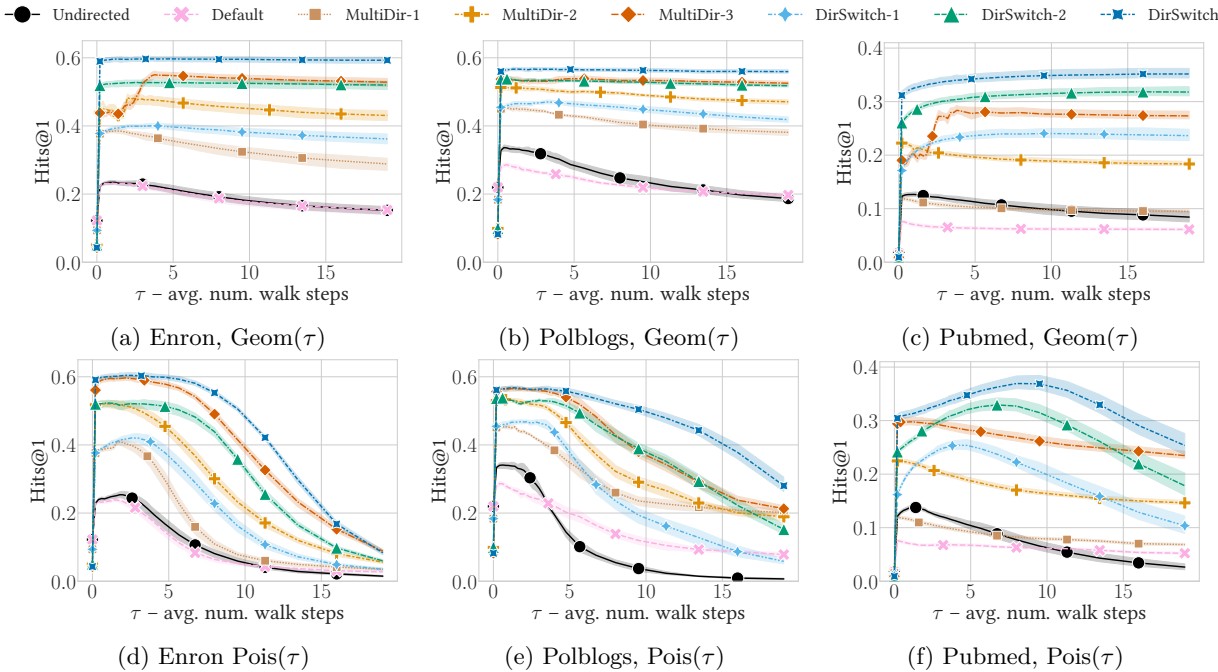

Figure 15: Evaluation of edge direction expressivity for four graphs using the geometric, $\mathrm{Geom}(\tau)$, and Poisson, $\mathrm{Pois}(\tau)$, walk length distributions. The y-axes represent graph alignment accuracy under 15% edge removal, while the x-axes correspond to $\tau$, the average walk length. The curve colours and styles denote different sets of edge direction specifiers, $\boldsymbol{\sigma}$.

# G   Additional DirSwitch embedding quality evaluation

This section provides further analysis and discussion of the results presented in Section 6.3, focusing on the benefits of multi-scale embeddings, the limitations introduced by fixed embedding dimensionality and the presence of 2-step homophiliy in the Pokec dataset.

**Multi-scale embeddings and diminishing returns.**   The results in Tables 4 and 6 offer clear evidence of the benefits of multi-scale embeddings. Across all datasets, we observe at least moderate improvements when transitioning from single-scale to multi-scale walk length distributions, with several datasets exhibiting substantial accuracy gains. For instance, on Cora-ML, the classification accuracy for undirected embeddings increases from 31% to 76%; on Citeseer, from 76% to 89%; and on Cora, from 59% to 69% when switching from `Geom` to `Geom-4`. These results highlight the importance of embedding expressivity in unsupervised learning.

Similar trends are observed for proximity embeddings in Table 6. For example, EU-Email improves from 30% to 72%, and Polblogs from 64% to 85% under the same change in walk length distributions.

However, the benefits of multi-scale embeddings diminish as the number of multi-directional neighbourhoods increases, particularly for MultiDir-$r$ and DirSwitch-$r$. In Cora-ML, for example, DirSwitch-1 improves from 57% to 82% when switching from `Geom` to `Geom-4`, whereas DirSwitch-3 shows a more modest improvement from 79% to 86%.

This trend is even more pronounced in the results for $p = 512$ embedding dimensions (Tables 16 and 18). In these cases, increasing both the number of multi-directional neighbourhoods and the number of scales can lead to accuracy degradation. For example, in Table 16, the accuracy for Cora declines as $r$ increases and when using `Binom-3` or `Geom-4`. A similar pattern is observed for EU-Email in Table 18.

This behaviour arises from the diminishing returns of combining multi-directional and multi-scale embeddings via concatenation under fixed dimensionality constraints. As described in Section 4, the total embedding dimension $p$ must be divided among all combinations of direction specifiers and walk length distributions. When $p$ is fixed, increasing the number of concatenated components reduces the dimensionality available for each, limiting their expressivity. As a result, performance gains plateau at $p = 1024$ and begin to decline at $p = 512$.

While increasing $p$ can help alleviate this issue by allowing more capacity for embedding components, this approach has practical limitations. Larger embeddings require more memory and computational resources, and may introduce challenges such as overfitting or the curse of dimensionality in downstream tasks. Future work is needed to explore more efficient ways to compress and integrate multi-directional and multi-scale information into compact embeddings without sacrificing expressivity.

**Homophily in the Pokec dataset.**   As noted in the main paper, the classification results for the Pokec dataset in Table 4 exhibit behaviour that diverges from other heterophilic datasets. In particular, we observe a notable accuracy increase when using Geom-4, especially for undirected edges: accuracy rises from 62% with Binom-3 to 72% with Geom-4. While Pokec typically follows the heterophilic trend—where undirected embeddings underperform compared to DirSwitch-$r$ and MultiDir-$r$—this pattern is reversed for Geom-4.

To understand this, we need to examine the structural properties of the Pokec dataset. Pokec is an online Slovak social network, with nodes representing users and labels corresponding to reported gender. As reported by Lim et al. (2021), who curated this benchmark, the graph is heterophilic at the 1-step level due to the predominance of heterosexual connections. This is reflected in a low 1-step homophily score of 0.43.

However, the homophily increases significantly at the 2-step level, rising to 0.61. This indicates that while direct connections tend to be between users of different genders, the extended 2-step neighbourhoods are more gender-homogeneous. In such cases, undirected smoothing becomes more effective, especially when the embedding method emphasizes these mid-range neighbourhoods.

This is precisely what the Geom-4 distribution achieves. As seen in Figure 11d, one of its components is $\text{Geom}(k - 2; \tau = 3)$, which emphasizes 2-step neighbourhoods more than the other multi-scale distributions. This focus on 2-step structure explains the performance gain observed with Geom-4 and undirected edges.

Table 15: Further investigation into the Pokec dataset. The walk length distributions are single-scale indicator distributions that place the entire probability mass on a single $k$-value.

(a) $p = 512$

| EDGE DIRECTIONS | $\mathbf{1}_{k=0}$ | $\mathbf{1}_{k=1}$ | $\mathbf{1}_{k=2}$ | $\mathbf{1}_{k=3}$ |
|---|---|---|---|---|
| DEFAULT | $61.4_{\pm0.1}$ | $63.0_{\pm0.1}$ | $63.2_{\pm0.1}$ | $62.5_{\pm0.1}$ |
| UNDIRECTED | $61.4_{\pm0.1}$ | $63.0_{\pm0.1}$ | $69.9_{\pm0.1}$ | $66.2_{\pm0.1}$ |
| MULTIDIR-1 | $61.4_{\pm0.1}$ | $63.4_{\pm0.1}$ | $65.3_{\pm0.1}$ | $63.3_{\pm0.1}$ |
| MULTIDIR-2 | $61.4_{\pm0.1}$ | $63.4_{\pm0.1}$ | $68.0_{\pm0.1}$ | $63.9_{\pm0.1}$ |
| DIRSWITCH-1 | $61.4_{\pm0.1}$ | $63.4_{\pm0.1}$ | $68.0_{\pm0.1}$ | $66.3_{\pm0.1}$ |
| DIRSWITCH-2 | $61.4_{\pm0.1}$ | $63.4_{\pm0.1}$ | $68.0_{\pm0.1}$ | $66.1_{\pm0.1}$ |
| DIRSWITCH-3 | $61.4_{\pm0.1}$ | $60.7_{\pm0.1}$ | $67.1_{\pm0.1}$ | $63.5_{\pm0.1}$ |

(b) $p = 1024$

| EDGE DIRECTIONS | $\mathbf{1}_{k=0}$ | $\mathbf{1}_{k=1}$ | $\mathbf{1}_{k=2}$ | $\mathbf{1}_{k=3}$ |
|---|---|---|---|---|
| DEFAULT | $61.4_{\pm0.1}$ | $63.0_{\pm0.1}$ | $63.1_{\pm0.1}$ | $62.5_{\pm0.1}$ |
| UNDIRECTED | $61.4_{\pm0.1}$ | $63.0_{\pm0.1}$ | $69.9_{\pm0.0}$ | $66.2_{\pm0.1}$ |
| MULTIDIR-1 | $61.4_{\pm0.1}$ | $63.4_{\pm0.1}$ | $65.3_{\pm0.1}$ | $63.3_{\pm0.1}$ |
| MULTIDIR-2 | $61.4_{\pm0.1}$ | $63.4_{\pm0.1}$ | $68.0_{\pm0.1}$ | $63.9_{\pm0.1}$ |
| DIRSWITCH-1 | $61.4_{\pm0.1}$ | $63.4_{\pm0.1}$ | $68.0_{\pm0.1}$ | $66.3_{\pm0.1}$ |
| DIRSWITCH-2 | $61.4_{\pm0.1}$ | $63.4_{\pm0.1}$ | $68.0_{\pm0.1}$ | $66.1_{\pm0.1}$ |
| DIRSWITCH-3 | $61.4_{\pm0.1}$ | $63.4_{\pm0.1}$ | $68.0_{\pm0.1}$ | $65.9_{\pm0.1}$ |

To further validate this interpretation, we conduct an additional experiment using indicator walk length distributions that place all probability mass on a single step length:

$$P_{\mathtt{w}} = \mathbf{1}_{k=\tau} = \begin{cases} 1 & \text{if } k = \tau \\ 0 & \text{otherwise} \end{cases}. \tag{35}$$

This isolates the effect of each specific neighbourhood scale. The results, shown in Table 15, confirm that 2-step neighbourhoods ($\mathbf{1}_{k=2}$) yield the highest accuracy for undirected edges. In contrast, no similar performance gain is observed for directed edges (Default), reinforcing the notion that the combination of undirected smoothing and mid-range reachability is key to improved performance on Pokec.

Beyond explaining this anomaly, the analysis highlights a broader point: homophily and heterophily are not binary properties but exist on a spectrum. The optimal embedding strategy may vary across this spectrum, and models tailored exclusively for one end may perform poorly on datasets lying in the middle. This underscores the utility of DirSwitch, which we have shown to perform consistently well across both homophilic and heterophilic settings.

**Unsupervised community detection.** Our evaluation of DirSwitch in Section 6.3 focused on node classification, which offers a convenient and objective metric: classification accuracy.

In contrast, evaluating unsupervised downstream tasks is more challenging, as these tasks often lack a single "correct" solution. For example, many valid groupings of nodes can exist in a graph, yet only one such grouping is typically labelled for evaluation. Similar issues arise in other unsupervised tasks such as anomaly detection. As a result, these evaluations often require more nuanced analysis.

Nevertheless, since DirSwitch is intended for unsupervised embedding learning, it is important to assess its performance on unsupervised tasks as well. We therefore include a community detection benchmark. For this, we use the same proximity node classification datasets from Section 6.3, treating the node labels as ground-truth communities. We apply $k$-means clustering (Arthur & Vassilvitskii, 2007) to the embeddings, setting the number of clusters equal to the number of labelled communities. Performance is reported using normalized mutual information (NMI), averaged over 10 runs with different random seeds to compute mean and standard deviation.

Tables 20 and 21 present the community detection results for DirSwitch-$r$ with $p = 512$ and $p = 1024$, respectively. The findings mirror those from the node classification experiments: DirSwitch-$r$ generally performs well across datasets, while the sparse citation graphs (CoCite, Pubmed, and Cora (subelj)) tend to benefit from fully undirected edge orientation.

Results on the Polblogs dataset are particularly noteworthy due to their large standard deviations. This reflects the challenge of evaluating unsupervised tasks: we interpret these fluctuations as evidence that multiple valid community structures exist, with $k$-means sometimes recovering the one aligned with the labels, and sometimes converging to an alternative.

Table 16: Node classification accuracy for message-passing ReachNEs with $p = 512$ embedding dimensions. Columns correspond to different datasets and multi-scale walk length distributions, while rows represent various edge direction specifiers. Each entry reports the mean accuracy and standard deviation. Bold blue highlights the highest accuracy in each column, with light blue indicating results within one standard deviation of the best. Similarly, bold orange denotes the lowest accuracy, and light orange represents values within one standard deviation of the worst.

| Edge directions | Arxiv | | | | | Arxiv Year | | | | |
|---|---|---|---|---|---|---|---|---|---|---|
| | Geom | Pois | Geom-$\mathcal{U}$ | Binom-3 | Geom-4 | Geom | Pois | Geom-$\mathcal{U}$ | Binom-3 | Geom-4 |
| Default | **55.0±0.2** | **42.4±0.4** | **55.4±0.2** | **54.2±0.2** | **55.4±0.2** | 36.5±0.3 | 35.2±0.5 | **36.7±0.2** | 37.8±0.3 | **38.2±0.3** |
| Undirected | **61.6±0.2** | 64.9±0.2 | 67.0±0.2 | **69.8±0.2** | **69.1±0.2** | **36.5±0.2** | 36.5±0.2 | 37.1±0.2 | **37.6±0.3** | 38.3±0.2 |
| MultiDir-1 | 59.8±0.3 | 59.4±0.3 | 60.1±0.3 | 60.2±0.3 | 60.0±0.3 | 37.0±0.3 | **35.1±0.7** | 38.1±0.2 | 39.1±0.2 | 39.4±0.3 |
| MultiDir-2 | 61.2±0.3 | 65.3±0.3 | 65.1±0.3 | 64.4±0.3 | 64.6±0.3 | 38.6±0.3 | 38.0±0.3 | 40.5±0.3 | 41.1±0.2 | 41.0±0.3 |
| MultiDir-3 | 60.9±0.3 | 65.5±0.3 | 65.3±0.3 | 64.7±0.3 | 64.1±0.2 | **39.7±0.3** | 40.2±0.3 | **41.4±0.3** | **42.1±0.3** | 41.3±0.2 |
| DirSwitch-1 | 61.5±0.3 | **65.9±0.2** | **67.3±0.3** | 69.5±0.2 | 68.6±0.3 | 38.2±0.2 | 37.8±0.2 | 39.9±0.3 | 39.8±0.3 | 41.2±0.3 |
| DirSwitch-2 | 61.5±0.3 | 65.7±0.2 | 66.7±0.3 | 68.9±0.3 | 67.8±0.3 | 39.6±0.3 | 39.5±0.3 | 40.9±0.3 | 41.1±0.3 | 41.5±0.3 |
| DirSwitch-3 | 60.8±0.3 | 65.6±0.3 | 65.9±0.3 | 67.7±0.3 | 66.3±0.3 | 39.6±0.3 | **40.5±0.3** | 41.4±0.2 | 42.1±0.2 | **42.2±0.3** |

| Edge directions | Cora-ML | | | | | Citeseer | | | | | Cora | | | | |
|---|---|---|---|---|---|---|---|---|---|---|---|---|---|---|---|
| | Geom | Pois | Geom-$\mathcal{U}$ | Binom-3 | Geom-4 | Geom | Pois | Geom-$\mathcal{U}$ | Binom-3 | Geom-4 | Geom | Pois | Geom-$\mathcal{U}$ | Binom-3 | Geom-4 |
| Default | **50.5±3.1** | **51.9±2.1** | **64.2±2.6** | **70.5±1.3** | **73.5±1.7** | **70.7±1.8** | **58.4±2.2** | **74.1±2.6** | **75.8±1.9** | **78.2±1.6** | **57.4±0.7** | **54.1±0.8** | **60.7±0.8** | **60.9±0.8** | 61.1±0.9 |
| Undirected | 60.4±2.5 | 68.1±2.2 | 73.8±1.8 | 82.2±1.3 | 83.7±1.4 | 80.6±1.4 | 86.6±1.5 | 87.0±1.5 | **93.9±0.9** | **92.8±1.1** | 64.9±0.7 | 68.1±0.7 | **69.2±0.6** | **70.1±0.9** | **70.0±0.8** |
| MultiDir-1 | 68.5±2.4 | 67.5±2.4 | 75.2±2.6 | 77.2±2.2 | 79.4±2.3 | 76.1±1.9 | 75.4±2.1 | 80.7±1.9 | 81.9±1.5 | 82.8±1.5 | 65.5±0.7 | 64.8±0.8 | 65.3±1.1 | 63.5±1.0 | **62.7±0.9** |
| MultiDir-2 | 79.1±1.7 | 78.8±1.6 | 82.7±1.6 | 83.1±1.4 | 84.3±1.2 | **83.9±1.7** | 85.1±1.7 | 86.0±1.4 | 85.8±1.6 | 86.6±1.2 | **66.7±1.1** | 68.1±0.9 | 66.2±0.9 | 64.5±1.0 | 63.6±1.1 |
| MultiDir-3 | **83.1±1.5** | **83.6±1.7** | 85.2±1.5 | 84.8±1.5 | 84.7±1.6 | 85.2±1.5 | 88.0±1.4 | 87.0±1.3 | 87.1±1.2 | 87.2±0.9 | 65.0±0.8 | 67.2±1.0 | 64.4±1.0 | **62.0±1.1** | **61.1±0.9** |
| DirSwitch-1 | 72.9±1.9 | 73.7±2.1 | 81.3±1.9 | 85.0±1.5 | 85.2±1.2 | 82.8±1.0 | 88.4±1.2 | **89.2±1.3** | 91.5±0.9 | 90.7±1.2 | **67.5±0.9** | **69.3±0.7** | 69.1±0.9 | 69.2±1.0 | 68.4±0.7 |
| DirSwitch-2 | 79.5±2.0 | 80.8±1.9 | 84.1±1.6 | 85.7±1.9 | 85.7±1.5 | 85.3±1.1 | **88.8±0.9** | 88.4±1.1 | 90.0±1.2 | 88.8±1.3 | 66.7±0.9 | 68.6±0.8 | 67.4±0.8 | 67.4±0.6 | 65.8±0.7 |
| DirSwitch-3 | 82.8±1.3 | 83.5±1.4 | **85.3±1.0** | **86.1±1.3** | 85.7±1.4 | **85.4±1.3** | 88.2±1.1 | 87.1±0.9 | 88.4±0.8 | 87.5±1.1 | 64.9±0.9 | 67.1±0.9 | 64.5±0.9 | 64.7±0.5 | **62.2±0.6** |

| Edge directions | Roman Empire | | | | | Pokec | | | | | Snap-Patents | | | | |
|---|---|---|---|---|---|---|---|---|---|---|---|---|---|---|---|
| | Geom | Pois | Geom-$\mathcal{U}$ | Binom-3 | Geom-4 | Geom | Pois | Geom-$\mathcal{U}$ | Binom-3 | Geom-4 | Geom | Pois | Geom-$\mathcal{U}$ | Binom-3 | Geom-4 |
| Default | 69.3±0.9 | **32.2±0.6** | **67.0±0.5** | **49.9±0.6** | **69.6±0.6** | **60.3±0.1** | **59.0±0.8** | **61.6±0.1** | **60.7±0.2** | 65.5±0.1 | 35.6±0.1 | 41.7±0.1 | 39.1±0.0 | 40.8±0.1 | 40.9±0.1 |
| Undirected | **67.4±0.9** | 47.0±0.9 | 69.3±0.7 | 63.7±0.6 | **70.0±0.7** | **60.3±0.1** | 59.4±0.4 | 62.0±0.1 | 62.3±0.1 | **71.7±0.1** | **30.8±0.1** | **29.9±0.0** | **31.7±0.1** | **32.7±0.1** | **33.9±0.4** |
| MultiDir-1 | 70.1±0.8 | 40.7±0.8 | 68.8±0.5 | 59.6±0.8 | 71.5±0.6 | 61.2±0.1 | 60.2±0.0 | 62.3±0.1 | 61.7±0.1 | 67.3±0.1 | 40.9±0.0 | 43.4±0.3 | 43.9±0.2 | 46.7±0.1 | 45.2±0.1 |
| MultiDir-2 | 71.6±0.6 | 63.2±0.7 | 72.0±0.6 | 66.7±0.5 | 75.5±0.7 | 61.6±0.1 | 61.2±0.2 | 62.7±0.1 | 67.9±0.1 | | 42.9±0.0 | 44.0±0.1 | 45.3±0.2 | **47.4±0.2** | 45.0±0.1 |
| MultiDir-3 | **72.0±0.5** | 69.1±0.5 | 73.0±0.8 | 67.4±0.7 | 72.5±0.4 | **61.8±0.1** | **63.5±0.1** | 63.1±0.1 | 62.3±0.1 | 62.4±0.2 | 44.2±0.1 | 47.2±0.1 | 45.2±0.0 | 46.4±0.1 | 44.8±0.1 |
| DirSwitch-1 | 71.6±0.5 | 57.1±0.9 | 73.8±0.5 | 69.2±0.7 | 75.3±0.7 | 61.2±0.1 | 60.4±0.3 | 62.8±0.1 | 62.8±0.1 | 69.9±0.0 | 40.7±0.0 | 40.8±0.1 | 40.4±0.1 | 41.6±0.1 | 42.7±0.4 |
| DirSwitch-2 | 71.9±0.5 | 65.5±0.7 | **74.7±0.4** | **70.6±0.7** | **76.1±0.6** | 61.5±0.1 | 60.9±0.2 | **63.5±0.1** | **63.6±0.1** | 69.9±0.1 | 44.2±0.2 | 46.0±0.1 | 44.7±0.1 | 44.9±0.2 | 45.2±0.3 |
| DirSwitch-3 | 71.8±0.5 | **69.1±0.5** | 74.3±0.5 | 70.1±0.4 | 72.7±0.5 | 61.6±0.1 | 62.7±0.1 | 62.9±0.1 | 62.1±0.1 | **62.0±0.1** | **44.4±0.1** | **48.8±0.1** | **46.3±0.1** | 46.9±0.2 | **46.9±0.2** |

Lastly, we compare DirSwitch-$r$ against the state-of-the-art proximity embedding methods from Section 6.4. We reuse the same hyperparameters as in the node classification setting, without performing additional tuning (see Section D.2).

The results are shown in Table 22. DirSwitch and APP (Zhou et al., 2017) achieve the best overall performance. APP obtains higher NMI scores on EU-Email, CoCite, and Pubmed, while DirSwitch performs best on Fly Larva. The methods are tied on Cora (subelj). As before, we do not consider Polblogs due to its high variance.

As discussed in Section 5, APP is a sampled random walk method that asymptotically performs the same matrix decomposition as proximity DirSwitch with $\boldsymbol{\sigma} = 0$ and $P_w = \text{Geom}(\tau)$. It is therefore noteworthy that APP outperforms DirSwitch in some cases.

One possible explanation is that the hyperparameters—tuned for node classification—may happen to favour APP in the community detection setting. Another plausible explanation is that APP benefits from random walk sampling itself. Similar effects have been observed in recommendation systems, where sampling-based matrix factorization often outperforms direct SVD due to reduced overfitting (Koren et al., 2009). Further research is needed to better understand the relationship between the sampled factorization in APP and the explicit decomposition used in proximity ReachNEs.

Table 17: Relative improvements in node classification accuracy for message-passing ReachNEs with $p = 512$. The values reflect the maximum accuracy for per row and dataset in Table 16. The top row displays absolute accuracies for the default edge directions, with standard deviations expressed as percentages. The subsequent rows present the relative improvements compared to the top row. The table is structured with the four homophilic datasets on the left and the four heterophilic datasets on the right.

| Edge directions | Cora-ML | Citeseer | Cora | Arxiv | Roman Empire | Arxiv Year | Pokec | Snap-Patents |
|---|---|---|---|---|---|---|---|---|
| Default | 73.5±2.9% | 78.2±2.6% | 61.1±1.3% | 55.4±0.5% | 69.6±0.9% | 38.2±0.8% | 65.5±0.4% | 41.7±0.2% |
| Undirected | +13.9% | **+20.1%** | +14.6% | +25.9% | +0.7% | +0.3% | **+9.5%** | -18.8% |
| MultiDir-1 | +8.1% | +6.0% | +7.1% | +8.7% | +2.8% | +3.1% | +2.8% | +12.0% |
| MultiDir-2 | +14.8% | +10.8% | +11.4% | +17.8% | +8.5% | +7.8% | +3.8% | +13.7% |
| MultiDir-3 | +16.0% | +12.6% | +9.9% | +18.2% | +5.0% | +10.4% | -3.1% | +13.1% |
| DirSwitch-1 | +16.0% | +17.0% | +13.3% | +25.4% | +8.3% | +7.9% | +6.8% | +2.4% |
| DirSwitch-2 | +16.7% | +15.2% | +12.2% | +24.3% | **+9.4%** | +8.7% | +6.8% | +10.4% |
| DirSwitch-3 | **+17.3%** | +13.1% | +9.8% | +22.1% | +6.8% | **+10.5%** | -3.9% | **+17.1%** |

Table 18: Node classification accuracy for proximity ReachNEs with $p = 512$ embedding dimensions. Columns correspond to different datasets and multi-scale walk length distributions, while rows represent various edge direction specifiers. The values denote average accuracies with standard deviations. Bold blue highlights the best results in each column, with light blue indicating results within one standard deviation. Similarly, bold orange marks the worst results, with light orange showing values within one standard deviation of the lowest performance.

| Edge directions | Flylarva | | | | | EU-Email | | | | | Polblogs | | | | |
|---|---|---|---|---|---|---|---|---|---|---|---|---|---|---|---|
| | Geom | Pois | Geom-$\mathcal{U}$ | Binom-3 | Geom-4 | Geom | Pois | Geom-$\mathcal{U}$ | Binom-3 | Geom-4 | Geom | Pois | Geom-$\mathcal{U}$ | Binom-3 | Geom-4 |
| Default | 45.3±1.8 | 46.1±2.4 | 48.8±1.9 | 49.2±2.0 | 49.8±2.0 | 46.1±3.1 | 56.6±3.5 | 64.2±3.5 | 68.8±3.4 | 70.4±3.6 | 71.7±2.7 | 74.7±2.2 | 82.6±2.2 | 85.0±2.1 | 85.6±1.9 |
| Undirected | 47.8±2.0 | 50.4±1.9 | 53.6±2.0 | 55.1±1.7 | 54.9±2.0 | 50.3±3.8 | 58.6±3.3 | 70.1±3.2 | 75.3±2.7 | 75.8±2.6 | 77.0±2.6 | 79.2±2.7 | 85.6±1.8 | 87.4±2.1 | 87.0±1.9 |
| MultiDir-1 | 48.5±1.9 | 48.5±2.1 | 51.4±2.3 | 52.5±1.9 | 52.0±1.8 | 65.4±3.0 | 67.5±3.1 | 73.7±3.1 | 73.8±3.0 | 73.7±2.7 | 81.7±1.9 | 83.1±2.1 | 86.4±2.0 | 87.7±1.7 | 88.4±1.7 |
| MultiDir-2 | 51.8±2.3 | 51.4±1.7 | 53.2±1.9 | 52.7±2.3 | 52.7±2.1 | 74.0±2.7 | 74.5±2.7 | 75.2±3.3 | 72.5±2.9 | 70.4±2.7 | 87.9±1.8 | 87.6±1.7 | 89.1±1.7 | 89.2±1.6 | 89.6±1.6 |
| MultiDir-3 | 53.4±1.9 | 53.5±2.1 | 53.7±1.9 | 54.0±1.6 | 54.7±1.5 | 73.9±2.3 | 75.3±3.0 | 70.0±2.5 | 66.4±3.4 | 63.6±2.7 | 89.0±1.4 | 89.5±1.3 | 89.8±1.8 | 89.5±2.0 | 89.2±1.4 |
| DirSwitch-1 | 50.0±2.3 | 51.5±1.7 | 54.2±1.7 | 56.1±2.1 | 55.4±1.5 | 66.4±3.3 | 69.5±3.4 | 74.9±2.9 | 75.7±2.3 | 75.4±2.6 | 83.7±2.2 | 84.1±1.8 | 88.3±2.1 | 87.5±2.0 | 89.7±1.5 |
| DirSwitch-2 | 52.4±2.1 | 52.8±2.0 | 55.2±2.0 | 55.9±2.1 | 55.5±1.9 | 73.6±2.8 | 74.0±3.1 | 74.9±2.7 | 72.6±2.5 | 70.1±2.7 | 87.6±2.1 | 88.3±2.0 | 89.5±1.8 | 87.8±1.8 | 90.0±1.6 |
| DirSwitch-3 | 54.2±1.9 | 54.6±1.6 | 54.9±1.9 | 56.2±1.7 | 56.0±2.2 | 74.8±3.0 | 74.6±2.7 | 70.5±3.0 | 66.2±3.0 | 63.9±3.3 | 88.8±1.7 | 89.4±1.6 | 89.7±1.6 | 89.7±1.4 | 90.0±1.9 |

| Edge directions | CoCite | | | | | Pubmed | | | | | Cora (subelj) | | | | |
|---|---|---|---|---|---|---|---|---|---|---|---|---|---|---|---|
| | Geom | Pois | Geom-$\mathcal{U}$ | Binom-3 | Geom-4 | Geom | Pois | Geom-$\mathcal{U}$ | Binom-3 | Geom-4 | Geom | Pois | Geom-$\mathcal{U}$ | Binom-3 | Geom-4 |
| Default | 37.6±0.5 | 38.2±0.6 | 38.9±0.6 | 38.9±0.6 | 38.9±0.5 | 71.2±0.7 | 71.0±0.7 | 71.3±0.6 | 70.7±0.7 | 69.9±0.7 | 55.8±0.9 | 55.8±0.8 | 56.4±0.8 | 56.5±0.8 | 56.3±0.6 |
| Undirected | 44.7±0.5 | 45.7±0.4 | 46.4±0.5 | 47.7±0.6 | 47.3±0.5 | 81.5±0.6 | 81.8±0.6 | 81.6±0.6 | 82.1±0.6 | 81.5±0.7 | 65.1±0.7 | 65.7±0.8 | 66.1±0.8 | 66.4±0.8 | 66.3±0.7 |
| MultiDir-1 | 38.8±0.6 | 39.3±0.6 | 40.0±0.5 | 40.1±0.5 | 40.3±0.5 | 71.5±0.6 | 71.0±0.6 | 70.2±0.7 | 68.9±0.7 | 68.4±0.8 | 55.5±0.7 | 55.6±0.9 | 55.5±0.7 | 55.3±0.7 | 55.3±0.9 |
| MultiDir-2 | 41.5±0.6 | 42.3±0.5 | 42.9±0.7 | 44.0±0.5 | 44.0±0.5 | 76.2±0.6 | 76.0±0.7 | 75.1±0.7 | 74.4±0.8 | 74.1±0.6 | 58.1±0.7 | 58.6±0.6 | 58.6±0.8 | 58.7±0.7 | 58.9±0.7 |
| MultiDir-3 | 42.3±0.5 | 43.2±0.5 | 44.3±0.5 | 45.9±0.4 | 45.6±0.3 | 78.3±0.6 | 78.0±0.7 | 77.0±0.7 | 76.7±0.8 | 75.9±0.7 | 58.8±0.9 | 59.3±0.9 | 59.0±0.7 | 58.8±1.0 | 59.7±0.8 |
| DirSwitch-1 | 44.3±0.5 | 45.4±0.4 | 46.2±0.5 | 47.8±0.5 | 47.3±0.6 | 81.1±0.7 | 81.3±0.6 | 81.2±0.6 | 81.5±0.6 | 80.6±0.7 | 63.4±0.8 | 64.2±0.7 | 64.6±0.8 | 65.3±0.7 | 65.0±0.7 |
| DirSwitch-2 | 43.1±0.6 | 44.5±0.5 | 45.4±0.4 | 48.0±0.4 | 46.8±0.5 | 80.8±0.6 | 81.0±0.7 | 80.5±0.6 | 80.9±0.7 | 80.0±0.6 | 61.1±0.8 | 62.4±0.7 | 62.6±0.8 | 64.0±0.8 | 63.4±0.8 |
| DirSwitch-3 | 41.8±0.5 | 43.5±0.5 | 44.5±0.5 | 47.1±0.5 | 46.3±0.5 | 80.1±0.7 | 80.4±0.7 | 80.0±0.7 | 79.8±0.6 | 78.6±0.7 | 58.0±0.7 | 59.6±0.7 | 60.1±0.8 | 61.0±0.9 | 61.2±0.8 |

Table 19: Relative improvements in node classification accuracy for proximity ReachNEs with $p = 512$. The values reflect the maximum accuracy for per row and dataset in Table 18. The top row displays absolute accuracies for the default edge directions, with standard deviations expressed as percentages. The subsequent rows present the relative improvements compared to the top row. The table is structured with the three denser graphs on the left and the three sparser graphs on the right.

| Edge directions | FlyLarva | EU-Email | Polblogs | CoCite | Pubmed | Cora (subelj) |
|---|---|---|---|---|---|---|
| Default | 49.8±4.1% | 70.4±4.8% | 85.6±2.6% | 38.9±1.4% | 71.3±0.9% | 56.5±1.4% |
| Undirected | +10.7% | **+7.8%** | +2.0% | +22.4% | **+15.1%** | **+17.7%** |
| MultiDir-1 | +5.4% | +4.9% | +3.2% | +3.5% | +0.3% | -1.5% |
| MultiDir-2 | +6.8% | +6.8% | +4.6% | +13.1% | +6.9% | +4.4% |
| MultiDir-3 | +9.8% | +7.1% | +4.8% | +17.8% | +9.8% | +5.8% |
| DirSwitch-1 | **+12.8%** | +7.6% | +4.8% | +22.9% | +14.3% | +15.6% |
| DirSwitch-2 | +12.4% | +6.5% | **+5.1%** | **+23.2%** | +13.5% | +13.4% |
| DirSwitch-3 | **+12.8%** | +6.4% | **+5.1%** | +20.9% | +12.7% | +8.3% |

Table 20: Community detection normalized mutual information (NMI) scores for proximity ReachNEs with $p = 512$ embedding dimensions. Columns correspond to different datasets and multi-scale walk length distributions, while rows represent various edge direction specifiers. The values denote average NMI with standard deviations. Bold blue highlights the best results in each column, with light blue indicating results within one standard deviation. Similarly, bold orange marks the worst results, with light orange showing values within one standard deviation of the lowest performance.

| Edge directions | Fly Larva | | | | | EU-Email | | | | | Polblogs | | | | |
|---|---|---|---|---|---|---|---|---|---|---|---|---|---|---|---|
| | Geom | Pois | Geom-$\mathcal{U}$ | Binom-3 | Geom-4 | Geom | Pois | Geom-$\mathcal{U}$ | Binom-3 | Geom-4 | Geom | Pois | Geom-$\mathcal{U}$ | Binom-3 | Geom-4 |
| Default | 54.2±0.5 | 55.0±0.4 | 55.7±0.4 | 56.1±0.4 | 56.2±0.3 | 51.6±3.0 | 60.7±1.1 | 59.7±1.1 | 61.2±1.0 | 61.3±0.9 | 5.8±11.0 | 11.3±15.3 | 15.1±17.9 | 13.3±18.4 | 18.1±18.7 |
| Undirected | 54.9±0.4 | 56.1±0.3 | 56.5±0.3 | 57.7±0.3 | 57.7±0.4 | 56.4±4.9 | 64.4±1.5 | 65.8±1.2 | 64.5±1.0 | 65.3±0.8 | 16.9±17.6 | 21.3±12.7 | 25.7±15.8 | 28.2±16.6 | 30.1±17.9 |
| MultiDir-1 | 55.3±0.5 | 56.0±0.4 | 56.6±0.5 | 56.7±0.5 | 57.2±0.3 | 57.9±2.0 | 60.4±1.9 | 61.9±1.0 | 63.0±1.1 | 63.1±0.8 | 25.8±14.9 | 35.0±13.3 | 36.8±10.5 | 34.6±12.2 | 38.7±12.4 |
| MultiDir-2 | 56.4±0.4 | 57.1±0.3 | 57.6±0.4 | 57.3±0.4 | 58.0±0.3 | 61.8±1.2 | 65.0±1.4 | 65.1±0.8 | 65.1±0.6 | 64.2±0.4 | 40.2±17.5 | 47.1±10.8 | 44.7±15.3 | 41.8±14.5 | 45.8±10.3 |
| MultiDir-3 | 57.0±0.5 | 57.7±0.4 | 58.4±0.4 | 57.6±0.3 | 58.3±0.2 | 64.0±0.8 | 66.5±0.8 | 64.7±0.5 | 60.0±0.6 | 58.6±0.6 | 27.0±16.6 | 41.9±17.1 | 43.3±16.2 | 46.7±10.4 | 45.1±14.5 |
| DirSwitch-1 | 55.5±0.3 | 56.6±0.5 | 57.2±0.3 | 58.2±0.4 | 58.2±0.4 | 59.8±2.3 | 64.0±1.0 | 65.0±1.1 | 64.9±0.9 | 65.6±0.6 | 17.5±13.3 | 23.8±15.5 | 31.2±17.9 | 25.7±16.1 | 26.8±16.8 |
| DirSwitch-2 | 56.4±0.4 | 57.2±0.4 | 58.0±0.3 | 58.9±0.4 | 59.0±0.3 | 62.8±1.3 | 66.0±1.0 | 65.8±0.8 | 65.3±0.8 | 64.7±0.5 | 23.0±15.6 | 30.2±16.3 | 32.8±18.1 | 29.5±17.8 | 29.8±17.3 |
| DirSwitch-3 | 57.1±0.4 | 57.8±0.6 | 58.9±0.3 | 59.2±0.3 | 59.8±0.4 | 64.0±0.9 | 66.7±0.6 | 65.0±0.5 | 60.4±0.5 | 59.3±0.8 | 27.3±17.3 | 45.4±16.7 | 32.9±16.6 | 29.3±16.3 | 30.5±15.5 |

| Edge directions | CoCite | | | | | Pubmed | | | | | Cora (subelj) | | | | |
|---|---|---|---|---|---|---|---|---|---|---|---|---|---|---|---|
| | Geom | Pois | Geom-$\mathcal{U}$ | Binom-3 | Geom-4 | Geom | Pois | Geom-$\mathcal{U}$ | Binom-3 | Geom-4 | Geom | Pois | Geom-$\mathcal{U}$ | Binom-3 | Geom-4 |
| Default | 14.9±2.3 | 16.7±1.7 | 17.0±1.5 | 17.1±1.8 | 17.2±2.2 | 2.7±0.9 | 2.7±1.0 | 2.7±1.0 | 2.4±0.6 | 2.6±1.3 | 31.4±1.1 | 31.8±0.8 | 32.2±0.7 | 33.7±0.7 | 33.3±0.6 |
| Undirected | 20.4±1.1 | 22.4±1.0 | 23.9±1.0 | 28.0±0.5 | 26.7±0.9 | 11.9±4.1 | 15.4±2.9 | 17.7±2.4 | 21.6±4.0 | 20.6±3.0 | 43.2±0.4 | 45.9±0.4 | 46.8±0.4 | 49.8±0.2 | 48.6±0.3 |
| MultiDir-1 | 16.7±0.6 | 17.3±0.5 | 17.9±0.7 | 19.3±0.6 | 18.7±0.6 | 4.5±1.2 | 4.6±1.4 | 4.9±1.4 | 5.1±1.5 | 5.8±0.8 | 31.5±0.8 | 31.7±0.9 | 31.9±0.6 | 33.0±0.7 | 32.4±0.5 |
| MultiDir-2 | 18.1±0.7 | 19.2±0.6 | 20.1±0.7 | 22.6±0.6 | 22.4±0.4 | 5.7±1.4 | 6.2±1.3 | 6.6±1.1 | 7.7±1.4 | 7.5±1.0 | 35.8±0.8 | 36.4±0.8 | 35.6±0.7 | 36.2±0.5 | 35.7±0.7 |
| MultiDir-3 | 18.9±0.9 | 20.2±0.9 | 21.6±0.6 | 23.9±0.7 | 23.5±0.4 | 6.9±2.1 | 8.0±2.1 | 8.2±1.5 | 9.6±1.7 | 8.1±0.6 | 36.8±0.7 | 38.5±0.5 | 37.7±0.5 | 37.8±0.5 | 38.1±0.6 |
| DirSwitch-1 | 19.8±1.2 | 21.8±1.0 | 23.4±1.2 | 28.8±0.9 | 27.8±1.0 | 12.0±3.9 | 14.4±2.8 | 15.1±2.0 | 21.3±3.8 | 18.7±3.0 | 43.0±0.4 | 45.3±0.5 | 46.0±0.4 | 49.6±0.3 | 48.1±0.3 |
| DirSwitch-2 | 19.7±1.2 | 21.3±1.0 | 24.0±1.3 | 29.6±0.7 | 27.7±0.8 | 8.8±3.2 | 12.1±2.3 | 11.8±2.4 | 19.0±4.1 | 15.3±3.3 | 41.9±0.6 | 44.1±0.4 | 44.7±0.6 | 47.9±0.4 | 46.5±0.3 |
| DirSwitch-3 | 18.5±0.9 | 21.1±1.0 | 23.9±1.1 | 29.3±0.7 | 28.0±0.5 | 8.9±3.1 | 11.0±2.5 | 10.5±3.1 | 17.3±2.9 | 13.6±3.3 | 38.5±0.5 | 41.7±0.5 | 41.7±0.4 | 44.6±0.3 | 43.6±0.3 |

Table 21: Community detection normalized mutual information (NMI) scores for proximity ReachNEs with $p = 1024$ embedding dimensions. Columns correspond to different datasets and multi-scale walk length distributions, while rows represent various edge direction specifiers. The values denote average NMI with standard deviations. Bold blue highlights the best results in each column, with light blue indicating results within one standard deviation. Similarly, bold orange marks the worst results, with light orange showing values within one standard deviation of the lowest performance.

| Edge directions | Fly Larva | | | | | EU-Email | | | | | Polblogs | | | | |
|---|---|---|---|---|---|---|---|---|---|---|---|---|---|---|---|
| | Geom | Pois | Geom-$\mathcal{U}$ | Binom-3 | Geom-4 | Geom | Pois | Geom-$\mathcal{U}$ | Binom-3 | Geom-4 | Geom | Pois | Geom-$\mathcal{U}$ | Binom-3 | Geom-4 |
| Default | 52.9±0.5 | 54.3±0.5 | 55.0±0.4 | 55.6±0.4 | 55.7±0.4 | 54.0±1.8 | 59.1±1.6 | 60.0±1.3 | 62.1±1.3 | 60.5±1.0 | 11.1±15.3 | 13.8±17.9 | 16.4±18.4 | 12.4±18.3 | 13.3±18.0 |
| Undirected | 54.0±0.6 | 55.7±0.4 | 55.9±0.3 | 57.5±0.4 | 57.5±0.3 | 55.2±2.6 | 63.5±1.1 | 63.8±1.3 | 63.8±0.9 | 64.6±0.9 | 11.8±15.9 | 25.5±19.7 | 24.0±16.0 | 22.3±17.7 | 28.4±18.2 |
| MultiDir-1 | 54.4±0.5 | 55.4±0.5 | 55.9±0.4 | 56.5±0.4 | 56.8±0.4 | 51.9±4.1 | 61.5±1.7 | 62.1±1.1 | 62.5±1.0 | 63.0±1.0 | 20.3±18.5 | 29.6±13.7 | 28.0±15.1 | 36.1±15.6 | 38.2±14.3 |
| MultiDir-2 | 55.7±0.4 | 56.6±0.3 | 57.0±0.4 | 57.1±0.4 | 57.6±0.4 | 59.3±1.7 | 63.1±1.5 | 64.3±1.2 | 65.8±0.9 | 65.9±0.8 | 33.8±21.7 | 41.8±17.2 | 36.0±19.3 | 43.6±15.3 | 45.9±13.5 |
| MultiDir-3 | 56.3±0.5 | 57.1±0.3 | 57.6±0.3 | 57.7±0.4 | 58.3±0.4 | 61.9±1.5 | 65.8±1.0 | 65.7±0.9 | 65.5±0.5 | 64.3±0.6 | 28.3±17.9 | 43.6±17.8 | 41.7±19.4 | 48.5±11.1 | 45.0±15.4 |
| DirSwitch-1 | 54.7±0.4 | 56.2±0.4 | 56.7±0.3 | 57.8±0.3 | 57.9±0.3 | 53.6±5.3 | 64.5±1.3 | 65.0±1.3 | 64.7±1.1 | 65.0±1.3 | 11.2±14.1 | 22.0±13.7 | 23.9±15.2 | 29.5±18.0 | 25.6±15.9 |
| DirSwitch-2 | 55.6±0.4 | 56.6±0.5 | 57.3±0.4 | 58.5±0.4 | 58.5±0.3 | 59.5±1.8 | 63.3±1.3 | 64.7±1.1 | 65.0±0.8 | 65.7±0.7 | 16.2±14.5 | 23.2±13.3 | 31.7±17.8 | 27.9±16.9 | 28.8±17.2 |
| DirSwitch-3 | 56.5±0.3 | 57.2±0.3 | 58.0±0.3 | 58.9±0.3 | 59.2±0.3 | 62.4±1.7 | 65.7±1.1 | 65.8±0.8 | 65.2±0.5 | 64.6±0.7 | 23.3±15.4 | 35.5±18.0 | 31.9±18.7 | 38.4±17.4 | 27.7±17.6 |

| Edge directions | CoCite | | | | | Pubmed | | | | | Cora (subelj) | | | | |
|---|---|---|---|---|---|---|---|---|---|---|---|---|---|---|---|
| | Geom | Pois | Geom-$\mathcal{U}$ | Binom-3 | Geom-4 | Geom | Pois | Geom-$\mathcal{U}$ | Binom-3 | Geom-4 | Geom | Pois | Geom-$\mathcal{U}$ | Binom-3 | Geom-4 |
| Default | 14.7±3.0 | 17.1±1.5 | 15.7±2.7 | 17.4±1.5 | 16.7±2.7 | 2.7±0.9 | 2.5±1.0 | 2.8±1.1 | 2.4±0.9 | 2.3±0.9 | 30.5±1.6 | 30.9±1.4 | 31.7±0.8 | 33.8±0.7 | 33.3±0.9 |
| Undirected | 20.0±1.3 | 22.7±0.8 | 23.4±1.1 | 27.6±0.8 | 26.2±1.0 | 10.8±2.9 | 15.0±3.2 | 16.5±3.8 | 21.1±4.1 | 20.6±2.7 | 42.9±0.6 | 45.6±0.5 | 46.4±0.4 | 49.8±0.3 | 49.0±0.4 |
| MultiDir-1 | 17.0±0.6 | 17.2±0.7 | 17.7±0.6 | 19.6±0.5 | 19.2±0.5 | 3.8±1.9 | 4.3±1.6 | 5.1±1.5 | 5.6±1.8 | 5.4±1.8 | 31.6±0.9 | 31.5±1.1 | 31.6±0.9 | 33.0±0.8 | 32.6±0.7 |
| MultiDir-2 | 18.2±0.5 | 19.3±0.6 | 20.1±0.7 | 22.8±0.5 | 22.1±0.7 | 5.1±1.7 | 6.8±1.2 | 6.4±1.1 | 7.2±2.3 | 7.1±1.6 | 36.7±0.8 | 37.0±0.8 | 36.2±0.7 | 37.1±0.7 | 36.4±0.6 |
| MultiDir-3 | 19.2±0.9 | 20.0±0.8 | 21.0±0.6 | 24.1±0.6 | 23.4±0.4 | 6.6±1.6 | 7.2±1.4 | 8.1±1.8 | 10.7±1.7 | 8.8±1.5 | 38.8±0.6 | 39.8±0.5 | 39.3±0.5 | 39.4±0.5 | 38.9±0.4 |
| DirSwitch-1 | 19.3±1.3 | 21.6±0.9 | 22.8±1.0 | 27.6±1.0 | 26.8±1.1 | 11.1±4.0 | 14.3±3.0 | 15.5±3.6 | 21.7±3.2 | 19.6±3.3 | 42.7±0.5 | 45.1±0.4 | 46.4±0.4 | 49.7±0.3 | 48.4±0.3 |
| DirSwitch-2 | 19.3±1.0 | 21.1±1.0 | 23.6±0.8 | 28.6±1.2 | 27.5±1.2 | 10.3±3.8 | 13.2±3.2 | 13.9±2.5 | 21.2±4.1 | 17.5±3.9 | 42.4±0.4 | 44.6±0.4 | 45.5±0.3 | 49.3±0.2 | 47.8±0.2 |
| DirSwitch-3 | 19.3±1.2 | 21.0±1.0 | 23.6±1.3 | 29.4±0.6 | 27.5±0.6 | 8.7±3.3 | 10.8±2.8 | 11.3±2.6 | 19.2±3.5 | 14.3±3.3 | 40.9±0.5 | 43.3±0.4 | 44.1±0.5 | 47.8±0.3 | 46.2±0.3 |

Table 22: Community detection normalized mutual information (NMI) scores for proximity node embedding models. Bold indicate the top NMI scores, and results within one standard deviation, for each dataset. Average and standard deviations are calculated over 10 different $k$-means initialisaions, and 5 different random seeds for the embedding models.

| Model | Fly Larva | EU-Email | Polblogs | CoCite | Pubmed | Cora (subelj) |
|---|---|---|---|---|---|---|
| HOPE[1] | 46.3 ± 0.8 | 19.9 ± 0.5 | 9.4 ± 1.3 | 2.3 ± 0.4 | 0.4 ± 0.2 | 12.4 ± 1.1 |
| APP[2] | 54.9 ± 0.4 | 68.7 ± 1.0 | 41.0 ± 12.9 | 31.7 ± 0.6 | 23.9 ± 0.2 | 50.0 ± 0.3 |
| NERD[3] | 50.8 ± 0.4 | 50.0 ± 1.5 | 15.1 ± 16.4 | 3.3 ± 0.0 | 4.9 ± 4.6 | 22.0 ± 0.6 |
| DGGAN[4] | 35.2 ± 0.7 | 21.1 ± 0.4 | 1.4 ± 0.0 | 3.6 ± 0.0 | 0.2 ± 0.1 | 4.2 ± 0.0 |
| BLADE[5] | 54.3 ± 0.8 | 42.5 ± 1.2 | 19.6 ± 9.6 | 4.1 ± 0.6 | 0.6 ± 0.4 | 9.3 ± 0.6 |
| DirSwitch | 59.6 ± 0.7 | 65.0 ± 0.7 | 26.8 ± 16.4 | 27.8 ± 0.9 | 21.6 ± 3.8 | 49.7 ± 0.3 |

[1] Ou et al. (2016)  [2] Zhou et al. (2017)  [3] Khosla et al. (2020)  [4] Zhu et al. (2021b)  [5] Virinchi & Saladi (2023)

