# OpenReview forum: "Disobeying Directions:  Switching Random Walk Filters for Unsupervised Node Embedding Learning on Directed Graphs"
_TMLR — Accepted by TMLR_

### Review · Reviewer_DJyF · 2025-04-25

**Summary Of Contributions:**

This paper studies the topic of directed graph (digraph) embedding. It focuses on two main phenomena: local sinks and expressivity. This work shows that directed graphs suffer from local sinks and thus have biased information propagation on sink nodes, while also having difficulty in expressing the multiple directed neighborhoods. The authors propose ReachNE to analyze both phenomena. In addition, the authors propose DirSwitch to strike a balance between both challenges. Extensive experimental results show the existence of the challenges and the effectiveness of the proposed DirSwitch.

**Audience:**

Yes

**Broader Impact Concerns:**

No.

**Claims And Evidence:**

No

**Requested Changes:**

Please see 'Weaknesses'.

**Strengths And Weaknesses:**

# Strengths
1. The studied problem is important and fundamental to graph embedding, as the graph structural properties are relevant to all downstream embedding and learning models.
2. The paper is written in a clear way that facilitates understanding. I can easily follow the main ideas and arguments of the paper.
3. The experimental results are very extensive. The experiments cover 14 datasets, some of which are very large scale (>1M nodes). They also cover datasets with various properties, including both homophilic and heterophilic, dense and sparse.

# Weaknesses
1. One major weakness of this paper is that the paper seems unnecessarily long. It would be better if the authors can use more concise writings, especially Sections 1-3. For example, some background information (e.g. Table 1) can be made simpler.
2. Some concepts in this paper are introduced vaguely and with no concrete definitions. This causes confusion when reading this paper. For example:

    -  I find the first several paragraphs in Section 2.3 confusing. The authors say that "proximity embeddings capture structural equivalence, where embedding similarity reflects spatial closeness; ...message passing embeddings represent automorphic equivalence, emphasizing local connectivity patterns and allowing for similarity between distant nodes". There are multiple points of confusion there. First, the difference between 'structural equivalence' and 'automorphic equivalence' can be better illustrated. Second, 'structural equivalence', from my understanding, should indicate the equivalence of a node's local structures, which should be more related to 'allowing for similarity between distant nodes' used to describe message-passing embeddings. Similarly, 'spatial closeness' should also be more related to 'local connectivity patterns', also used to describe message passing embeddings. Therefore, these descriptions lead to a lot of confusion, and it is important that the authors clarify that.
    -   In Section 3.2, the term 'expressivity' is not formally/strictly defined. The authors use node alignment in the experiments, but that is only one notion of 'expressivity'. The authors should better introduce more about 'expressivity', such as a definition, or some examples.

3. Some figures in this paper are not clearly visible. For example, unnecessary nodes in Figure 1 can be removed to highlight the main ones. Figure 2 is so dense that readers cannot very clearly capture its implications.

---

> ### Author Response · Authors · 2025-06-05
> **Rebuttal to Reviewer DjyF**
>
> We thank the reviewer for acknowledging the importance of our work, the clarity of our presentation, and the depth of our experimental analysis. We recognize that the paper is long, and we sincerely appreciate the time and effort you invested in reviewing it.
>
> We also appreciate your constructive suggestions for improvement. Please find our responses below.
>
> - **Length of the paper**:
>
>   We agree that the paper is lengthy. In response to your comment, we revised Sections 1–3 to make the writing more concise. These edits are highlighted in blue in the revised version.
>
>   While this led to a reduction of approximately one page, we have since added new content in response to feedback from all reviewers. As a result, the current version is about 1.5 pages longer than the original. Nevertheless, we hope you appreciate the effort to streamline the early sections, and we ask for your understanding regarding the expanded length driven by the revisions.
>
> - **Confusing presentation in Section 2.3**:
>
>   We agree that the original explanation of how node equivalences relate to the reduction methods lacked clarity. We have now revised and expanded Sections 2.3.1 and 2.3.2 to explicitly clarify these connections.
>
>   Additionally, we have added a new subsection (Section 5.2) to the Related Works, where we discuss how structural and automorphic equivalences have been addressed in prior literature. We hope this resolves any confusion and improves the overall clarity of the presentation.
>
> - **Expressivity not being formally defined**:
>
>   We have reworked Section 3.2 to provide a clearer and more precise introduction to the notion of expressivity. Furthermore, we added Section 5.4 to the Related Works to contextualize our approach within the broader literature on expressivity in graph representation learning.
>
> - **Some figures are not clearly visible**:
>
>   We have revised Figure 1 by removing unnecessary nodes, as suggested. We agree that this significantly improves its clarity.
>
>   Similarly, we have updated Figure 4 to use a more "zoomed-in" view of the graphs, improving focus and legibility.

---

### Review · Reviewer_8Lg8 · 2025-05-12

**Summary Of Contributions:**

This paper proposes new ways to leverage directed edges for unsupervised graph learning. While most node embedding techniques focus on undirected graphs, directed graphs require more involved approaches due to the asymmetric propagation of information. In particular, local sinks (nodes without out-neighbors) and diverse multi-step neighborhood structures (dependent on the direction of each step) are digraph-specific phenomena which must be taken into account.

The authors first introduce Reachability Node Embeddings (ReachNEs), a variant of linearized Graph Convolutional Networks (GCNs) where embeddings stem from a power series of adjacency matrices. Each term in the series corresponds to a number of propagation steps $k$, and the associated coefficient is given by a distribution on the length of the random walk. Crucially, the adjacency matrices considered can be either out-adjacencies, in-adjacencies or undirected adjacencies, and mixing is encouraged. The final embeddings are then obtained using either matrix factorization (in the absence of node features) or message-passing (in the presence of node features).

Then, they explain how ReachNEs shed light on the importance of local sinks, and how they enable multiple neighborhood structures to be considered. Local sinks give rise to low-entropy reachability distributions, which are often mitigated by undirected propagation. To go beyond one-hop neighborhoods, one can make the embeddings multi-directional (by concatenating outputs of different sequences of transition matrices) and multi-scale (by concatenating outputs of random walk distributions with different parameters).

These considerations lead to the proposal of DirSwitch, an embedding generator which starts the random walk with a few directed steps (in or out) before switching to undirected propagation for the remaining layers. Numerical experiments demonstrate the ability of DirSwitch to tackle local sinks while preserving multiple neighborhood structures. The quality of the resulting embeddings is better or comparable to state-of-the-art alternatives.

**Audience:**

Yes

**Broader Impact Concerns:**

Nothing to mention here.

**Claims And Evidence:**

Yes

**Requested Changes:**

## Major changes

- Wherever necessary, clarify which ideas were present in the existing literature. Some locations where this would be useful:
  - The use of random walk length distributions for convolution filters
  - The connection between proximity (resp. message-passing) and structural (resp. automorphic) equivalences

> As a baseline for our experiments, we define MultiDir-r similarly to DirSwitch-r, but without the switch to undirected edges.

- Explain the difference between DirSwitch-$r$ and MultiDir-$r$: the sentence above is not clear enough. What happens instead of the switch to undirected edges? Does the last direction in $\sigma$ get repeated forever? Or does the sequence just stop there, keeping a length of $r < K$?

> Ideally, embeddings should retain all relevant graph information, allowing them to be used independently and efficiently for downstream tasks without requiring access to the full graph. We refer to an embedding model’s ability to preserve graph information as its expressivity.

- Can the expressivity of DirSwitch be evaluated in a more formal way? The evaluations on the graph alignment problem are empirical in nature, a suitable theoretical counterpart might be the Weisfeiler-Lehman test. Rossi et al. (2023) compared their Dir-GNN to a directed variant of WL, which may be relevant here too? _Note: this is a more demanding addition than the other ones in my review and thus I don't regard it as necessary for acceptance, it is mainly driven by curiosity._

## Minor questions

> [P7] Another modification to the definition of R is the application of the elementwise thresholding function

- Does the $1/\log n$ threshold have any special significance here?

> [P13]  Unless otherwise specified, we compute truncated reachability using K = 12 steps.

- Is this consistent with the graph diameters in the datasets?

> [P15] To achieve this, we use ReachNEs message-passing embeddings combined with node attributes derived from graph structure, including out- and in-degrees and the four digraph local clustering coefficients defined by Fagiolo (2007).

- Please give a list of the node attributes you used.

> [P16] Figure 7

- In Figure 7 and others like it, the comparison between DirSwitch-$r$ and MultiDir-$r$ is difficult because the pairs of models don't have the same color or the same markers. Is there a way to make it visually more obvious which pairs go together?

> [P16] In the limit $\tau \to \infty$, DirSwitch embeddings converge to identical representations within a weakly connected component. In contrast, MultiDir embeddings retain finer granularity, as equivalence groups in the directed case remain more distinct.

- Why is that the case?

> [P17] Table 4

- Why are tables 4 and 6 separated? What dictated such a split between the datasets?

## Typos and notation

> [P2] For instance, in Figure 1b, node 1 only receives information from the sink node (node 2).

- Nodes reversed?

> [P30] Unlike standard rSVD, SP-rSVD also computes a second projection $H = R^T G \in \mathbb{R}^{n \times p}$ in the same pass

- $H$ is already used for the embeddings, this might be confusing for readers.

**Strengths And Weaknesses:**

## Strengths

- The paper is exceptionally clear and pedagogical.
- The presented method is very flexible and easy to understand.
- It allows fine-grained control over the way that directed edges are considered, without separating the out- and in-neighborhoods entirely or needing to learn their respective contributions.
- The numerical experiments are thorough and convincing. They cover embeddings with or without node features on a wide variety of benchmarks.
- Whenever some instances do not behave like the rest (e.g. the Roman Empire graph), the authors provide additional details on their individual structure and properties to explain these exceptions.

## Weaknesses

- Due to the thoughtful exposition with lots of context, it is sometimes hard to discern novel contributions from previously-explored ideas.
- The difference between DirSwitch-$r$ and MultiDir-$r$ is not clear enough.
- Unlike the local sink effect, the correct expression of multiple neighborhood structures is never quantified directly

---

> ### Author Response · Authors · 2025-06-06
> **Rebuttal to Reviewer 8Lg8**
>
> Thank you for the time and effort you devoted to reviewing our paper.
> We much appreciate your positive assessment of our pedagogical presentation, thorough experiments, and detailed analysis. This paper has been a substantial undertaking, and your feedback is greatly valued.
>
> We also acknowledge the weaknesses you pointed out and have revised the paper to address them. Below, we summarize the key changes we’ve made in response to your comments.

---

> > ### Comment · Reviewer_8Lg8 · 2025-06-16
> >
> > Thank you, I am satisfied with the revision and rebuttal.

---

> ### Author Response · Authors · 2025-06-06
> **Major Changes**
>
> - **Clarifying Relation to Existing Literature**
>
>   We have substantially revised the Related Works section (Section 5) to better clarify how ReachNEs and DirSwitch relate to prior work. In particular:
>   - Section 5.1 has been reorganized and rewritten for clarity and improved structure, and we now explicitly discuss how our use of walk length distributions differs from previous literature.
>   - Section 5.2 is a new subsection that discusses how structural and automorphic equivalence have been addressed in previous node embedding works.
>   - We also revised Section 2.3 to better highlight how ReachNEs captures both types of node equivalences.
>
> - **Clarifying the Difference Between DirSwitch and MultiDir**
>
>   We updated Section 4 (specifically the first paragraph on page 12) to clearly explain the distinction between DirSwitch and MultiDir.
>   MultiDir operates exactly as you suggested: it continues the walk in the direction specified by the last symbol in $\sigma$ until the walk truncation length $K$.
>
> - **More Formal Evaluation of Expressivity**
>
>   Unfortunately, we were not able to provide a more formal expressivity analysis of DirSwitch at this time. As you rightly point out, the WL expressivity framework used by Rossi et al. (2023) is highly relevant. However, their results rely on the assumption of injective aggregation functions — an assumption that does not hold for ReachNEs, which is fully unsupervised, linear, and lacks non-linear layers or trainable aggregation functions.
>
>   To clarify our position within this landscape, we have added Section 5.4 to the Related Works, where we discuss how expressivity is defined in the graph learning literature and how ReachNEs fits into this picture. We also revised Section 3.2 to improve the clarity of our expressivity definitions — a change that also addresses similar feedback from Reviewer DJyF.

---

> ### Author Response · Authors · 2025-06-06
> **Minor Changes**
>
> - **Clarifying the $1/\log(n)$ Threshold**
>
>   To clarify: the only threshold applied in the algorithm is $1/n$. The interval $[1/n, 1/\log(n)]$ is presented purely to help readers interpret the function $f$.
>   Specifically, $x = 1/\log(n)$ is the point at which the difference between $f(x)/\log n$ and $x$ is maximized, offering the most contrast enhancement. (We divided $f(x)$ by $\log n$ so that $f(x=1)/\log n = 1$).
>
> - **Consistency of $K=12$ Truncation**
>
>   While we did not compute graph diameters (due to high cost), we instead report average path lengths in Table 2. Except for the Roman Empire graph, all graphs have average path lengths below 7.5. Thus, $K=12$ likely covers the vast majority of relevant interactions.
>
>   It's worth noting that the walk length distribution $P_w$ — not $K$ — is what primarily determines the effective number of steps. Therefore, $K$ only matters when it starts truncating the support of $P_w$.
>
> - **List of Node Attributes Used for Graph Alignment**
>
>   We have updated the text on page 17 to include the full list of clustering coefficients used as node attributes.
>
> - **Why MultiDir Retains More Granularity as $\tau \rightarrow \infty$**
>
>   In directed graphs, the stationary distribution is not uniform across nodes in a weakly connected component. Instead, it reflects the structure of *reaches* (see Veerman & Lyons (2020)). This allows MultiDir to preserve more granularity even as $\tau \to \infty$.
>
>   We include a short explanation at the end of Appendix C.1 and added a pointer to this discussion on page 18. For further details, we refer to Veerman & Lyons (2020), which provides a comprehensive treatment of this concept.
>
> - **Why Tables 4 and 6 Are Separated**
>
>   Tables 4 and 6 are separated based on whether the datasets include node attributes. We evaluate message-passing ReachNEs on datasets with node features, and proximity ReachNEs on those without. This distinction naturally leads to the separation of results. We have added a clarification to the **Results overview** paragraph on P18 to make this rationale explicit.

---

> ### Author Response · Authors · 2025-06-06
> **Typos and Notation**
>
> - **Nodes Reversed**
>
>   The sentence *"For instance, in Figure 1b, node 1 only receives information from the sink node (node 2)"* is correct. While it may seem intuitive to interpret edges as representing the direction of information flow, ReachNEs follows a *pull-based* information gathering mechanism.
>
>   That is, a random walk starting from node $i$ explores its neighborhood by *gathering* information from the nodes it visits, not by sending information outward.
>
>   This behavior is detailed in Appendix A.2, where we present the message-passing equations for directed graphs.
>
> - **Notation for SP-rSVD**
>
>   Thank you for pointing this out — we have updated the notation to use $F$ instead of $H$ to avoid confusion with the message-passing hidden representations.

---

### Review · Reviewer_fMrB · 2025-05-22

**Summary Of Contributions:**

1.  **ReachNEs: A Unifying Reachability Framework**
    The authors introduce Reachability Node Embeddings (ReachNEs), a general framework for analyzing and developing node embeddings on digraphs.

2.  **DirSwitch: Switching Random Walks to Mitigate Sink Bias and Represent Neighborhood Multiplicity**
    To specifically address the "local sink" problem (nodes without outgoing edges obstructing information flow) and the challenge of representing multiple distinct directed neighborhoods in digraphs, the paper proposes DirSwitch.

3.  **Comprehensive Empirical Validation**
    The authors conduct extensive experiments on up to 14 node classification benchmark datasets.

4.  **Extension to Self-Supervised GNNs**
    The paper also showcases the flexibility of the DirSwitch concept by applying it as a digraph extension to self-supervised message-passing Graph Neural Networks (GNNs), such as GraphSAGE.

**Audience:**

Yes

**Claims And Evidence:**

Yes

**Requested Changes:**

* **Essential: Enhanced Complexity Analysis for DirSwitch-r:**
    * **Change:** Provide a more explicit analysis of the computational time and memory complexity specifically for the DirSwitch-r model, considering the $2^r$ factor for component generation and concatenation. This could be in the main text or prominently in an appendix. Empirical runtime profiling on some of the larger datasets for different values of $r$ would also be highly valuable.

* **Essential: Guidelines for Hyperparameter Selection:**
    * **Change:** Add a dedicated subsection (or expand an existing one, perhaps in the appendix with clear signposting from the main text) discussing the sensitivity to key hyperparameters like $r$, the choice of $P_w(k)$ and its parameters (e.g., $\tau$), and the embedding dimension $p$. This section should ideally offer some practical advice, rules of thumb, or default recommendations based on the authors' extensive experiments, and discuss the computational versus performance trade-offs.


* **Recommended: Exploration of at Least One Additional Unsupervised Downstream Task:**
    * **Change:** If feasible within a revision timeframe, include experimental results on at least one additional, purely unsupervised downstream task, such as node clustering or link prediction for directed edges.

* **Optional: Metric for Directionality Preservation:**
    * **Change:** Consider discussing the possibility of a metric to quantify how much essential local directional information is preserved by the initial $r$ directed steps, or conversely, what type of directional information might be lost at longer ranges. This is more of a suggestion for future thought/work.

**Strengths And Weaknesses:**

**Strengths:**

* **Clear Motivation:** The paper clearly articulates the challenges of local sinks and neighborhood multiplicity in digraphs. The ReachNEs framework provides a mathematically sound and unifying perspective on random-walk and message-passing embeddings, offering analytical tractability.
* **Intuitive Filter Design:** The core idea of DirSwitch—decoupling local directed steps from global undirected steps—is a simple and effective solution. It directly targets sink-induced bias while preserving essential local directional information, which is a key challenge in digraph representation learning.
* **Insightful Analysis:** The paper offers valuable analyses of its results, such as the differing behavior of models on homophilic versus heterophilic graphs and the influence of graph densit.

**Weaknesses:**

* **Computational Complexity and Scalability of DirSwitch-r:** The DirSwitch-r model concatenates $2^r$ embedding components. While effective, the exponential growth in the number of components with $r$ raises concerns about computational and memory costs, especially when the base embedding dimension $p/(2^r s)$ becomes very small for a fixed $p$ or when $r$ is large. The paper acknowledges diminishing returns but could further detail these practical scalability limits.
* **Hyperparameter Sensitivity and Guidance:** The choice of the switch parameter $r$, the walk length distribution $P_w(k)$ (e.g., Geometric, Poisson) and its parameters (e.g., $\tau$), and the embedding dimension $p$ significantly impacts performance. The paper demonstrates these effects empirically but offers limited prescriptive guidance for selecting these crucial hyperparameters for new datasets beyond grid search.
* **Limited Scope of Downstream Unsupervised Tasks:** The evaluation primarily focuses on node classification (often using labels in a downstream supervised classifier, though embeddings are learned unsupervisedly) and graph alignment. The utility of DirSwitch embeddings in purely unsupervised downstream tasks like community detection, anomaly detection, or link prediction in digraphs remains largely unexplored future work.
* **Impact of Finite Walk Truncation:** While the reachability matrix $R$ is theoretically defined with infinite sums, experiments often use truncated walks (e.g., $K=12$). The potential discrepancy between theoretical asymptotic analyses of $P_w(k)$  and the practical effects of this truncation could be more explicitly discussed.

---

> ### Author Response · Authors · 2025-06-05
> **Rebuttal to Reviewer fMrB**
>
> Thank you for taking the time to review our paper and for expressing appreciation of our motivation, presentation, and analysis. A great deal of effort went into this work, so your positive feedback is especially welcome!
>
> We also value your constructive suggestions and broadly agree with all your proposed improvements. Below, we summarize how we have addressed each of them in the revised manuscript.
>
> - **Essential: Enhanced Complexity Analysis for DirSwitch-$r$**
>
>   We have added **Section A.5** to the Appendix, where we present a full computational complexity analysis of DirSwitch. This includes both theoretical time and space complexity for proximity and message-passing variants, as well as empirical run time measurements on synthetic and real graph datasets.
>
> - **Essential: Guidelines for Hyperparameter Selection**
>
>   We have added **Section A.6** to the Appendix, which provides practical guidance on selecting DirSwitch hyperparameters. This includes a detailed discussion of the trade-off between expressivity and computation, and the diminishing returns associated with increasing $r$ and the number of walk length distributions $s$. We also reference this section at the end of **Section 4**, where we introduce DirSwitch.
>
> - **Recommended: Exploration of at Least One Additional Unsupervised Downstream Task**
>
>   We have added a **community detection experiment** to **Appendix G**, using the same node classification datasets employed for proximity embeddings. Evaluation is done via KMeans clustering and normalized mutual information (NMI). While we recognize the importance of unsupervised tasks, we also highlight the challenges they present — in particular, that many datasets allow for multiple valid clusterings, of which only one is labeled. In such cases, meaningful evaluation often requires more extensive analysis. As a result, the main paper is focused primarily on node classification for its clarity and ease of interpretation.
>
> - **Optional: Metric for Directionality Preservation**
>
>   We greatly appreciate this suggestion and fully agree that quantifying how much directional information is preserved at different values of $r$ is an important avenue for future work. This is also related to the issue of diminishing returns, which we have now expanded upon in **Section 7** (Limitations and Future Work) and in **Appendix A.6**.

---

### Author Response · Authors · 2025-06-05
**Response to the reviewers**

First, we would like to sincerely thank the reviewers and the action editor for the time and effort spent reviewing our paper. We recognize that the manuscript is lengthy, which makes us especially appreciative of your detailed and thoughtful feedback — thank you!

We have now uploaded a revised version of the paper in which we have addressed all reviewer comments to the best of our ability. For convenience, all changes to the main paper are highlighted in blue. We have also added several new sections to the appendix, which are not highlighted.

For a detailed account of the changes, please refer to our individual responses to each reviewer below.

---

### Decision · Action_Editor_XsSe · 2025-06-17

**Recommendation:** Accept as is

**Audience:**

Yes

**Audience Explanation:**

Yes, the focus area of the paper (learning on directed graphs) is certainly an area of interest for graph machine learning -- especially in many important industrial contexts. Hence there is certainly a solid group of TMLR readers that would appreciate this result.

**Claims And Evidence:**

Yes

**Claims Explanation:**

The paper -- especially after its revision -- provides a well-motivated and thoroughly validated proposal. In terms of technical quality, this is clearly above the TMLR bar. After the revision, all three reviewers unanimously agree with this conclusion.